# THE CURIOUS CASE OF IN-TRAINING COMPRESSION OF STATE SPACE MODELS

**Makram Chahine**
CSAIL, MIT
Cambridge, MA 02139, USA
chahine@mit.edu

**Philipp Nazari**
Max Planck Institute for Intelligent Systems &
ETH Zürich &
ELLIS Institute Tübingen
philipp.nazari@tuebingen.mpg.de

**Daniela Rus**
CSAIL, MIT
Cambridge, MA 02139, USA
rus@csail.mit.edu

**T. Konstantin Rusch**
ELLIS Institute Tübingen &
Max Planck Institute for Intelligent Systems &
Tübingen AI Center &
Liquid AI
tkrusch@tue.ellis.eu

## ABSTRACT

State Space Models (SSMs), developed to tackle long sequence modeling tasks efficiently, offer both parallelizable training and fast inference. At their core are recurrent dynamical systems that maintain a hidden state, with update costs scaling with the state dimension. A key design challenge is striking the right balance between maximizing expressivity and limiting this computational burden. Control theory, and more specifically Hankel singular value analysis, provides a potent framework for the measure of energy for each state, as well as the balanced truncation of the original system down to a smaller representation with performance guarantees. Leveraging the eigenvalue stability properties of Hankel matrices, we apply this lens to SSMs *during training*, where only dimensions of high influence are identified and preserved. Our approach, COMPRESSM, applies to Linear Time-Invariant SSMs such as Linear Recurrent Units, but is also extendable to selective models. Experiments show that in-training reduction significantly accelerates optimization while preserving expressivity, with compressed models retaining task-critical structure lost by models trained directly at smaller dimension. In other words, SSMs that begin large and shrink during training achieve computational efficiency while maintaining higher performance. Project code is available at github.com/camail-official/compressm.

## 1 INTRODUCTION

State Space Models (SSMs) (Gu et al., 2021; Hasani et al., 2022; Smith et al., 2022; Orvieto et al., 2023; Rusch & Rus, 2025) have recently emerged as a powerful alternative to established sequence models such as Recurrent Neural Networks (RNNs) and Transformers. They combine the parallelizable training efficiency of scaled dot-product attention (Vaswani, 2017) with the computational and memory advantages of RNNs, enabling strong performance across large-scale language, vision, and audio modeling tasks (Gu & Dao, 2024; Goel et al., 2022; Nguyen et al., 2022).

Despite their efficient structure and recent progress in hardware-aware implementations, current SSMs remain computationally intensive. While both memory and runtime scale with sequence length, the size of the SSM state further amplifies these costs. Reducing the state dimension therefore provides an effective strategy to simultaneously reduce memory usage and runtime. This can be achieved by leveraging techniques from structured compression. However, most existing approaches are commonly applied post-training: a large model is trained to completion and only compressed afterwards. Popular examples include knowledge distillation (Hinton et al., 2015), post-training quantization (Jacob et al., 2018), low-rank factorization (Hu et al., 2022), and structured pruning (Li et al., 2016). All of these methods typically require the costly upfront training of a large network.

In this article, we address the issue of costly pre-training by introducing COMPRESSM, a principled in-training compression technique that effectively reduces the dimension of the SSM while largely preserving the expressive power of uncompressed models. We motivate our approach by the control-theoretic origins of SSMs (Kalman, 1960). In particular, we draw on balanced truncation (Antoulas, 2005), a classical Model Order Reduction (MOR) technique that approximates a high-dimensional state space system with a low-dimensional one while retaining its essential input–output behavior. Observing that the dominant Hankel Singular Values (HSVs) of an SSM are rank-preserving during training, we propose truncating dimensions associated with small Hankel singular values once they fall below a predefined relative threshold.

**Main contributions.** In the subsequent sections, we will demonstrate the following features of COMPRESSM:

- We rigorously justify the validity of our in-training compression approach by establishing means to identify and track the HSVs of an SSM during training, and showing that dominant singular values are rank-preserving. Thus, SSM dimensions associated with small HSVs can be safely truncated.
- We show that COMPRESSM is broadly applicable, including to SSMs with structured state matrices such as diagonal matrices, with simple extensions to Linear Time-Varying systems discussed.
- We provide an extensive empirical evaluation demonstrating that COMPRESSM largely preserves the expressive power of uncompressed models.
- COMPRESSM significantly accelerates training, while achieving similar or higher accuracy than larger uncompressed models by truncating large portions of the state early in training.

## 2 MATHEMATICAL PRELIMINARIES

### 2.1 DISCRETE LINEAR TIME INVARIANT SYSTEMS

Let $\mathcal{G}$ be a discrete *Linear Time-Invariant (LTI)* system described by state equations:

$$\begin{aligned} \boldsymbol{h}(k+1) &= \boldsymbol{A}\,\boldsymbol{h}(k) + \boldsymbol{B}\,\boldsymbol{x}(k)\,, \quad \boldsymbol{h}(0) = \boldsymbol{h}_0 \\ \boldsymbol{y}(k) &= \boldsymbol{C}\,\boldsymbol{h}(k) + \boldsymbol{D}\,\boldsymbol{x}(k), \end{aligned} \tag{1}$$

where $\boldsymbol{h} \in \mathbb{R}^n$ is the state, $\boldsymbol{x} \in \mathbb{R}^p$ the input, and $\boldsymbol{y} \in \mathbb{R}^q$ the output, with $\boldsymbol{A} \in \mathbb{R}^{n \times n}$, $\boldsymbol{B} \in \mathbb{R}^{n \times p}$, $\boldsymbol{C} \in \mathbb{R}^{q \times n}$, and $\boldsymbol{D} \in \mathbb{R}^{q \times p}$.

A more general class of systems are *Linear Time-Varying (LTV)*, where the matrices $\boldsymbol{A}, \boldsymbol{B}, \boldsymbol{C}, \boldsymbol{D}$ are functions of time. Such systems become relevant in the context of *selective* SSMs, where the system matrices depend on the input. For now, we restrict to the LTI case as the base framework. The LTV case is discussed in Appendix E.1.

The LTI framework provides a tractable and well-understood setting in which powerful tools such as Gramians and balanced truncation can be developed. Before introducing these concepts, we formalize the standard assumptions of stability, controllability, and observability, which ensure that the system is both well-posed and non-degenerate. For precise definitions of terms and a detailed background presentation we refer the reader to chapters 4, 5, and 6 of Chen (1999).

**Assumption 2.1.** *The system is stable, i.e all the eigenvalues of $\boldsymbol{A}$ are of amplitude less than 1.*

**Assumption 2.2.** *The pair $(\boldsymbol{A}, \boldsymbol{B})$ is controllable, i.e., the state $\boldsymbol{h}$ can be steered from any initial state to any final state in finite time.*

**Assumption 2.3.** *The pair $(\boldsymbol{A}, \boldsymbol{C})$ is observable, i.e., observing the output $\boldsymbol{y}$ and the input $\boldsymbol{x}$ of the system for some finite time suffices to determine the initial state $\boldsymbol{h}_0$.*

#### 2.1.1 CONTROLLABILITY AND OBSERVABILITY GRAMIANS

The concepts of controllability and observability can be captured quantitatively by matrix-valued energy measures called Gramians. These respectively encode how easily internal states can be excited by inputs or observed from outputs.

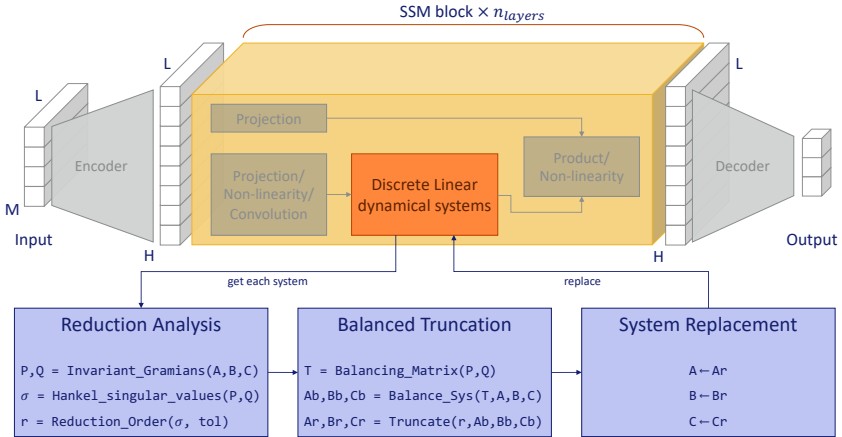

Figure 1: Overview of the proposed balanced truncation pipeline. The method applies at the level of the discrete linear dynamical systems inside SSM layers, independently of surrounding design choices such as projections, non-linearities, convolutions, or skip connections. Each dynamical system is isolated, balanced via its controllability and observability Gramians, and truncated according to Hankel singular values before being reinserted into the model.

Under assumptions 2.1 and 2.2, there exists a unique symmetric positive definite solution $\boldsymbol{P} \in \mathbb{R}^{n \times n}$ to the discrete Lyapunov equation:

$$\boldsymbol{A}\boldsymbol{P}\boldsymbol{A}^\top - \boldsymbol{P} + \boldsymbol{B}\boldsymbol{B}^\top = 0, \qquad\qquad \boldsymbol{P} = \sum_{i=0}^{\infty} \boldsymbol{A}^i \boldsymbol{B}\boldsymbol{B}^\top (\boldsymbol{A}^\top)^i. \qquad (2)$$

$\boldsymbol{P}$ is known as the *discrete controllability Gramian*. It intuitively captures how much energy from the input can reach each state dimension over time. Large entries indicate states that are easily influenced by inputs.

Under assumptions 2.1 and 2.3, there exists a unique symmetric positive definite solution $\boldsymbol{Q} \in \mathbb{R}^{n \times n}$ to the discrete Lyapunov equation:

$$\boldsymbol{A}^\top \boldsymbol{Q}\boldsymbol{A} - \boldsymbol{Q} + \boldsymbol{C}^\top \boldsymbol{C} = 0, \qquad\qquad \boldsymbol{Q} = \sum_{i=0}^{\infty} (\boldsymbol{A}^\top)^i \boldsymbol{C}^\top \boldsymbol{C}\boldsymbol{A}^i. \qquad (3)$$

$\boldsymbol{Q}$ is known as the *discrete observability Gramian*. It similarly measures how much each state contributes to the outputs over time. Large entries correspond to state directions that are easily observed through the outputs.

### 2.1.2 BALANCED REALIZATIONS

The notions of controllability and observability are key for establishing a diagonal balanced realization, in which the system is transformed so that both Gramians are equal and diagonal. This provides a natural coordinate system where each state dimension has a well-defined "importance."

**Definition 2.4** (State Space Realization). A discrete-time linear system $\mathcal{G}$ is fully characterized by its input–output map

$$\mathcal{G} : \{\boldsymbol{x}(k)\}_{k \geq 0} \mapsto \{\boldsymbol{y}(k)\}_{k \geq 0}.$$

A *realization* of $\mathcal{G}$ is any quadruple of matrices $(\boldsymbol{A}, \boldsymbol{B}, \boldsymbol{C}, \boldsymbol{D})$ and state $\boldsymbol{h}(k) \in \mathbb{R}^n$ that *realizes* this input–output map via the dynamics given by Equation 1. Many different realizations can realize the same map. In particular, if $(\boldsymbol{A}, \boldsymbol{B}, \boldsymbol{C}, \boldsymbol{D})$ is a realization, then so is $(\boldsymbol{T}^{-1}\boldsymbol{A}\boldsymbol{T}, \boldsymbol{T}^{-1}\boldsymbol{B}, \boldsymbol{C}\boldsymbol{T}, \boldsymbol{D})$ for any invertible $\boldsymbol{T} \in \mathbb{R}^{n \times n}$.

**Definition 2.5** (Minimal/Balanced realizations). A realization is called *minimal* if it is both controllable and observable. The corresponding state dimension $n$ is called the *order* of the realization.

A realization is said to be *balanced* if $\boldsymbol{P} = \boldsymbol{Q}$. In this case we denote the common matrix by $\boldsymbol{W}$, and refer to it simply as *the Gramian* of the balanced system.

**Theorem 2.6** (Antoulas (2005)). *Any stable, minimal discrete LTI system admits a balanced realization, in which the controllability and observability Gramians coincide as $W = diag(\sigma) = diag(\sigma_1, \ldots, \sigma_n)$, with $\sigma_1 \geq \cdots \geq \sigma_n > 0$ called "Hankel singular values" (HSV). This diagonal balanced realization can be explicitly constructed.*

The HSVs $\sigma$ can also be computed in decreasing order from non-balanced realizations via:

$$\sigma = \text{sort}_\downarrow \left( \sqrt{\text{spec}(PQ)} \right). \tag{4}$$

The HSVs quantify the joint controllability and observability of each state. Large values indicate state directions that both strongly affect the output and are strongly influenced by the input, while small values correspond to weakly contributing states.

### 2.1.3 BALANCED TRUNCATION

Balanced truncation is a MOR scheme leveraging the ordering of Hankel singular values to obtain a lower-dimensional approximation of the system, while guaranteeing stability and error bounds.

Consider a stable minimal balanced realization $(A, B, C, D)$ of $\mathcal{G}$ with Gramian $W = diag(\sigma) = diag(\Sigma_1, \Sigma_2)$ where $\Sigma_1$ is diagonal of size $r$ and $\Sigma_2$ of size $n - r$, with the smallest entry in $\Sigma_1$ larger than the largest in $\Sigma_2$. The state space matrices can be rewritten as:

$$A = \begin{bmatrix} A_{1,1} & A_{1,2} \\ A_{2,1} & A_{2,2} \end{bmatrix}, \ B = \begin{bmatrix} B_1 \\ B_2 \end{bmatrix}, \ C = [C_1 \quad C_2], \tag{5}$$

with $A_{1,1} \in \mathbb{R}^{r \times r}$, $B_1 \in \mathbb{R}^{r \times p}$, and $C_1 \in \mathbb{R}^{q \times r}$.

It is well established that the reduced system $\hat{\mathcal{G}}$ defined by $A_{1,1}, B_1, C_1, D$ is stable and

$$\|\mathcal{G} - \hat{\mathcal{G}}\|_\infty \leq 2 \sum_{i=r+1}^{n} \sigma_i. \tag{6}$$

Balanced truncation thus provides a principled way to reduce model order while controlling the approximation error in terms of discarded Hankel singular values. This makes it a central tool for simplifying state space models while preserving their dominant dynamics.

### 2.2 SPECTRAL STABILITY OF HERMITIAN MATRICES

In practice, training SSMs with gradient descent modifies the learned state matrices incrementally. Understanding how the downstream Hankel singular values shift under such perturbations is therefore critical to establish in-training reduction protocols. For Hermitian matrices, Weyl's theorem provides a powerful tool.

Let $W$ and $W'$ be Hermitian matrices of size $n$ (*i.e.* symmetric for real-valued matrices), and let $\delta W = W' - W$. Also, $\forall i \in [1, \ldots, n]$, let $\sigma_i(W)$ represent the $i$-th largest eigenvalue of $W$. The ordering $\sigma_1(W) \geq \cdots \geq \sigma_n(W)$ can always be established as the eigenvalues of Hermitian matrices are guaranteed to be real.

**Theorem 2.7** (Weyl (1912)). *$\forall i \in [1, \ldots, n]$, $\sigma_i(W)$ is Lipschitz-continuous on the space of Hermitian matrices with operator norm:*

$$|\sigma_i(W') - \sigma_i(W)| \leq \max_{i=1,\ldots,n} (|\sigma_i(\delta W)|) = \max(|\sigma_1(\delta W)|, |\sigma_n(\delta W)|). \tag{7}$$

In other words, each of the eigenvalues of $W$ can at most fluctuate by the largest absolute eigenvalue of the perturbation $\delta W$, providing a bound on spectral variations under Hermitian perturbations. This bound will turn out to be crucial for our in-training reduction scheme (see Section 3.3).

## 3 THE PROPOSED IN-TRAINING REDUCTION SCHEME

Our general pipeline (illustrated in Figure 1) is designed to be applicable to all types of SSMs as it surgically acts on the dynamical systems within SSM layers of models regardless of the choice of projections, non-linear activations, convolutions, skip connections, etc.

To achieve significant gains in training time, we aim to reduce the model's hidden state dimension where possible early on in training. Typically, we attempt to reduce SSM state dimensions at snapshots of the model obtained at fixed intervals at early stages of training (*e.g.* during learning rate warm-up). Section 3.3 provides justification for the validity of this approach.

## 3.1 COMPRESSM: THE ALGORITHM

At a given training step, we proceed per block. For an input feature vector sequence $\boldsymbol{x} \in \mathbb{R}^{H \times L}$, where $H$ is the inner dimension and $L$ the sequence length, common SSM blocks contain either a single Multi-Input Multi-Output (MIMO) system that transforms the sequence $\boldsymbol{x} \in \mathbb{R}^{H \times L}$ to $\boldsymbol{y} \in \mathbb{R}^{H \times L}$, or $H$ independent per-channel Single-Input Single-Output (SISO) systems that transform $\boldsymbol{x}^i \in \mathbb{R}^{1 \times L}$ to $\boldsymbol{y}^i \in \mathbb{R}^{1 \times L}$, for $i \in \{1, \ldots, H\}$. In the latter case, we proceed per channel.

For a given block (and possibly a given channel index), the reduction algorithm works as follows:

1. Extract the discrete linear system matrices $\boldsymbol{A}, \boldsymbol{B}, \boldsymbol{C}$ from the model weights. We denote by $n$ the current order (rank) of the system.

2. Solve Equation 2 and Equation 3 (using Equation 14 if $\boldsymbol{A}$ is diagonal, as is the case for many modern SSMs) to obtain the Gramians $\boldsymbol{P}$ and $\boldsymbol{Q}$ respectively.

3. Compute the Hankel singular values $\boldsymbol{\sigma}$ via Equation 4.

4. Find the smallest rank $r$ such that the top-$r$ singular values retain a predetermined threshold $(1 - \tau) \in [0, 1]$ of the total energy,

$$r = \min_k \Big\{ \sum_{i=1}^{k} \sigma_i \ \geq \ (1 - \tau) \sum_{i=1}^{n} \sigma_i \Big\}, \tag{8}$$

5. If the rank is smaller than a given fraction of the initial state dimension (i.e. the reduction is large enough to warrant the trouble), compute the balancing transformation matrix $\boldsymbol{T}$ from Theorem 2.6. Otherwise, leave the system unchanged.

6. Transform the original system to its diagonal balanced realization,

$$(\boldsymbol{A}_b, \boldsymbol{B}_b, \boldsymbol{C}_b) = (\boldsymbol{T}^{-1} \boldsymbol{A} \boldsymbol{T}, \boldsymbol{T}^{-1} \boldsymbol{B}, \boldsymbol{C} \boldsymbol{T}) \tag{9}$$

7. Truncate the balanced system down to rank $r$ (with a slight abuse of tensor slicing notation),

$$(\boldsymbol{A}_r, \boldsymbol{B}_r, \boldsymbol{C}_r) = (\boldsymbol{A}_b[: r, : r], \boldsymbol{B}_b[: r, :], \boldsymbol{C}_b[:, : r]) \tag{10}$$

8. Replace the model weights for the dynamical system matrices. Based on the architecture this might require diagonalizing the truncated system to ensure computational consistency.

$$(\boldsymbol{A}, \boldsymbol{B}, \boldsymbol{C}) \leftarrow (\boldsymbol{A}_r, \boldsymbol{B}_r, \boldsymbol{C}_r) \tag{11}$$

Note that the above algorithm can not deliver improved small models if there is no clear correlation between state dimension and model performance. The fact that COMPRESSM applies an *in-training* compression scheme enables a significant increase in performance per training time. Indeed, we validate the speedup experimentally in Section 4.2, and provide an in-depth complexity and time reduction analysis in Section D of the Appendix.

## 3.2 PRAGMATIC VARIANT

In ablations presented in Appendix C, we observe that training can sustain successive moderate balanced truncations without divergence from the large-model upper-bound performance, until impactful HSVs are removed. This observation motivates a slight tweak to the COMPRESSM algorithm. Although one does not have access to the upper bound trained at the large dimension throughout, we can track the validation metric during training and assume that, as long as no drop in performance occurs, the reduction does not cause any noticeable global performance drop.

In practice, the procedure is implemented with a simple safeguard. At each *fixed-fraction* reduction step (10% per reduction is reasonable), we first save the current (pre-reduction) checkpoint. We then apply truncation, train the model for a very small number of steps, and evaluate it on the validation

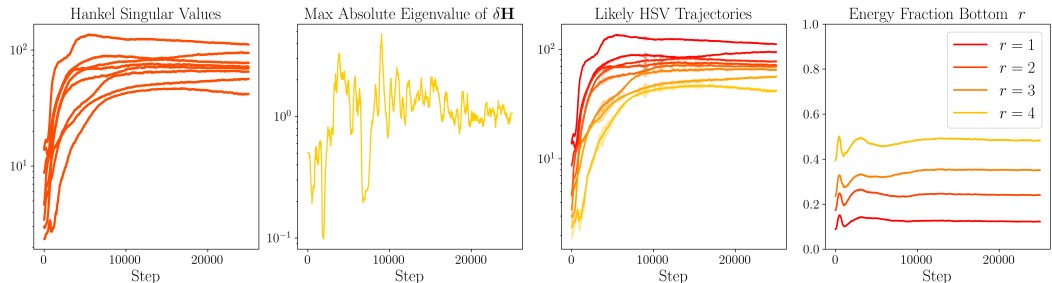

Figure 2: In-training per-step analysis of Hankel singular value dynamics for a single LRU block with state dimension of 8 on the sMNIST dataset for the first 25k steps. The leftmost plot shows the raw HSVs (as a set). The middle-left plot depicts the maximum absolute eigenvalue of $\delta\boldsymbol{H}$ as described in Section 3.3. The middle-right plot overlays the maximum variation bound as an error margin around each HSV, with each shade now representing a highly probable path for a specific state dimension obtained by step by step linear sum assignment solving. The rightmost plot shows the relative contribution of the bottom $r$ HSVs to the total energy.

set. If validation performance continues to improve, we proceed to the next reduction. If performance degrades, we discard the reduced version and revert to the previously saved checkpoint, after which no further reductions are applied. Results with this approach are provided in Section B.2.

This protocol ensures that model quality always remains close to that of the unreduced baseline, without the need to explicitly tune the number of reductions or a fixed tolerance level.

### 3.3  IN-TRAINING REDUCTION

The validity of our proposed in-training truncation relies on several non-trivial properties, which we first motivate intuitively before formalizing. First, the method requires tracking how the relative importance of individual states evolves with training. Second, even if we can measure importance continuously, training dynamics may render initially insignificant dimensions crucial at later stages. Early truncation of such dimensions would therefore be undesirable. Consequently, it is necessary that the relative ordering of importance remains approximately stable, at least for the dimensions with low initial contribution. Finally, since our balanced truncation approach relies on the energy contribution of each dimension relative to the total system energy, it is desirable that the cumulative importance of the bottom-$r$ dimensions does not increase substantially during training. Otherwise, dimensions could converge to a regime of near-equal importance, making early truncation unjustified even if the ordering is preserved.

To leverage Hankel singular value analysis during training, we must first establish a protocol for tracking individual state importance as the system evolves under gradient updates. This existence of such a protocol is non-trivial and its development is central to this work.

Indeed, between gradient steps, model weights are updated according to the negative gradient of the loss with respect to the parameters. At the SSM level, this translates into the state matrices being incrementally updated such that a discrete system described by $(\boldsymbol{A}, \boldsymbol{B}, \boldsymbol{C})$ becomes a *different* dynamical system $(\boldsymbol{A}', \boldsymbol{B}', \boldsymbol{C}')$, where

$$\boldsymbol{A}' = \boldsymbol{A} + \delta\boldsymbol{A}, \quad \boldsymbol{B}' = \boldsymbol{B} + \delta\boldsymbol{B}, \quad \boldsymbol{C}' = \boldsymbol{C} + \delta\boldsymbol{C}. \tag{12}$$

The omission of $\boldsymbol{D}$ is deliberate for both notational simplicity, but also since it is often fixed as a skip layer and not learned. Throughout this section, we adopt the convention that plain symbols denote pre-update quantities, while primed symbols denote their post-update counterparts.

Now, using the expressions of the controllability and observability Gramians given respectively by Equation 2 and Equation 3, one can see that both Gramians are continuous with respect to the gradient perturbation to the system matrices thus we can also consider,

$$\boldsymbol{P}' = \boldsymbol{P} + \delta\boldsymbol{P}, \quad \boldsymbol{Q}' = \boldsymbol{Q} + \delta\boldsymbol{Q}, \tag{13}$$

where $\delta\boldsymbol{P}$ is some continuous function of $(\delta\boldsymbol{A}, \delta\boldsymbol{B})$, and similarly $\delta\boldsymbol{Q}$ of $(\delta\boldsymbol{A}, \delta\boldsymbol{C})$.

Also recall that the Hankel singular values can be obtained as the square root of the eigenvalues of $PQ$ (Equation 4). Generally, $PQ$ need not be symmetric, but we use this form for computational efficiency. Noticing that $PQ$ is similar to the symmetric positive definite matrix $P^{\frac{1}{2}}QP^{\frac{1}{2}}$, we let $H = (P^{\frac{1}{2}}QP^{\frac{1}{2}})^{\frac{1}{2}}$. The matrix $H$'s eigenvalues are exactly the Hankel singular values. In addition, by composition, $H$ is continuous with respect to the perturbations to the system, so we write $H' = H + \delta H$ with $\delta H$ a continuous function of $(\delta A, \delta B, \delta C)$.

**Lemma 3.1** (Continuity of Hankel singular values under training updates)*. By application of Weyl's Theorem 2.7 to the matrix $H$ and its perturbation $H'$, between gradient steps, each Hankel singular value can at most change by the largest absolute eigenvalue of $\delta H = H' - H$.*

While the previous argument establishes that Hankel singular values evolve continuously with gradient updates, the remaining conditions—stability of relative ordering and low contribution of the bottom-$r$ dimensions—cannot be guaranteed theoretically. However, empirical evidence strongly suggests that standard training dynamics are favorable in practice.

We examine the case of a single LRU block trained on the sMNIST dataset with state dimension of 8 for visual clarity in Figure 2 (and provide plots for larger models and datasets in Appendix B.3). To keep track of the state dimension to HSV value correspondence, we overlay the maximum absolute eigenvalue of $\delta H$, on top of each HSV. Empirically, this allows us to robustly identify probable HSV trajectories as the continuity bound is consistently small enough to ensure each eigenvalue is clearly isolated, with minimal bound overlaps and rare gradual HSV relative order crossings occurring (prediction of evolution established via linear sum assignment solution in such cases). Furthermore, the cumulative contribution of the bottom-$r$ HSVs stabilizes rapidly. Indeed, after a small initial number of steps, we observe that both the ordering of singular values remains constant, and dimensions of low importance seldom gain substantial relative energy during training.

In sum, these observations provide empirical justification for early in-training truncation: dimensions identified as negligible at early stages typically remain so throughout training. Consequently, truncation decisions made during training rarely conflict with the final importance ranking, making our approach both effective and robust in practice.

## 4 EXPERIMENTS

### 4.1 EXPERIMENTAL SETUP

In order to empirically validate in-training balanced truncation, we train a linear recurrent unit (LRU) (Orvieto et al., 2023) on datasets of different complexity, ranging from sMNIST to tasks from the long range arena (LRA) (Tay et al., 2020).

We use the same training pipeline as Rusch & Rus (2025); Walker et al. (2025), accounting for additional quirks of LRU training like a learning rate factor for the sequence mixing layer. The LRA dataloaders are borrowed from Smith et al. (2022), the hyperparameters largely taken from (Orvieto et al., 2023, Table 10). We summarize them in Table 3.

Reduction starts from the full model order reported in column 4 of Table 3. For all datasets but IMDB and sMNIST, we attempt four equidistant reduction steps during the warm-up period of the learning rate, which equals $10\%$ of the total steps. Doing so ensures maximal speed up potential for the subsequent $90\%$ of training, while also staying robust to large early training perturbations.

As sMNIST is trained without learning rate decay, we attempt truncation during all of training. For IMDB, we include a waiting stage and attempt reduction inside a smaller time window. The details can be found in Appendix A.3. Generally, reductions are only executed if the reduced dimension is less than $95\%$ of the current state dimension.

The baselines we compare against are non-reduced models trained all along with each block initialized at the average final order of the compressed ones to allow for a fair comparison. To increase our baseline statistics and further establish correlations between state dimension and model performance, we train further models with different state dimensions.

Table 1: Average final state dimension (mean $\pm$ std) and Top-3 runs mean performance with/without reduction for LRU under different tolerances $\tau$.

| Dataset | Metric | $\tau = 1.5 \cdot 10^{-1}$ | $\tau = 1 \cdot 10^{-1}$ | $\tau = 7 \cdot 10^{-2}$ | $\tau = 5 \cdot 10^{-2}$ | $\tau = 3 \cdot 10^{-2}$ | $\tau = 2 \cdot 10^{-2}$ | $\tau = 0$ |
|---|---|---|---|---|---|---|---|---|
| CIFAR10 | State dim | $57.4 \pm 1.5$ | $92.6 \pm 4.2$ | $126.0 \pm 4.0$ | $160.8 \pm 5.4$ | $213.6 \pm 6.1$ | $327.2 \pm 16.0$ | 384 |
| | CompreSSM | $\mathbf{84.4 \pm 0.2}$ | $\mathbf{85.7 \pm 0.1}$ | $\mathbf{86.0 \pm 0.1}$ | $85.8 \pm 0.1$ | $86.0 \pm 0.2$ | $\mathbf{86.1 \pm 0.2}$ | - |
| | Baseline | $78.2 \pm 0.7$ | $81.8 \pm 0.3$ | $83.7 \pm 0.2$ | $84.2 \pm 0.5$ | $84.9 \pm 0.0$ | $86.0 \pm 0.1$ | $86.5 \pm 0.3$ |
| ListOps | State dim | $56.8 \pm 3.4$ | $81.8 \pm 4.9$ | $109.8 \pm 3.9$ | $135.4 \pm 6.8$ | $167.6 \pm 5.7$ | $213.8 \pm 28.0$ | 256 |
| | CompreSSM | $\mathbf{48.3 \pm 0.7}$ | $\mathbf{51.8 \pm 0.9}$ | $48.2 \pm 1.1$ | $47.5 \pm 1.6$ | $\mathbf{49.2 \pm 0.3}$ | $47.1 \pm 1.4$ | - |
| | Baseline | $43.4 \pm 0.4$ | $46.3 \pm 0.5$ | $\mathbf{49.4 \pm 1.8}$ | $49.2 \pm 0.7$ | $48.2 \pm 2.1$ | $\mathbf{47.6 \pm 1.7}$ | $49.7 \pm 0.8$ |
| AAN | State dim | $53.6 \pm 1.9$ | $84.4 \pm 1.4$ | $111.0 \pm 2.0$ | $136.6 \pm 2.9$ | $170.0 \pm 2.4$ | $203.2 \pm 13.7$ | 256 |
| | CompreSSM | $87.2 \pm 0.3$ | $87.5 \pm 0.1$ | $87.4 \pm 0.3$ | $87.2 \pm 0.0$ | $87.6 \pm 0.3$ | $\mathbf{87.9 \pm 0.2}$ | - |
| | Baseline | $\mathbf{87.5 \pm 0.3}$ | $\mathbf{87.9 \pm 0.3}$ | $\mathbf{87.8 \pm 0.2}$ | $\mathbf{87.8 \pm 0.5}$ | $87.3 \pm 0.4$ | $87.4 \pm 0.5$ | $\mathbf{87.3 \pm 0.3}$ |
| IMDB | State dim | $95.0 \pm 2.3$ | $119.6 \pm 2.2$ | $136.8 \pm 1.9$ | $150.4 \pm 1.2$ | $165.0 \pm 1.3$ | $192.0 \pm 0.0$ | 192.0 |
| | CompreSSM | $82.2 \pm 0.2$ | $82.8 \pm 0.1$ | $83.7 \pm 0.4$ | $83.8 \pm 0.3$ | $84.1 \pm 0.4$ | $84.4 \pm 0.2$ | - |
| | Baseline | $\mathbf{82.7 \pm 0.1}$ | $\mathbf{83.5 \pm 0.1}$ | $\mathbf{83.7 \pm 0.0}$ | $\mathbf{84.0 \pm 0.4}$ | $\mathbf{84.3 \pm 0.0}$ | $\mathbf{84.5 \pm 0.1}$ | $84.7 \pm 0.1$ |
| Pathfinder | State dim | $34.6 \pm 1.9$ | $51.2 \pm 1.7$ | $65.6 \pm 2.3$ | $81.2 \pm 1.6$ | $105.0 \pm 2.1$ | $129.8 \pm 5.2$ | 256 |
| | CompreSSM | $96.6 \pm 1.3$ | $\mathbf{97.9 \pm 0.1}$ | $97.6 \pm 0.5$ | $97.8 \pm 0.4$ | $98.0 \pm 0.1$ | $98.0 \pm 0.1$ | - |
| | Baseline | $\mathbf{97.3 \pm 0.2}$ | $\mathbf{97.9 \pm 0.1}$ | $\mathbf{98.0 \pm 0.1}$ | $\mathbf{98.1 \pm 0.0}$ | $\mathbf{98.2 \pm 0.0}$ | $\mathbf{98.1 \pm 0.1}$ | $98.3 \pm 0.1$ |

| Dataset | Metric | $\tau = 4 \cdot 10^{-2}$ | $\tau = 2 \cdot 10^{-2}$ | $\tau = 1 \cdot 10^{-2}$ | $\tau = 5 \cdot 10^{-3}$ | $\tau = 2 \cdot 10^{-3}$ | $\tau = 1 \cdot 10^{-3}$ | $\tau = 0$ |
|---|---|---|---|---|---|---|---|---|
| sMNIST | State dim | $12.7 \pm 3.0$ | $27.6 \pm 1.8$ | $46.8 \pm 3.2$ | $76.3 \pm 7.5$ | $148.1 \pm 9.8$ | $191.4 \pm 4.7$ | 256 |
| | CompreSSM | $\mathbf{95.9 \pm 0.2}$ | $\mathbf{96.9 \pm 0.0}$ | $\mathbf{96.9 \pm 0.1}$ | $\mathbf{96.9 \pm 0.1}$ | $97.0 \pm 0.1$ | $\mathbf{97.2 \pm 0.3}$ | - |
| | Baseline | $92.6 \pm 0.5$ | $96.0 \pm 0.2$ | $95.9 \pm 0.1$ | $96.4 \pm 0.2$ | $\mathbf{97.3 \pm 0.2}$ | $\mathbf{97.3 \pm 0.1}$ | $97.3 \pm 0.1$ |

## 4.2 EMPIRICAL RESULTS

We repeat all experiments using five different random seeds and report mean top-3 as well as top-1 performance. Table 1 contains the top-3 results. The top-1 results can be found in the appendix (Table 4). The state dimensions reported for multi-block models are the averages of the SSM orders per-block. It is accompanied by Figure 4, which presents the performance visually and contains further non-reduced benchmark models.

For some datasets, for example AAN or Pathfinder, our baseline experiments reveal a small correlation between state dimension and model performance, given the other model parameters taken from Orvieto et al. (2023) (see entries 3 and 5 in Table 1 or figures 4c and 4e). On these datasets, CompreSSM can not deliver better performance for small models.

However, on datasets where state dimension does correlate with model performance, CompreSSM improves small model performance. On CIFAR10, for example, model performance almost stays constant as a function of reduction tolerance (and thus for different state dimensions), while the non-compressed counterparts exhibit an approximately 10% performance drop (see entry 1 in Table 1 or Figure 3a). The sMNIST results paint a similar picture (entry 5 in Table 1 or Figure 4a).

Similarly, on ListOps, the performance of non-compressed models drops significantly for state dimensions smaller than 120 (see Figure 4f). While the compressed models perform on par with their uncompressed counterparts for larger state dimensions, smaller models outperform the baseline.

On Pathfinder, we can observe a similar trend (Figure 4e), where the unreduced model performance does not show a strong correlation with the state dimension. Just for the smallest state dimension, the unreduced model sees a small drop in performance and is outperformed by the top-1 reduced model.

The IMDB results reveal the importance of the prerequisites mentioned in Section 3.1 (see Figure 4d). Even after significantly reducing the parameters and increasing the droprate (see Appendix A), the unreduced models only learn for roughly $8k$ steps before overfitting. However, in order to do in-training balanced truncation, the actual training phase needs to be long enough to allow for a couple of reduction steps with a large enough spacing so that the model can follow the training dynamics for a while without being pruned. Indeed, for non-aggressive pruning (that is, pruning with small tolerance $\tau$), the top reduced models often outperform the baseline.

As mentioned previously, an advantage of CompreSSM is that it comes with a training speedup; as the state dimensions get pruned, training speeds up. Indeed, Figure 3b demonstrates this effect on CIFAR10. CompreSSM at dimension 92 preserves nearly full baseline accuracy (85.7% vs. 86.5%) and achieves a $1.5\times$ speedup. By comparison, training directly at dimension 92 is only

marginally faster ($1.6\times$ speedup) yet markedly worse, reaching just $81.8\%$ accuracy. A detailed breakdown of in-training speedup gains is provided in Appendix D.

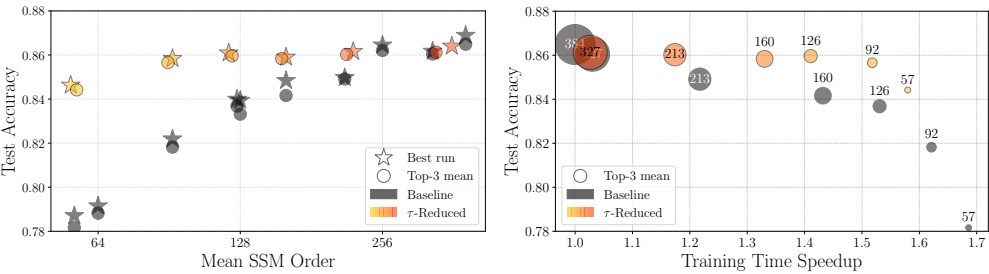

(a) Test accuracy vs. Final state dimension      (b) Test accuracy vs. Training speedup factor

Figure 3: Subfigure (a) shows the performance of different models trained on CIFAR10 as a function of the state dimension. Grey data indicates non-reduced models, and the shades of orange correspond to reduced models, with tolerance decreasing with redness. The circles represents the top-3 mean, while the star corresponds to the top-1 model. Subfigure (b) shows top-3 performance as a function of training speedup with respect to the original large model training without reductions ($n = 384$ baseline). Marker diameter is proportional to the final model order (also annotated) and in-between models are omitted for visual decluttering.

## 4.3 COMPARISON TO OTHER MODEL ORDER REDUCTION TECHNIQUES

We compare COMPRESSM to other model order reduction approaches, namely reduction with Hankel Nuclear Norm (HNN) regularization (Forgione et al., 2024; Schwerdtner et al., 2025), as well as Knowledge Distillation (KD) (Hinton et al., 2015). The approaches are described in Appendix F, with implementation details provided as well. Results of the experiments are summarized in Table 2.

The former experiment reveals three clear findings: (i) HNN regularization is computationally expensive, slowing training by an order of magnitude at least due to repeated Gramian-eigenvalue evaluations (at every single gradient step); (ii) HNN-constrained models underperform unconstrained baselines even before reduction; and (iii) when targeting small final dimensions, COMPRESSM achieves higher accuracy while being over dramatically faster.

On the other hand, knowledge distillation performs on par with COMPRESSM as long as the teacher has a similar state dimension as the student. However, as the model is further compressed, KD's performance drops significantly, while COMPRESSM is able to maintain performance. KD furthermore requires training the large teacher model to completion before training a smaller student model, as well as doing a forward pass through the teacher even when training the student. This can be substantially slower than COMPRESSM.

## 5 RELATED WORK

**Model compression techniques in machine learning.** The question of model compression has received considerable attention in the literature (Deng et al., 2020; Zhu et al., 2024), leveraging various techniques (and mixtures thereof) such as pruning (Han et al., 2015; Frantar & Alistarh, 2023), which removes redundant parameters to reduce network size; quantization (Xiao et al., 2022; Gholami et al., 2021), which compresses models by lowering numerical precision; low-rank factorization (Lin et al., 2024), which exploits structure in weight matrices to reduce dimensionality; and knowledge distillation (Hinton et al., 2015; Gou et al., 2020), where a smaller model learns to mimic a larger one.

**In-training vs. post-training paradigms.** Techniques such as quantization-aware training (Choi et al., 2018; Zhang et al., 2018), and dynamic pruning (Hoefler et al., 2021; Wimmer et al., 2022) perform in-training compression during the optimization process. In contrast, most methods like

Table 2: MOR techniques on sMNIST and CIFAR10. Top-3 mean test accuracies (%) and training speedup factors are reported. HNN Regularization is evaluated on sMNIST, and Knowledge Distillation is evaluated on CIFAR10. $n$ represents models' final order (rounded).

| | Method | Metric | n = 13 | n = 28 | n = 47 | n = 76 | n = 148 | n = 191 | n = 256 |
|---|---|---|---|---|---|---|---|---|---|
| **sMNIST** | Baseline | Acc. (%) | 92.6 | 96.0 | 95.9 | 96.4 | **97.3** | **97.3** | **97.3** |
| | | Speed × | 3.1 | 2.8 | 2.7 | 2.4 | 2.0 | 1.7 | 1.0 |
| | COMPRESSM | Acc. (%) | **95.9** | **96.9** | **96.9** | **96.9** | 97.0 | 97.2 | – |
| | | Speed × | 2.8 | 2.6 | 2.5 | 2.3 | 1.9 | 1.6 | – |
| | HNN Reg. | Acc. (%) | 91.7 | 95.8 | 95.8 | 95.8 | 95.8 | 95.8 | 95.9 |
| | | Speed × | 0.06 | 0.06 | 0.06 | 0.06 | 0.06 | 0.06 | 0.06 |
| | **Method** | **Metric** | **n = 57** | **n = 93** | **n = 126** | **n = 161** | **n = 214** | **n = 327** | **n = 384** |
| **CIFAR10** | Baseline | Acc. (%) | 78.2 | 81.8 | 83.7 | 84.2 | 84.9 | 86.0 | 86.5 |
| | | Speed × | 1.69 | 1.62 | 1.53 | 1.43 | 1.22 | 1.03 | 1.00 |
| | COMPRESSM | Acc. (%) | **84.4** | **85.7** | **86.0** | **85.8** | **86.0** | 86.1 | – |
| | | Speed × | 1.58 | 1.52 | 1.41 | 1.33 | 1.17 | 1.03 | – |
| | KD | Acc. (%) | 79.4 | 83.5 | 84.4 | 85.3 | **86.0** | **87.0** | – |
| | | Speed × | 0.55 | 0.52 | 0.61 | 0.51 | 0.49 | 0.45 | – |

Deep Compression (Han et al., 2015) follow a post-training paradigm, applying pruning or quantization after convergence and relying on retraining or fine-tuning to recover accuracy.

**Compression in SSMs.** Work on compressing SSMs has primarily explored quantization. Post-training quantization has been applied to stabilize SSM inference under 8-bit constraints (Abreu et al., 2024; Chiang et al., 2024), while quantization-aware training has been used to maintain accuracy below 8 bits (Zhao et al., 2025) and to improve robustness for deployment on specialized or analog hardware (Siegel et al., 2024; 2025).

By contrast, control-theoretic MOR approaches have been applied to diagonal S4 layers, but only as a post-hoc step to initialize retraining (Ezoe & Sato, 2024). Elsewhere, regularizers based on the Hankel nuclear norm or modal $\ell_1$ have been shown to encourage parsimonious state representations during training, though without explicit intermediate truncation events, and with a price to pay in terms of both per step compute time as well as optimal performance (Forgione et al., 2024; Schwerdtner et al., 2025). Finally, $\mathcal{H}_2$-optimal reductions have been proposed as a competitor to balanced truncation in offline SSM MOR settings (Sakamoto & Sato, 2025). This approach is computationally expensive, requiring gradient on a complex optimization problem, without the solution guaranteeing stability of the reduced order system.

Though not a compression strategy, the authors in Yu et al. (2024) approach the problem of HSV efficiency from the perspective of initialization, aiming to avoid fast decaying singular values.

To the best of our knowledge, our contribution is the first to propose a principled *in-training* model order reduction method applicable to a broad spectrum of SSMs.

## 6 CONCLUSION

We propose COMPRESSM: a framework for principled *in-training* compression of SSMs. Leveraging control theory, we utilize Weyl's theorem to track the evolution of HSVs during learning. Empirically, we observe that the relative contributions of individual state components remain highly stable. This allows us to safely apply balanced truncation early in the training process.

COMPRESSM is put to the test with LRUs on a range of LRA tasks. Provided there is a correlation between state capacity and model performance, we verify empirically that compressed models outperform their uncompressed counterparts while delivering better performance per unit of training time. In short, COMPRESSM enables performance of high capacity models at *both* the inference and training overheads of smaller counterparts.

Finally, we provide an implementation strategy for linear time-varying systems. In the selective case, we propose averaging the dynamics over the input space before applying the reduction scheme. Future work will focus on extending COMPRESSM to linear attention models, e.g., Gated Linear Attention (Yang et al., 2023), Mamba2 (Dao & Gu, 2024), and Gated DeltaNet (Yang et al., 2024).

ACKNOWLEDGMENTS

Philipp Nazari was supported by the Max Planck ETH Center for Learning Systems. This work was supported in part by the Hector Foundation, the Boeing Company, and the ONR Science of Autonomy program N00014-23-1-2354.

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

# A IMPLEMENTATION DETAILS

## A.1 SOLVING THE LYAPUNOV EQUATIONS FOR DIAGONAL SSMs

For SSMs with diagonal state transition matrix $\boldsymbol{A} = \mathrm{diag}(\lambda_1, \ldots, \lambda_n)$, which covers a lot of SSMs used today such as LRU (Orvieto et al., 2023) and S5 (Smith et al., 2022), Equation 2 and Equation 3 admit a simple, entry-wise closed-form solution:

$$\boldsymbol{P}_{ij} = \frac{\left(\boldsymbol{B}\boldsymbol{B}^\top\right)_{ij}}{1 - \lambda_i \lambda_j}, \quad \boldsymbol{Q}_{ij} = \frac{\left(\boldsymbol{C}^\top \boldsymbol{C}\right)_{ij}}{1 - \lambda_i \lambda_j} \quad \forall 1 \le i, j \le n. \tag{14}$$

In the non-diagonal case, one can either solve the Lyapunov equations by vectorization or use the argument put forward by Orvieto et al. (2023) and realize that every state transition matrix $\boldsymbol{A} \in \mathbb{R}^{n,n}$ can be diagonalized over $\mathbb{C}$ up to a small perturbation.

## A.2 HYPERPARAMETER SELECTION

The hyperparameters we used can be found in Table 3. For all but one LRA tasks we use the same ones as reported by (Orvieto et al., 2023, Table 10). The exception is IMDB, which we observed to overfit massively with the given hyperparameters. We mitigate this issue by increasing dropout and reducing the total number of layers.

Table 3: Hyperparameters for LRU experiments. $h$ refers to the dimension of the hidden state, $n$ to the state space dimension, $B$ to the batch size and $\alpha_{LR}$ to the learning rate factor applied to the sequence mixer (Orvieto et al., 2023).

| Task | Depth | $h$ | $n$ | Steps | $B$ | $\alpha_{LR}$ | Weight Decay | Dropout |
|------|-------|-----|-----|-------|-----|---------------|--------------|---------|
| sMNIST | 1 | 8 | 256 | $200k$ | 50 | - | - | 0.1 |
| CIFAR10 | 6 | 512 | 384 | $180k$ | 50 | 0.25 | 0.05 | 0.1 |
| ListOps | 6 | 128 | 256 | $80k$ | 32 | 0.5 | 0.05 | 0.0 |
| IMDB | 1 | 256 | 192 | $50k$ | 32 | 0.1 | 0.05 | 0.1 |
| AAN | 6 | 128 | 256 | $100k$ | 64 | 0.5 | 0.05 | 0.1 |
| Pathfinder | 6 | 192 | 256 | $500k$ | 64 | 0.25 | 0.05 | 0.0 |

On LRA tasks, the learning rate is warmed up from $10^{-7}$ to $10^{-3}$ for 10% of the total steps, before it is cosine-decayed back to $10^{-7}$. For sMNIST, the learning rate is fixed at $4 \cdot 10^{-4}$ for the entirety of training.

## A.3 REDUCTION DETAILS

We find that LRU overfits on IMDB, even after reducing the number of parameters by 6 and doubling the dropout rate compared to Orvieto et al. (2023). Training LRU on this dataset, we furthermore observe an initial training period in which the loss plateaus. Just on IMDB, we thus wait for an initial $1k$ steps before doing the balanced truncation. Instead of doing 4 reduction steps until the end of warmup, we also just do 2 until $3k$ steps in order to avoid entering the overfitting regime.

# B ADDITIONAL NUMERICAL RESULTS

This appendix expands upon the empirical findings presented in the main paper by providing a more complete set of numerical results and supporting analyses. We aim to complete the view of how COMPRESSM behaves, and to substantiate the assumptions underlying our in-training reduction framework. We first report extended performance curves and dimension statistics across all benchmarks, results for the pragmatic variant of the algorithm, followed by additional evaluations of the dynamical behavior of Hankel singular values throughout training. Together, these results offer a holistic understanding of the stability, robustness, and practical effectiveness of the proposed method.

## B.1 PERFORMANCE

In Figure 4 we provide the state dimension vs test performance plots for all datasets. These follow the same conventions and experimental setup as those described in the main text for Subfigure (a). The baselines again correspond to models that are trained from scratch at the with the mean of the reduced dimensions reached at the various tolerance levels. Time gain plots are not provided, but similar gains are observed across all experiments, with an in depth discussion and analysis on the subject provided in Section D of this appendix.

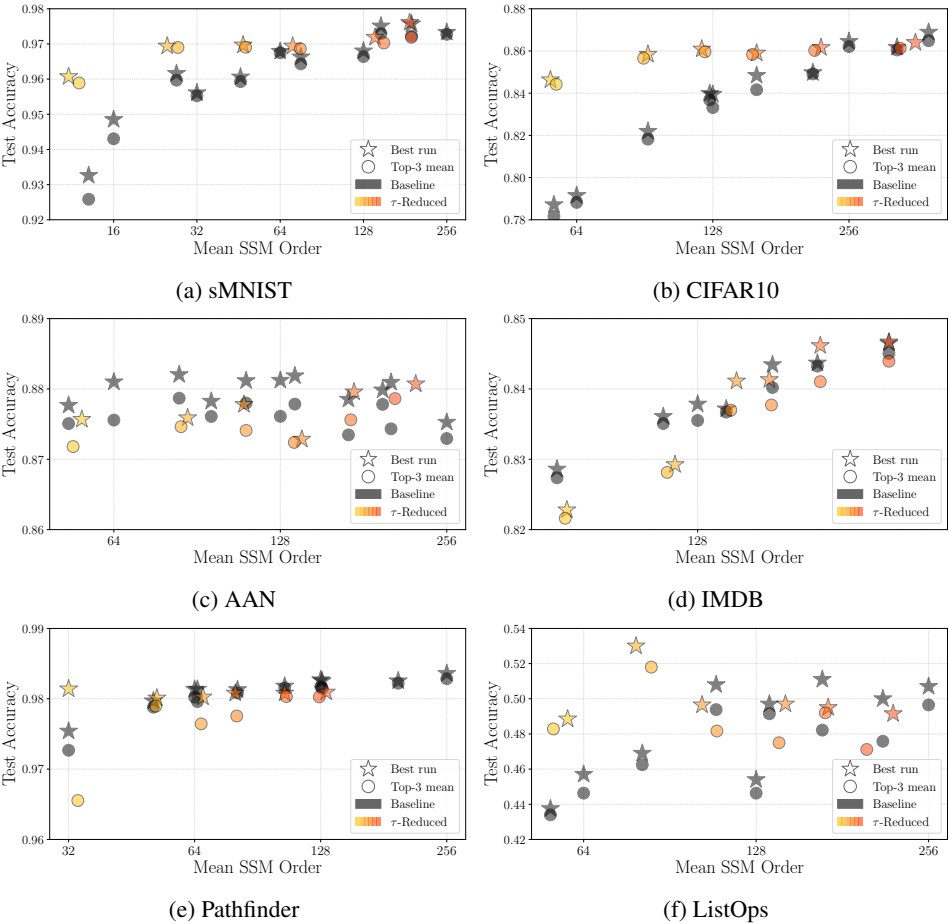

Figure 4: Test performance vs. final state dimension for all our experiments. Stars correspond to best performance, circles to the mean of the top-3 runs. Grey shapes correspond to non-reduced models, and the shades of orange to reduced models, with tolerance decreasing with redness.

Table 4: Final state dimension (mean ± std) and MAX performance with/without reduction for LRU under different tolerances $\tau$.

| Dataset | Metric | $\tau = 1.5 \cdot 10^{-1}$ | $\tau = 1 \cdot 10^{-1}$ | $\tau = 7 \cdot 10^{-2}$ | $\tau = 5 \cdot 10^{-2}$ | $\tau = 3 \cdot 10^{-2}$ | $\tau = 2 \cdot 10^{-2}$ | $\tau = 0$ |
|---|---|---|---|---|---|---|---|---|
| CIFAR10 | State dim | $57.4 \pm 1.5$ | $92.6 \pm 4.2$ | $126.0 \pm 4.0$ | $160.8 \pm 5.4$ | $213.6 \pm 6.1$ | $327.2 \pm 16.0$ | 384 |
|  | COMPRESSM | **84.6** | **85.8** | **86.1** | **85.9** | **86.2** | **86.4** | - |
|  | Baseline | 78.7 | 82.2 | 84.0 | 84.8 | 85.0 | 86.2 | 86.9 |
| ListOps | State dim | $56.8 \pm 3.4$ | $81.8 \pm 4.9$ | $109.8 \pm 3.9$ | $135.4 \pm 6.8$ | $167.6 \pm 5.7$ | $213.8 \pm 28.0$ | $256.0 \pm 0.0$ |
|  | COMPRESSM | **48.9** | **53.0** | 49.7 | **49.7** | 49.5 | 49.2 | - |
|  | Baseline | 43.8 | 46.9 | **50.8** | 49.7 | **51.1** | **50.0** | 50.7 |
| AAN | State dim | $53.6 \pm 1.9$ | $84.4 \pm 1.4$ | $111.0 \pm 2.0$ | $136.6 \pm 2.9$ | $170.0 \pm 2.4$ | $203.2 \pm 13.7$ | 256 |
|  | COMPRESSM | 87.6 | 87.6 | 87.8 | 87.3 | **88.0** | **88.1** | - |
|  | Baseline | **87.8** | **88.2** | **88.1** | **88.2** | 87.9 | **88.1** | 87.5 |
| IMDB | State dim | $95.0 \pm 2.3$ | $119.6 \pm 2.2$ | $136.8 \pm 1.9$ | $150.4 \pm 1.2$ | $165.0 \pm 1.3$ | $192.0 \pm 0.0$ | 192 |
|  | COMPRESSM | 82.3 | 82.9 | **84.1** | 84.1 | **84.6** | **84.7** | - |
|  | Baseline | **82.9** | **83.6** | 83.7 | **84.3** | 84.4 | **84.7** | 84.7 |
| Pathfinder | State dim | $34.6 \pm 1.9$ | $51.2 \pm 1.7$ | $65.6 \pm 2.3$ | $81.2 \pm 1.6$ | $105.0 \pm 2.1$ | $129.8 \pm 5.2$ | 256 |
|  | COMPRESSM | **98.1** | **98.0** | 98.0 | **98.1** | 98.1 | 98.1 | - |
|  | Baseline | 97.5 | **98.0** | **98.1** | **98.1** | **98.2** | **98.3** | 98.4 |

| Dataset | Metric | $\tau = 3 \cdot 10^{-2}$ | $\tau = 2 \cdot 10^{-2}$ | $\tau = 1 \cdot 10^{-2}$ | $\tau = 5 \cdot 10^{-3}$ | $\tau = 2 \cdot 10^{-3}$ | $\tau = 1 \cdot 10^{-3}$ | $\tau = 0$ |
|---|---|---|---|---|---|---|---|---|
| sMNIST | Dim (± std) | $12.7 \pm 3.0$ | $27.6 \pm 1.8$ | $46.8 \pm 3.2$ | $76.3 \pm 7.5$ | $148.1 \pm 9.8$ | $191.4 \pm 4.7$ | 256 |
|  | COMPRESSM | **96.1** | **96.9** | **97.0** | **96.9** | 97.2 | **97.6** | - |
|  | Baseline | 93.3 | 96.2 | 96.1 | 96.6 | **97.5** | **97.6** | 97.3 |

## B.2 PRAGMATIC APPROACH PERFORMANCE

We set out to simplify the tuning of hyperparameters linked to the COMPRESSM protocol. We avoid having to decide on the non-intuitive energy threshold and the number of reductions to be scheduled *a priori*. Instead, one selects: (i) the fraction of states to truncate at each reduction step, and (ii) an acceptable performance drop tolerance. We fix these to reasonable values: truncating a constant 10% of the current state dimension per reduction, with a tolerance level of 1% of the pre-reduction accuracy after training for an additional 1% of the total steps. Results for the sMNIST and CIFAR10 benchmarks are provided in Table 5.

Table 5: Pragmatic COMPRESSM results with fixed 10% reduction fraction and 1% performance tolerance. Final state dimension (mean ± std across all runs) and Top-3 mean test accuracy reported, consistent with Table 1.

| Dataset | Metric | Pragmatic | COMPRESSM | Baseline |
|---|---|---|---|---|
| sMNIST | State dim | $121.3 \pm 89.8$ | $191.4 \pm 4.7$ | 256 |
|  | Top-3 Acc. (%) | $96.9 \pm 0.2$ | $\mathbf{97.2 \pm 0.3}$ | $97.3 \pm 0.1$ |
| CIFAR10 | State dim | $307.0 \pm 43.9$ | $327.2 \pm 16.0$ | 384 |
|  | Top-3 Acc. (%) | $86.2 \pm 0.3$ | $86.1 \pm 0.2$ | $\mathbf{86.5 \pm 0.3}$ |

The pragmatic variant achieves competitive performance: $96.9\%$ on sMNIST (vs. $97.2\%$ for tolerance-based COMPRESSM at $\tau = 10^{-3}$) and $86.2\%$ on CIFAR10 (vs. $86.1\%$ at $\tau = 2 \cdot 10^{-2}$). However, the key trade-off lies in the **predictability of compression**. While tolerance-based COMPRESSM yields consistent final dimensions (e.g., $191.4 \pm 4.7$ on sMNIST), the pragmatic approach exhibits high variance ($121.3 \pm 89.8$), with final dimensions ranging from 15 to 230 across seeds. This variability arises because the acceptance criterion depends on the validation metric trajectory, which is sensitive to initialization and optimization dynamics.

Notably, some seeds achieve extreme compression (e.g., 15 states with $96.8\%$ accuracy on sMNIST), while others reject reductions early and remain close to the full model size. This behavior suggests that the pragmatic approach effectively adapts to the "intrinsic compressibility" of each training run, though at the cost of less predictable outcomes. For applications where consistent model sizes are required, the tolerance-based approach remains preferable; for exploratory compression where maximal reduction is desired, the pragmatic variant offers a simpler, hyperparameter-light alternative.

## B.3 EMPIRICAL IN-TRAINING HANKEL STABILITY

A central assumption underlying our reduction procedure is that the Hankel singular values (HSVs) of a learned SSM evolve smoothly during training and, after an initial transient, maintain a stable relative ordering. This stability is what enables us to reliably identify and truncate low-energy modes without waiting for the model to fully converge. In this section, we provide additional empirical evidence supporting this assumption across a wide range of datasets, model sizes, and architectural configurations.

Across all experiments—spanning sMNIST, IMDB, CIFAR10, and ListOps—we consistently observe the same qualitative behavior. Immediately after initialization, the HSV spectrum typically undergoes a brief reshaping phase in which the dominant modes separate from the rest. After this point (usually within the first few thousand training steps), the spectrum stabilizes and the relative ordering of HSVs becomes highly consistent. Larger modes drift slowly but retain their rank order, while smaller modes flatten and remain several orders of magnitude below the truncation threshold. This separation persists throughout training, even for deeper models and larger state dimensions.

These observations directly support the assumptions used in our analysis in the main text. First, they validate the claim that the HSV spectrum is well-behaved and exhibits only mild temporal variability once early training transients dissipate. Second, they confirm that the lower-energy portion of the spectrum remains largely inactive and can be removed with negligible impact on validation performance. Finally, they highlight that the qualitative structure of the HSVs is robust across tasks and model scales, lending practical reliability to in-training reductions.

Figures 21, 22, 23, and 24 illustrate these trends for representative runs across datasets. In each plot, we visualize the evolution of HSVs over the course of training, sampled at intervals on the order of thousands of steps. Because these intervals are much larger than the per-step analysis used in the main text, the exact perturbation continuity bounds become too noisy to compute meaningfully; nevertheless, we apply linear-sum-assignment tracking to produce a consistent alignment of HSV indices over time. In all cases, the dominant HSVs quickly settle into stable trajectories, while smaller HSVs decay toward near-zero values and form a clear truncation region. This structure justifies the design of our reduction heuristics and further demonstrates that the behavior exploited by COMPRESSM is intrinsic to training dynamics rather than specific to a single dataset or architecture.

## C    COMPRESSM ABLATIONS

In this section we present and discuss three ablations that justify the design choices adopted in COMPRESSM: first we illustrate the importance of the control theoretic approach of selecting the smallest HSVs for reduction as the actual capacity of recovery of models is limited, second, we look at the effect of increasing the number of reduction steps between the same initial and final state dimension, and last but not least, we evaluate the effect of performing reductions at different phases of training.

All ablations in this section are performed on the sMNIST dataset, with a single block LRU. The initial dimension considered is 256 and the final reduced model dimension is hardcoded to reach 32. The baselines that serve as references consist, like in the main text, in training the model without reduction with state dimensions 256 and 32. Similarly, results are obtained by averaging over ten random seeds.

### C.1    BALANCED TRUNCATION SANITY CHECK

The experimental setup consists in running COMPRESSM training with 4 equally spaced reductions (at the 5, 10, 15, 20k steps, total training for 200k steps). Each reduction keeps the same fraction $\rho = 0.595$ of the model, such that $256 \times \rho^4 = 32$. The ablation consists only in changing the selection scheme of the HSVs during reduction. We examine three variants: the correct balanced truncation approach removing the bottom HSVs, a random selection of HSVs to remove, and an adversarial example removing the top HSVs at each reduction step.

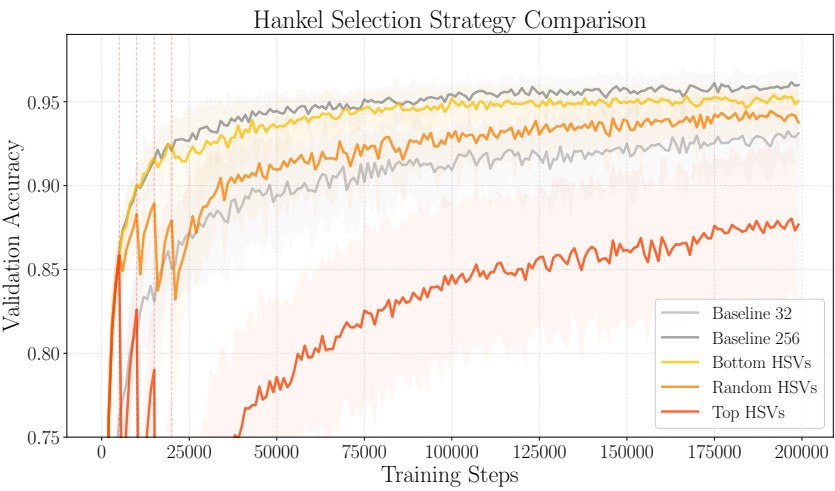

Figure 5: Validation accuracy evolution during sMNIST training with three HSV selection schemes. Each setup is averaged over ten seeds and the shaded regions around curves represent the standard deviation. The gray curves represent the non-reduced baselines with state dimension fixed to 32 and 256 (lighter and darker respectively). The yellow curve shows the correct reduction removing the bottom HSVs, the orange curve removes HSVs uniformly randomly, and the red curve removes the top HSVs. Dotted vertical lines show the steps at which reductions are performed.

Figure 5 shows the average model accuracy on the validation dataset during training, and Figure 6 depicts the HSVs evolution for a single seed across the three reduction schemes. Removing the top HSVs clearly incurs catastrophic performance damage, that is not recoverable even with 90% of subsequent training. These drops in performance are present yet less pronounced when removing random HSVs. In this case, the original gains gathered from training at larger state dimension are not completely lost, as the network is somewhat able to recover to perform slightly better than the baseline trained with dimension 32 from the start. In the case of proper bottom HSV truncation, there are no notable drops in performance in comparison to the 256 baseline, apart from a small departure at the final reduction step. The performance observations are also backed up by the HSVs evolution in each case. Indeed, we observe that the correct balanced truncation maintains a majority concentration of large HSVs throughout training, all within a single order of magnitude, maximizing

the per dimension contribution. Random truncation leads to a sparser spread of HSVs with many small surviving HSVs trailing an order of magnitude or two behind the leading dimensions. Expressivity is most diluted in the adversarial case, where removing the top values leads to dynamics with only a small proportion of HSVs carrying training while the largest bulk remains several orders of magnitude behind.

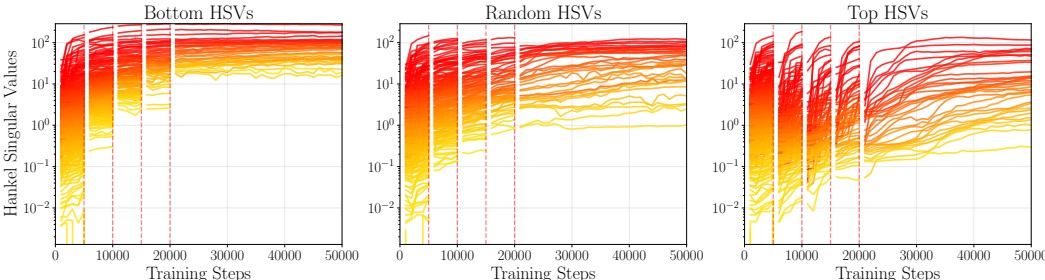

Figure 6: HSV evolution of a single seed for the three truncation schemes. From left to right we have the proper bottom HSV truncation, uniform random selection, and finally the adversarial top HSV truncation. Shades of red to yellow track the probable HSV trajectories between reductions. Dotted vertical lines show the steps at which reductions are performed. Note that all three plots share the same y axis limits for better HSV spread visualization. We show only the first 50k steps for better visualization of the HSV dynamics as the values evolve little for the subsequent portion of training.

The first conclusion from this ablation is that the COMPRESSM approach is sane, not only from a static dynamical system's perspective which is an established theoretic result, but crucially in the context of SSM training. Indeed, this shows that correct balanced truncation is indispensable as models are not able to recover from improper reductions. The argument for HSV continuity during training made in Section 3.3 provides the theoretical backing for this observation, as we establish that prominent as well as weaker HSVs maintain their relative ordering to a large extent. Hence, the argument holds both ways, small HSVs can be reduced without hindering long term training performance, but also, large HSVs cannot be removed without incurring performance losses, even with sustained post-reduction training.

## C.2 NUMBER OF REDUCTIONS

The experimental setup consists in running COMPRESSM training with $R$ equally spaced reductions occuring during the first 10% of training (total training for 200k steps), where we consider $R \in \{1, 2, 4, 8, 16\}$. Each individual reduction keeps the same fraction $\rho_R$ of the model, such that $256 \times \rho^R = 32$. The ablation studies the effect of smoother or more abrupt reductions on HSV evolution and performance.

Figure 7 shows the average model accuracy on the validation dataset during training, and Figure 8 depicts the HSVs evolution for a single seed across only the first four variants for clarity of presentation (1, 2, 4, and 8 reductions). The effect of increasing the number of reduction to go from an initial large state to the same final dimension seems to provide some performance improvement although marginal. The yellow curve for the most incremental reductions is the only one not to suffer a small drop in performance at the shared final reduction step of 20k steps. Intriguingly, the runs with 8 reductions do suffer a non-negligible drop yet are able to recover. Overall, the evidence suggests there might be a slight performance advantage in phasing out the reductions instead of clumping larger reductions into fewer occurrences. The compromise is to be found between performance gain and compute time. This is formalized and quantified in Section D. The evolution of HSVs in each case corroborates the observation of minimal effect of $R$ on the overall training procedure. Indeed, it is hard to discern any differences between the post-reduction HSVs in all cases considered, as all values seem to share similar distributions and populate the same order of magnitude.

This experiment sheds some light on the dependency of training dynamics on incremental reductions. Again, the HSV continuity analysis of Section 3.3 in the main text allows us to understand that whether we remove all the bottom HSVs at once or phase them out more progressively, the global ordering and evolution of HSVs is little affected. The subsequent post-training HSVs appear

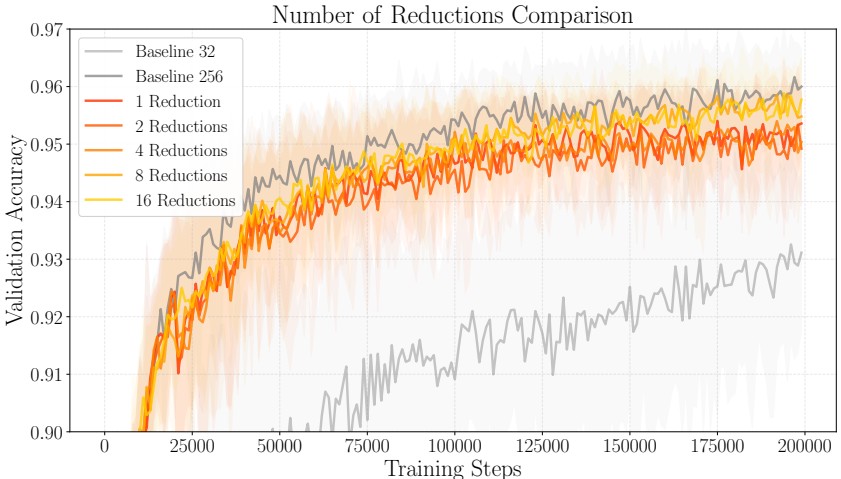

Figure 7: Validation accuracy evolution during sMNIST training with five increasing number of reductions between 256 and 32 (shades from red for $R = 1$ to yellow for $R = 16$). Each setup is averaged over ten seeds and the shaded regions around curves represent the standard deviation. The gray curves represent the non-reduced baselines with state dimension fixed to 32 and 256 (lighter and darker respectively).

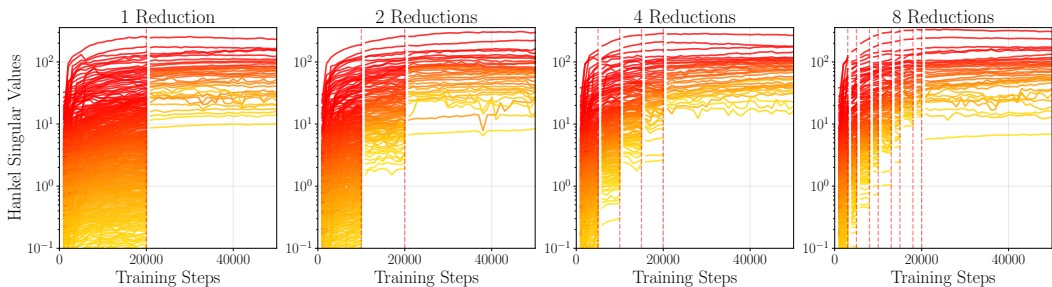

Figure 8: HSV evolution of a single seed for 1, 2, 4, and 8 reduction steps (from left to right). Shades of red to yellow track the probable HSV trajectories. Dotted vertical lines show the steps at which reductions are performed. Note that all four plots share the same y axis limits for better HSV spread visualization. We show only the first 50k steps for better visualization of the HSV dynamics as the values evolve little for the subsequent portion of training. We omit the 16 reductions for readability.

to behave very similarly. Incremental reductions might offer small performance benefits as other layers of the network as well as the optimizer state are less perturbed. Based on the model architecture, overhead costs of reduction, and performance optimality constraints, a suitable balance can be found for the frequency of reductions.

## C.3 REDUCTION WINDOW

The experimental setup consists in running COMPRESSM training with 4 equally spaced reductions at intervals of 5k steps, yet sliding the window during which the reductions are performed. Each reduction keeps the same fraction $\rho = 0.595$ of the model, such that $256 \times \rho^4 = 32$. We examine 5 windows: the scheme used in the main text applying reduction in the 0-20k window, then 25-45k, 50-70k, 100-120k and 150-170k step windows.

Figure 9 shows the average model accuracy on the validation dataset during training, and Figure 10 depicts the HSVs evolution for a single seed across only the first four variants for clarity of presentation (0-20k, 25-45k, 50-70k, 100-150k steps). Here also, there appears to be no conclusive evidence that early reductions lead to a larger drop in performance than later ones. Indeed, although the red curve (corresponding to reduction in the 0-20k window) appears to be slightly noisier than others, es-

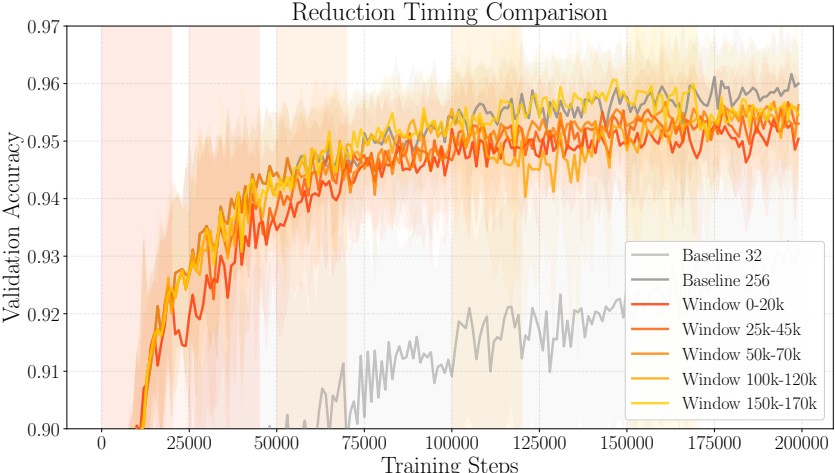

Figure 9: Validation accuracy evolution during sMNIST training with five increasingly late reduction windows (shades from red for 0-20k steps to yellow for 150-170k steps). Each setup is averaged over ten seeds and the shaded regions around curves represent the standard deviation. The vertical shaded regions delimit each reduction window with the corresponding color. The gray curves represent the non-reduced baselines with state dimension fixed to 32 and 256 (lighter and darker respectively).

pecially toward the end of training, its best performance validation accuracy does not underperform among the other reduced models. In fact, all reduced models reach practically the same maximum in the 170-200k end of training regime. This once more confirms that HSVs appear to behave in a continuous and relative rank preserving fashion, and largely justifies early reductions which lead to considerable training time gains as discussed and quantified in Section D. The noisiness observed in the performance curve for the earliest reduction is somewhat reproduced in the HSVs evolution plots, with a handful of the smallest values proving to be relatively erratic. However, the larger picture stays consistent with the observation that values maintain a similar spread and lie in the same order of magnitude regardless of reduction timing.

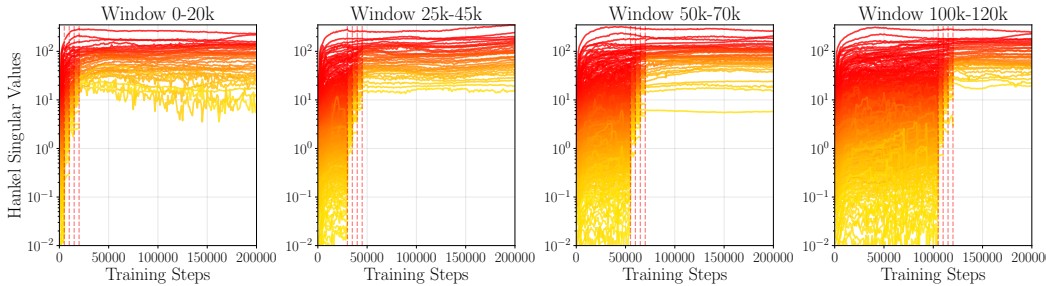

Figure 10: HSV evolution of a single seed for reductions within the 0-20k, 25-45k, 50-70k, 100-150k step windows (from left to right). Shades of red to yellow track the probable HSV trajectories. Dotted vertical lines show the steps at which reductions are performed. Note that all four plots share the same y axis limits for better HSV spread visualization. We omit the 150-170k steps window for readability.

Taken together, these results indicate that the precise timing of reductions has minimal impact on the final performance of the model. Even when reductions are performed very early—well before the model has partially converged—the HSV spectrum rapidly re-stabilizes, and the final validation accuracy matches that of runs where reductions occur much later. This aligns with our discussion in Section 3.3, where we showed that HSVs tend to evolve smoothly and preserve their relative ordering across training. Since delaying reductions does not offer any measurable performance benefit, applying them early remains the more effective choice in practice, as it preserves accuracy while providing substantially greater training-time savings.

## C.4 PRAGMATIC COMPRESSM.

Furthermore, we observe that reasonable balanced truncation practically does not hinder the upper bound performance, until it does. This observation can be used to slightly tweak the COMPRESSM algorithm. Although one does not have access to the upper bound trained with the large dimension all along, we can keep track of the validation metric during training and assume that while there are no drops in performance, reduction will incur no noticeable global performance penalties. In practice, the procedure is implemented with a simple safeguard. At each fixed fraction reduction step, we first save the current (pre-reduction) checkpoint. We then apply the reduction, train the model for a small number of steps, and evaluate it on the validation set. If validation performance continues to improve, we proceed to the next reduction. If performance degrades, we discard the reduced version and revert to the previously saved checkpoint, after which no further reductions are applied. This protocol ensures that model quality always remains close to that of the unreduced baseline without the need to explicitly predefine the number of reductions and a tolerance level in the absolute.

# D  TRAINING COMPLEXITY, SCALABILITY AND TIMING ANALYSIS

This section provides a detailed analysis of the computational costs involved in COMPRESSM, including both empirical timing results and a formal model of the wall-clock speedup achieved through in-training reductions. Beyond timing, we also include a study of the computational complexity of the key operations involved in the reduction pipeline and discuss pragmatic implementation considerations, such as GPU/CPU placement and library-level optimizations. Together, these analyses give a complete picture of where the dominant costs arise, which components scale favorably with reductions, and how to optimize end-to-end performance in practical implementations.

## D.1  IN-TRAINING SPEEDUP ANALYSIS

We train a single-block LRU model on the sMNIST dataset with an initial state dimension of $n = 256$ and a batch size of 50, such that at every 5000 gradient steps we artificially reduce the state dimension by removing 8 states. We record the wall-clock time of various operations during training on a single NVIDIA RTX 6000 GPU, broken down as follows:

1. **Gradient Steps:** Mean time spent on each gradient update during training,

2. **Inference:** Time spent on evaluation, divided into pure inference time and dataloading overhead,

3. **Model Reduction Pipeline:** Time spent on Hankel singular value (HSV) computation, model balanced truncation with diagonalization and replacement, and optimizer reinitialization.

Figure 11 summarizes the evolution of these components as the model is progressively reduced.

In plot (a), the average gradient-step duration decreases sharply with the state dimension, dropping by nearly a factor of three between $n = 256$ and $n = 8$. This confirms that the computational cost of training scales strongly with the state size, and that the benefits of dimensionality reduction manifest immediately within the training loop.

Plot (b) separates evaluation time into dataloading and inference components. Dataloading remains effectively constant across dimensions, as expected, while inference time exhibits a moderate reduction but at a slower rate than the gradient updates. This highlights that the primary performance gains of model reduction occur *in-training*, rather than in post-training inference.

Plot (c) details the components of the model-reduction pipeline: computation of Hankel singular values, generation of the reduced model through balanced truncation and diagonalization, and optimizer reinitialization. We observe that HSV computation dominates the total reduction time but decreases only slightly as the models shrink. Because dense matrix operations become unstable on the GPU for dimensions above roughly 200, HSV computations are performed on the CPU, introducing a constant overhead for transferring layers between devices. Consequently, the total reduction time does not scale down proportionally with the state dimension. Subsequent operations, including truncation, diagonalization, and parameter replacement, decrease substantially with model size, while optimizer reinitialization adds a small, constant overhead. It is important to note that a fixed JIT recompilation overhead of approximately 5 seconds is incurred at every reduction step, as new model shapes trigger re-compilation of the training graph—an overhead that dominates all other reduction costs and thus motivates minimizing the number of reductions performed.

Finally, plot (d) compares the relative time budgets for 5000 training steps across reduction stages. The breakdown clearly shows that the dominant gains arise within the gradient-step portion, which accounts for most of the overall runtime improvement—approximately $\times 2$ at $n = 128$ and $\times 3$ at $n = 8$. In contrast, dataloading and pure inference times show only marginal improvement, while the reduction overhead remains nearly constant, dominated by CPU-bound operations. These results illustrate where computational savings are primarily achieved and where residual overheads remain—demonstrating that in-training reductions yield the most substantial efficiency gains by directly accelerating the optimization loop.

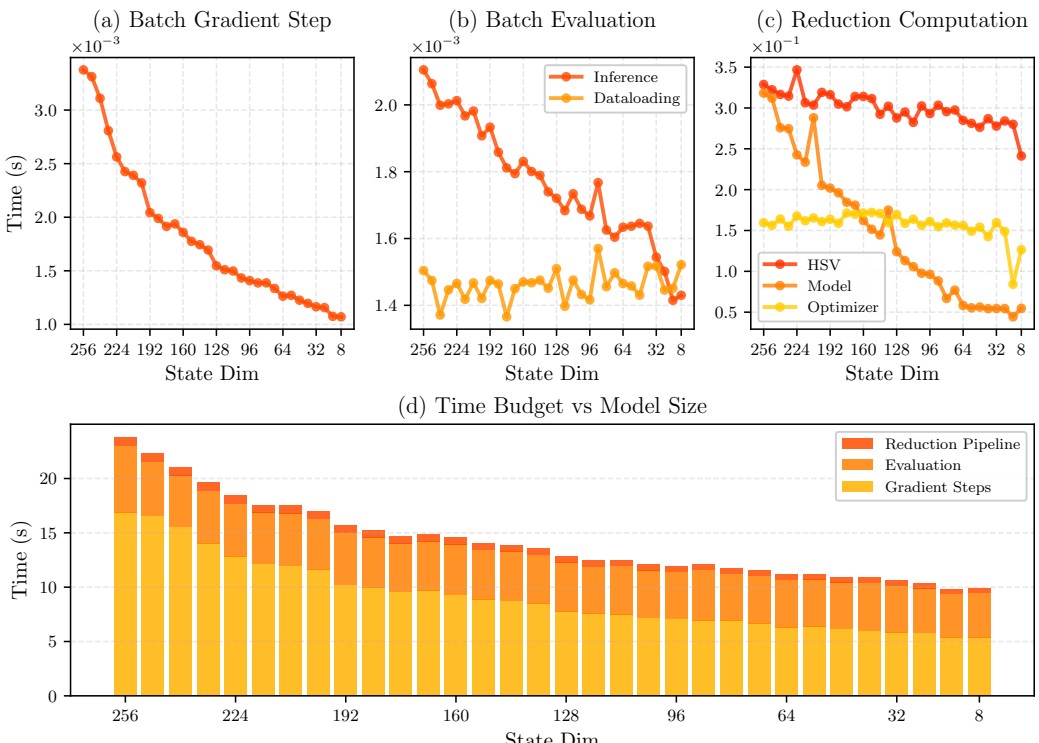

Figure 11: Timing analysis of in-training reductions. (a) Batch-gradient time decreases sharply with state dimension. (b) Evaluation time decomposed into dataloading (constant) and inference (moderately improved). (c) Breakdown of reduction-related computation (HSV, model truncation/diagonalization, optimizer reset). (d) Relative time budgets over 5000 steps, showing that most gains arise from faster gradient updates, while evaluation and reduction overheads remain comparatively small.

### D.1.1 QUANTIFYING IN-TRAINING COMPRESSION SPEEDUP

We now formalize the wall-clock speedup obtained when performing in-training model reductions. Let $s$ denote the number of gradient steps in an epoch and $E$ denote the total number of training epochs. Suppose we perform $R$ reductions at the ends of selected epochs. Let $n_0$ be the initial state dimension and denote by $n_k$ the state dimension *while training* in the $k$-th reduction phase (so $n_R$ is the final dimension). Let $E_k$ be the number of epochs run while the model has state dimension $n_k$, with $\sum_{k=0}^{R} E_k = E$. Define the following per-dimension timing functions:

- $t_{\text{train}}(n)$ — time per gradient step at state dimension $n$,
- $t_{\text{eval}}(n)$ — time per epoch for evaluation at state dimension $n$,
- $t_{\text{analysis}}(n)$ — time to run the reduction analysis (HSV computation, truncation/diagonalization, parameter replacement),
- $t_{\text{jit}}$ — fixed JIT recompilation overhead per reduction.

**Total wall-clock time without reductions:**

$$T_{\text{base}} = E\big(s\, t_{\text{train}}(n_0) + t_{\text{eval}}(n_0)\big).$$

**Total wall-clock time with reductions:**

$$T_{\text{red}} = \sum_{k=0}^{R} E_k\big(s\, t_{\text{train}}(n_k) + t_{\text{eval}}(n_k)\big) \;+\; \sum_{i=1}^{R}\big(t_{\text{analysis}}(n_{i-1}) + t_{\text{jit}}\big).$$

**Speedup:**

$$S = \frac{T_{\text{base}}}{T_{\text{red}}}.$$

**Reduction speedup prediction example** We instantiate the above formulas for the setup used in the experiments:

$$s = 5000, \qquad E = 40, \qquad R = 4.$$

Reductions occur at the ends of the first four epochs:

$$E_0 = 1, \; E_1 = 1, \; E_2 = 1, \; E_3 = 1, \; E_4 = 36.$$

State dimensions shrink from 256 to 96 via four reductions:

$$n_k = [256, 216, 176, 136, 96].$$

Measured timings (from Fig. 11):

| $n$ | 256 | 216 | 176 | 136 | 96 |
|---|---|---|---|---|---|
| $t_{\text{train}}(n)$ (s) | $3.4 \times 10^{-3}$ | $2.4 \times 10^{-3}$ | $1.9 \times 10^{-3}$ | $1.6 \times 10^{-3}$ | $1.4 \times 10^{-3}$ |
| $t_{\text{eval}}(n)$ (s) | $3.6 \times 10^{-3}$ | $3.4 \times 10^{-3}$ | $3.3 \times 10^{-3}$ | $3.2 \times 10^{-3}$ | $3.1 \times 10^{-3}$ |
| $t_{\text{analysis}}(n)$ (s) | 0.80 | 0.68 | 0.64 | 0.60 | 0.55 |

We time the JIT recompilation overhead cost as

$$t_{\text{jit}} \sim 5.0 \text{ s}.$$

**Baseline (no reductions).**

$$T_{\text{base}} = 40\left(5000 \cdot 3.4 \times 10^{-3} + 3.6 \times 10^{-3}\right) = 680.1 \text{ s}.$$

**With reductions.** Training+evaluation only subtotal :

$$T_{\text{red}(tr+ev)} = 298.6 \text{ s}.$$

Reduction overheads:

$$T_{\text{red}(ov)} = (0.80 + 0.68 + 0.64 + 0.60) + 4 \cdot 5.0 = 22.72 \text{ s}.$$

Total:

$$T_{\text{red}} = 321.3 \text{ s}.$$

**Speedup:**

$$S = \frac{680.1}{321.3} \approx 2.1.$$

**Interpretation.** Under the empirically measured per-dimension timings from the experiments, performing four aggressive early reductions that bring the state dimension from 256 down to 96 yields an overall wall-clock speedup of approximately $2.1\times$ for the full 40-epoch run. The calculation highlights the tension between per-step savings (which rapidly accumulate when many subsequent steps are executed at small $n$) and fixed reduction costs (CPU-bound analysis and JIT recompilation), illustrating how early reductions provide the most substantial gains while still incurring predictable overheads.

### D.2 REDUCTION PIPELINE COMPLEXITY

In this subsection we study the complexity of one application of the reduction pipeline (see Section 3.1).

Step 2: For diagonal transition matrix $\boldsymbol{A}$, as is the case for LRU, Equation 14 says that

$$\boldsymbol{P}_{ij} = \frac{\left(\boldsymbol{B}\boldsymbol{B}^\top\right)_{ij}}{1 - \lambda_i \lambda_j}, \quad \boldsymbol{Q}_{ij} = \frac{\left(\boldsymbol{C}^\top \boldsymbol{C}\right)_{ij}}{1 - \lambda_i \lambda_j} \quad \forall 1 \le i, j \le n. \tag{15}$$

For $\boldsymbol{B} \in \mathbb{R}^{n \times p}$, computing $\boldsymbol{B}\boldsymbol{B}^\top$ takes $\mathcal{O}(n^2 p)$ FLOPS. Similarly, computing $\boldsymbol{C}^\top \boldsymbol{C}$ for $\boldsymbol{C} \in \mathbb{R}^{q \times n}$ takes $\mathcal{O}(n^2 q)$ FLOPS. Thus, computing the Gramians $\boldsymbol{P}$ and $\boldsymbol{Q}$ requires $\mathcal{O}(n^2 p + 3n^2)$ and $\mathcal{O}(n^2 q + 3n^2)$ FLOPS, respectively.

Step 3: Computing the HSV form the spectrum of $\boldsymbol{PQ}$ requires computing the SVD of the product of the gramians, which requires $\mathcal{O}(n^3)$ FLOPS.

Step 4: finding the smallest rank $r$ that maintains at least $(1 - \tau)$ of the total energy requires computing the cumulative sum of the HSV, which is $\mathcal{O}(n)$ FLOPS.

Step 6: Transforming the system

$$(\boldsymbol{A}_b, \boldsymbol{B}_b, \boldsymbol{C}_b) = (\boldsymbol{T}^{-1}\boldsymbol{A}\boldsymbol{T}, \boldsymbol{T}^{-1}\boldsymbol{B}, \boldsymbol{C}\boldsymbol{T}) \tag{16}$$

to its balanced truncation costs $\mathcal{O}(n^3)$.

Thus, the total cost of balanced truncation is $\mathcal{O}(n^3 + n^2q + n^2p)$. Importantly, for LTI systems, the cost of balanced truncation is independent of the length of the input sequence.

To empirically validate these theoretical complexity bounds, we benchmark the key operations in the reduction pipeline for state dimensions ranging from $n = 1024$ to $n = 1$. Figure 12 shows the timing results for each dimension. The Gramian computation (Step 2) exhibits clear $\mathcal{O}(n^2)$ scaling, while the HSV computation via SVD (Step 3) and the balanced transformation (Step 6) both display $\mathcal{O}(n^3)$ behavior, consistent with our analysis. The full pipeline is dominated by the cubic operations, as expected from the total complexity of $\mathcal{O}(n^3 + n^2q)$.

Importantly, these benchmarks confirm that the reduction overhead remains modest even for large state dimensions. For example, at $n = 128$, the full reduction pipeline completes in under $0.1$ seconds, while at $n = 512$, it requires approximately 3 seconds. This overhead is negligible compared to the cumulative training time saved through in-training reduction, as demonstrated above in Section D.1, where models achieve significant speedups despite periodic reduction steps.

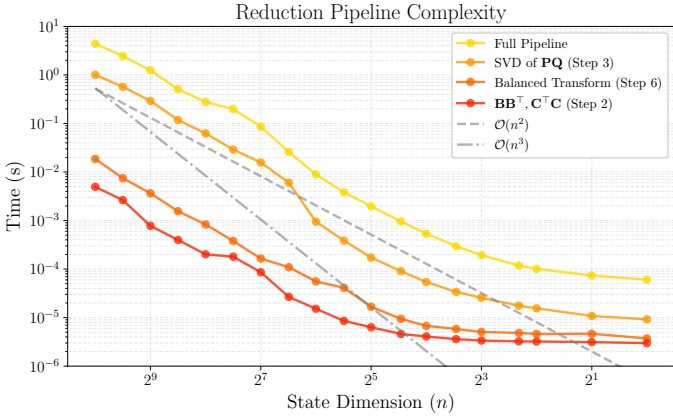

Figure 12: Empirical scaling of reduction pipeline operations. The Gramian computation (Step 2) is in red, the balanced transformation (Step 6) in dark orange, the HSV computation via SVD of $\boldsymbol{PQ}$ (Step 3) in bright orange and the full reduction pipeline in yellow. Gray reference lines show theoretical $\mathcal{O}(n^2)$ and $\mathcal{O}(n^3)$ scaling for asymptotic behavior comparison.

# E  SELECTIVE SSMs, SISO SYSTEMS AND MAMBA

In this section, we examine how COMPRESSM can be adapted to the broader landscape of selective state–space models, where the system dynamics depend explicitly on the input. We first outline practical strategies for handling such LTV/LPV architectures, emphasizing the computational challenges that arise when the system matrices vary with the signal. We then clarify the key distinction between per-channel SISO formulations and fully coupled MIMO state-space models, as this structural choice fundamentally shapes how state dimension influences model capacity and, consequently, how effective any reduction method can be. Finally, we present an in-depth case study on Mamba, demonstrating both the viability of our reduction framework in this selective, SISO setting and the specific limitations that emerge—highlighting where COMPRESSM succeeds, where the underlying architecture constrains performance gains, and how these insights point toward future extensions to richer MIMO-based designs.

## E.1  SELECTIVE SSMs HANDLING

Let $\mathcal{G}$ be a discrete *Linear Time-Varying (LTV)* system described by state equations:

$$
\begin{aligned}
\boldsymbol{h}(k+1) &= \boldsymbol{A}_k\, \boldsymbol{h}(k) + \boldsymbol{B}_k\, \boldsymbol{x}(k)\,, \quad \boldsymbol{h}(0) = \boldsymbol{h}_0 \\
\boldsymbol{y}(k) &= \boldsymbol{C}_k\, \boldsymbol{h}(k) + \boldsymbol{D}_k\, \boldsymbol{x}(k),
\end{aligned}
\tag{17}
$$

where the state matrices also depend on time.

Selective SSMs are built with dynamical systems that fall under a special case of LTV systems, the linear parameter varying (LPV) framework, with the parameter being the layer input. Such systems are also referred to as Linear Input Varying (LIV) and their general case state equations are given by,

$$
\boldsymbol{h}(k+1) = \boldsymbol{A}(\boldsymbol{x}(k))\, \boldsymbol{h}(k) + \boldsymbol{B}(\boldsymbol{x}(k))\, \boldsymbol{x}(k)\,, \quad \boldsymbol{h}(0) = \boldsymbol{h}_0
\tag{18}
$$

$$
\boldsymbol{y}(k) = \boldsymbol{C}(\boldsymbol{x}(k))\, \boldsymbol{h}(k) + \boldsymbol{D}(\boldsymbol{x}(k))\, \boldsymbol{x}(k),
\tag{19}
$$

For these systems, the controllability and observability Gramians are no longer stationary. In particular, for each possible input $\boldsymbol{x} \in \mathcal{X}$, one would in principle need to solve a set of Lyapunov inequalities of the form

$$
\boldsymbol{A}(\boldsymbol{x})\boldsymbol{P}(\boldsymbol{x})\boldsymbol{A}^\top(\boldsymbol{x}) - \boldsymbol{P}(\boldsymbol{x}) + \boldsymbol{B}(\boldsymbol{x})\boldsymbol{B}^\top(\boldsymbol{x}) \leq 0,
\tag{20}
$$

$$
\boldsymbol{A}^\top(\boldsymbol{x})\boldsymbol{Q}(\boldsymbol{x})\boldsymbol{A}(\boldsymbol{x}) - \boldsymbol{Q}(\boldsymbol{x}) + \boldsymbol{C}^\top(\boldsymbol{x})\boldsymbol{C}(\boldsymbol{x}) \leq 0,
\tag{21}
$$

so that the Gramians $\boldsymbol{P}$ and $\boldsymbol{Q}$ are input-dependent.

In practice, solving for fully input-dependent Gramians and applying the subsequent per input reduction is clearly neither computationally tractable nor practical. A common simplification is to seek input-invariant Gramians $\boldsymbol{P}, \boldsymbol{Q}$ that satisfy the inequalities for all $\boldsymbol{x} \in \mathcal{X}$; this reduces the problem to an LTI-like Lyapunov condition over all inputs, which can still be expensive for high-dimensional $\mathcal{X}$, when such a solution even exists. In practice this is still too constraining.

A cheaper alternative is simply averaging the dynamics over the input space:

$$
\bar{\boldsymbol{A}} = \frac{1}{|\mathcal{X}|} \sum_{\boldsymbol{x} \in \mathcal{X}} \boldsymbol{A}(\boldsymbol{x}), \quad \bar{\boldsymbol{B}} = \frac{1}{|\mathcal{X}|} \sum_{\boldsymbol{x} \in \mathcal{X}} \boldsymbol{B}(\boldsymbol{x}), \quad \bar{\boldsymbol{C}} = \frac{1}{|\mathcal{X}|} \sum_{\boldsymbol{x} \in \mathcal{X}} \boldsymbol{C}(\boldsymbol{x}),
\tag{22}
$$

The caveat is that the mean system may not be stable, controllable, or observable; one may therefore need to regularize the mean matrices to satisfy these assumptions before applying a single global reduction based on the LTI approach. We study the case of Mamba (Gu & Dao, 2024) and provide a practical implementation as well as results in Section E.3 (with the caveat discussed in the following section E.2).

## E.2  STATES IN MIMO VS SISO SYSTEMS

Sequence models based on state-space layers can differ in how their hidden state is used to map inputs to outputs. Indeed, some models employ a *multi-input multi-output* (MIMO) approach while others use a per channel *single-input single-output* (SISO) state-space structure.

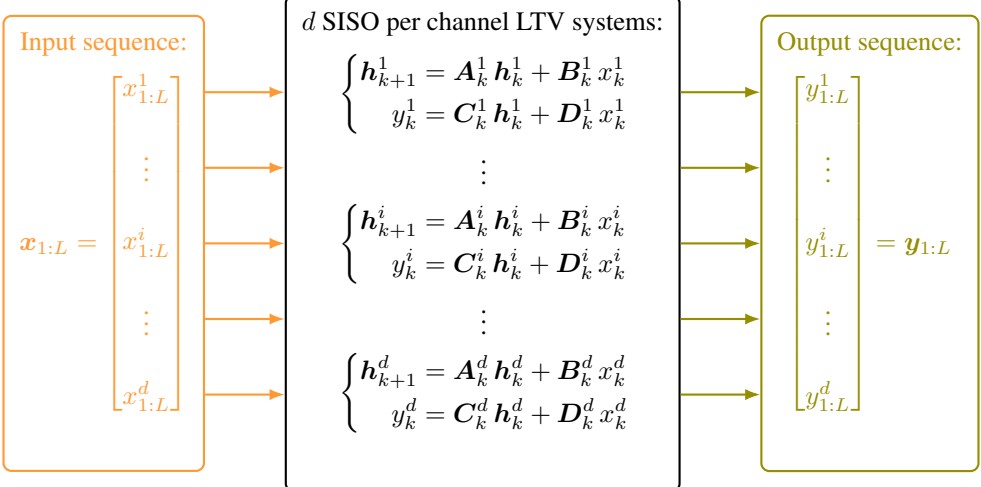

Figure 13: SISO state-space system applied independently to each input channel, with separate latent states per channel.

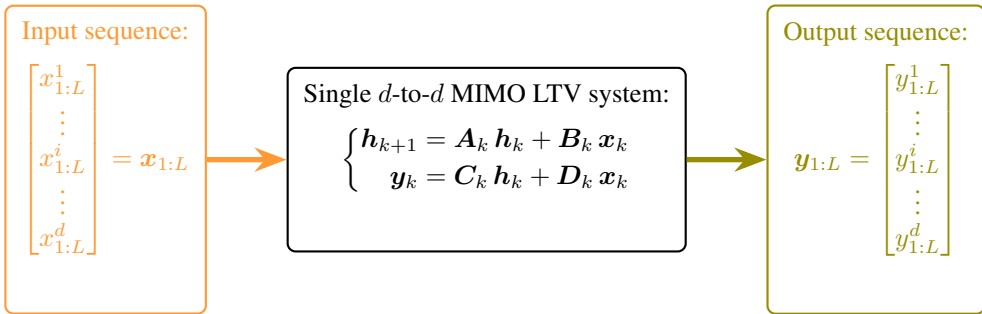

Figure 14: MIMO state-space system where a single latent state evolves jointly for all input channels and produces all outputs.

In a MIMO linear dynamical system, a state vector of dimension $n$ governs the evolution of an entire $d$-dimensional feature vector. The state therefore directly mediates interactions among all channels, and its expressive capacity contributes jointly to all output degrees of freedom. State size $n$ thus acts as a meaningful global capacity measure: increasing $n$ expands the shared dynamical subspace available to all features.

By contrast, many popular state-space architectures—including several LTI models (e.g., S4 (Gu et al., 2021), S5 (Smith et al., 2022)) and LTV models (e.g., Mamba (Gu & Dao, 2024), Liquid-S4 (Hasani et al., 2022) variants)—adopt a SISO formulation. In these designs, each feature channel is processed by an independent 1D state-space system, so the latent state predicts the evolution of a *single* input-output mapping rather than the entire feature vector. We provide visualizations of these differences in approach in figures 13 and 14. While per-channel SISO formulations scale efficiently due to parallelization, their per-channel factorization dilutes the expressive advantage normally associated with larger state dimensions. We observe that for LRA experiments, the state dimension hyperparameter does not seem to be a main performance driver (more on this in Section E.3). Future work will extend implementation to experiments where such models do benefit from larger states (language tasks are perhaps more suited).

Not all LTV architectures are limited to SISO formulations. Several state-of-the-art designs, such as Griffin (De et al., 2024), employ MIMO state transitions, while others —building on linear attention models (Katharopoulos et al., 2020)— like DeltaNet (Yang et al., 2025) use matrix-valued

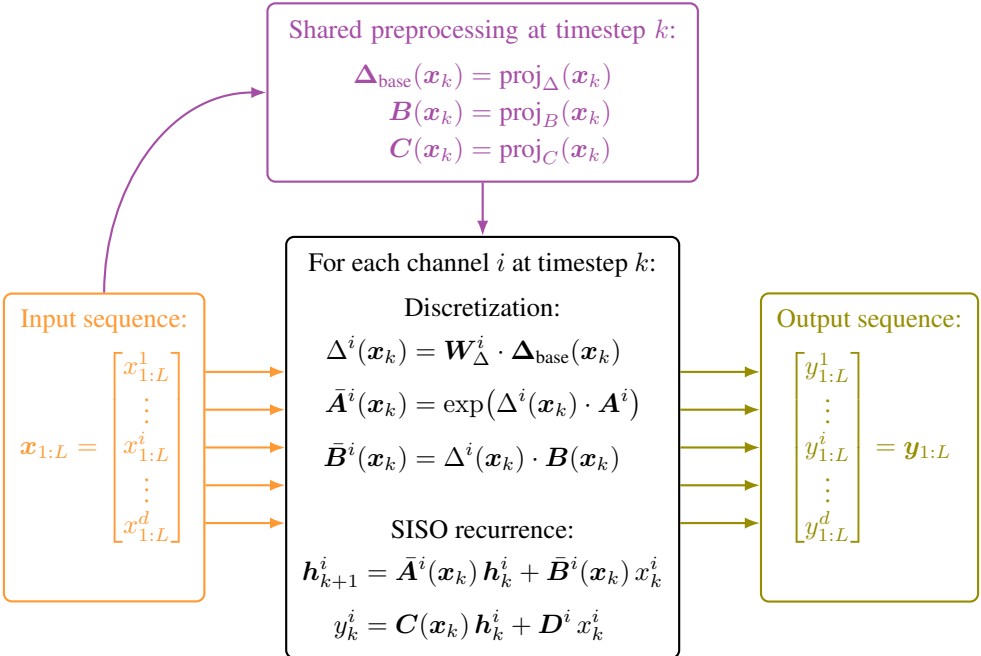

Figure 15: Vanilla S6 computation flow

dynamical updates. We consider extending our reduction framework to such systems an exciting direction for future work.

### E.3   MAMBA CASE STUDY

In this section, we take a deep dive into applying COMPRESSM to Mamba. Our goals are threefold: (i) to demonstrate the feasibility of selective system handling, (ii) to show that the framework's ideas extend cleanly to more intricate initializations such as Mamba's S6 layer, and (iii) to better understand the limitations and structural constraints that arise.

### E.3.1   MAMBA INITIALIZATION AND COMPUTATIONAL GRAPH

Reasoning about balanced truncation for Mamba is nontrivial because the per–channel SISO discrete LTV systems are coupled through shared projections and the input–dependent discretization. We begin by breaking down the computations inside the selective recurrent layer (S6), as illustrated by the diagram in Figure 15.

**Initialization and shared structure.** Mamba instantiates $d_{\text{inner}}$ single-input single-output (SISO) channels that all depend on a single sequence of projected features. At timestep $k$, the shared preprocessing block produces matrices $\boldsymbol{\Delta}_{\text{base}}(\boldsymbol{x}_k)$, $\boldsymbol{B}(\boldsymbol{x}_k)$, and $\boldsymbol{C}(\boldsymbol{x}_k)$ that are reused by every channel. Each channel $i$ owns a diagonal continuous-time drift matrix $\boldsymbol{A}^i$ and scales the shared discretization through a learned vector $\boldsymbol{W}_\Delta^i$, yielding $\Delta^i(\boldsymbol{x}_k) = \boldsymbol{W}_\Delta^i \boldsymbol{\Delta}_{\text{base}}(\boldsymbol{x}_k)$. The lifted discrete parameters are

$$\bar{\boldsymbol{A}}^i(\boldsymbol{x}_k) = \exp\big(\Delta^i(\boldsymbol{x}_k)\boldsymbol{A}^i\big), \qquad \bar{\boldsymbol{B}}^i(\boldsymbol{x}_k) = \Delta^i(\boldsymbol{x}_k)\boldsymbol{B}(\boldsymbol{x}_k), \qquad \bar{\boldsymbol{C}}^i(\boldsymbol{x}_k) = \boldsymbol{C}(\boldsymbol{x}_k).$$

The recurrence evolves according to

$$\boldsymbol{h}_{k+1}^i = \bar{\boldsymbol{A}}^i(\boldsymbol{x}_k)\,\boldsymbol{h}_k^i + \bar{\boldsymbol{B}}^i(\boldsymbol{x}_k)\,x_k^i,$$
$$y_k^i = \bar{\boldsymbol{C}}^i(\boldsymbol{x}_k)\,\boldsymbol{h}_k^i + \boldsymbol{D}^i\,x_k^i.$$

Although channels are nominally independent, they all consume the same $\boldsymbol{B}(\boldsymbol{x}_k)$ and $\boldsymbol{C}(\boldsymbol{x}_k)$, which couple their effective dynamics through the shared projections. This coupling is central to the difficulty of applying balanced truncation.

**Why balanced truncation is challenging.**

Classical balanced truncation assumes a time-invariant LTI system with fixed $(\boldsymbol{A}, \boldsymbol{B}, \boldsymbol{C})$. Mamba violates these assumptions in several ways:

(i) $\bar{\boldsymbol{A}}^i(\boldsymbol{x}_k), \bar{\boldsymbol{B}}^i(\boldsymbol{x}_k)$, and $\bar{\boldsymbol{C}}^i(\boldsymbol{x}_k)$ vary with inputs at every timestep, making the controllability and observability Gramians state-dependent;

(ii) $\boldsymbol{B}(\boldsymbol{x}_k)$ and $\boldsymbol{C}(\boldsymbol{x}_k)$ are shared across channels, so balancing each channel independently would break the shared projections shown in the first figure;

(iii) the selective-scan CUDA kernels rely on hard-coded loop bounds tied to the *full* state dimension, meaning that even if the desired reduced rank were known, the runtime cost would not change unless the kernels become rank-aware.

These factors make a naive application of balanced truncation both mathematically awkward and computationally inefficient.

### E.3.2 PRACTICAL REDUCTION WORKFLOW

To enable COMPRESSM with Mamba, we opt for a pragmatic implementation that relies on the recipe described and illustrated in Figure 16.

**Mean LTI surrogates.** During calibration, each channel $i$ accumulates running averages over inputs $\bar{\boldsymbol{B}}^i, \bar{\boldsymbol{C}}$, and $\bar{\boldsymbol{\Delta}}^i$. These define a stationary proxy

$$(\widetilde{\boldsymbol{A}}^i, \widetilde{\boldsymbol{B}}^i, \widetilde{\boldsymbol{C}}),$$

which approximates the expected discrete dynamics of channel $i$. This surrogate temporarily decouples the channels, enabling per-channel Gramian computation without destroying the shared projection structure.

**Per-channel balancing and rank selection.** From each surrogate we form controllability and observability Gramians $\boldsymbol{P}^i$ and $\boldsymbol{Q}^i$. Their product yields the Hankel singular values $\boldsymbol{\sigma}^i$, which determine the retained dimensionality $r^i$. We compute the balancing transform $\boldsymbol{T}^i$ and its inverse $(\boldsymbol{T}^i)^{-1}$ that simultaneously diagonalize the Gramians $\boldsymbol{P}^i$ and $\boldsymbol{Q}^i$.

Applying $(\boldsymbol{T}^i)^{-1}\widetilde{\boldsymbol{A}}^i\boldsymbol{T}^i$ and truncating to the first $r^i$ rows/columns produces $\boldsymbol{A}^i_{\mathrm{red}}$. Similarly,

$$\boldsymbol{B}^i_{\mathrm{red}} = (\boldsymbol{T}^i)^{-1}\widetilde{\boldsymbol{B}}^i\big[:r^i\big], \qquad \boldsymbol{C}^i_{\mathrm{red}} = \widetilde{\boldsymbol{C}}\,\boldsymbol{T}^i\big[:,:r^i\big].$$

A key implementation detail is that we *store both $\boldsymbol{T}^i$ and $(\boldsymbol{T}^i)^{-1}$*. This allows us to reapply the same rotation-and-truncation to fresh on-the-fly projections $\boldsymbol{B}(\boldsymbol{x}_k)$ and $\boldsymbol{C}(\boldsymbol{x}_k)$ at runtime. These cached transforms anchor the reduced subspace consistently across all timesteps.

**Runtime reuse of the stored transforms.** At execution time, Mamba still produces $\boldsymbol{B}(\boldsymbol{x}_k)$ and $\boldsymbol{C}(\boldsymbol{x}_k)$ at every timestep. For each channel $i$, we retrieve the cached $\boldsymbol{T}^i$ and $(\boldsymbol{T}^i)^{-1}$ and compute

$$\bar{\boldsymbol{B}}^i(\boldsymbol{x}_k) = (\boldsymbol{T}^i)^{-1}\boldsymbol{B}(\boldsymbol{x}_k), \qquad \bar{\boldsymbol{C}}^i(\boldsymbol{x}_k) = \boldsymbol{C}(\boldsymbol{x}_k)\boldsymbol{T}^i.$$

We then keep only the first $r^i$ coordinates before feeding the selective recurrence. The vector of ranks $\boldsymbol{r} = (r^1, \ldots, r^{d_{\mathrm{inner}}})$ is passed to the selective-scan CUDA kernels, which terminate their inner loops at $r^i$ for channel $i$. This eliminates all work on pruned coordinates and makes the runtime cost proportional to the reduced dimension.

Because the transforms are stored persistently and applied to both $\boldsymbol{B}(\boldsymbol{x}_k)$ and $\boldsymbol{C}(\boldsymbol{x}_k)$, every timestep is automatically consistent with the balanced dynamics of $\boldsymbol{A}^i$. Meanwhile, the kernels never expend computation beyond the retained modes. These mechanisms—depicted in Figure 16—preserve Mamba's expressive shared projections while enabling efficient selective reductions.

Overall, the workflow turns the highly coupled Mamba initialization into a collection of tractable mean systems, applies balanced truncation per channel, and enforces those ranks consistently throughout training and inference. Crucially, this is achieved without ever baking static reduced weights into the model: the reduction lives in the transforms and their runtime application, not in frozen parameter tensors.

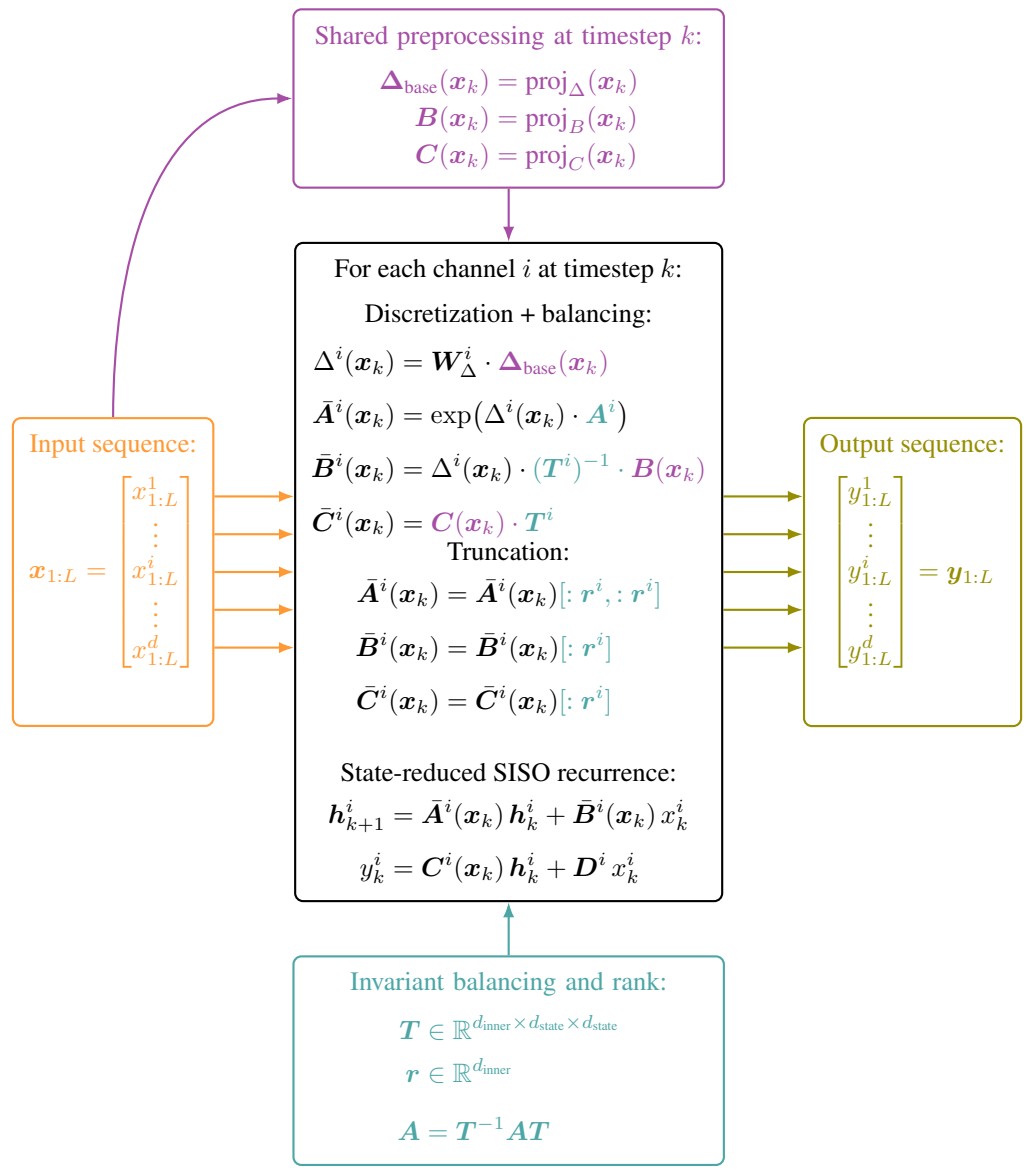

Figure 16: COMPRESSM S6 computation flow

### E.3.3 EXPERIMENTAL EVALUATION:

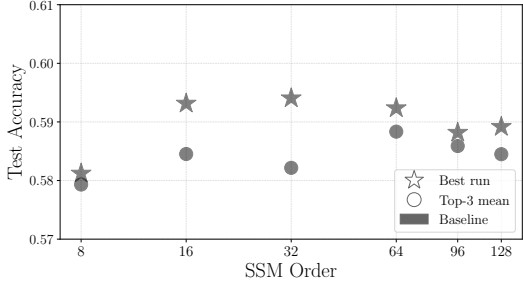

Figure 17: Performance of Mamba on CIFAR on five different random seeds. Grey markers correspond to non-reduced models. Stars correspond to best runs, circles to the top-3 mean.

We train Mamba on IMDB to showcase COMPRESSM for LTV systems (see Section E.1). The results can be found in Figure 18.

Figure 18a shows there is only a weak correlation between performance and state dimension in this setup, which we believe is due to the single-input-single-output (SISO) nature of Mamba. This observation is confirmed on CIFAR, performance is stable for state dimensions ranging from 128 to 8. Nevertheless, larger state-size does seem to help stabilize training, as indicated in Figure 18b.

Because of this weak correlation, we only evaluate COMPRESSM on IMDB to confirm the in-training speedup also holds for LTV systems.

We test the performance of COMPRESSM in this setting by compressing with $\tau = 0.001$. This means the model retains at least 99.99% of its energy at every reduction step. The final model has an average state dimension of $\sim 12$, starting from 128. This shows that the HSV spectrum is very tail-heavy, with most energy in just a handful of dimensions. This explains why random dropping of HSV (the red markers in Figure 18) performs competitively with balanced truncation (the orange markers), though at a higher variance.

Even though the correlation between state dimension and performance is not very strong, COMPRESSM yields competitive performance on Mamba, at a lower variance, especially when accounting for outliers. However, we believe the approach for LTV systems should be refined in later works to account for heavy-tailed distributions of HSV.

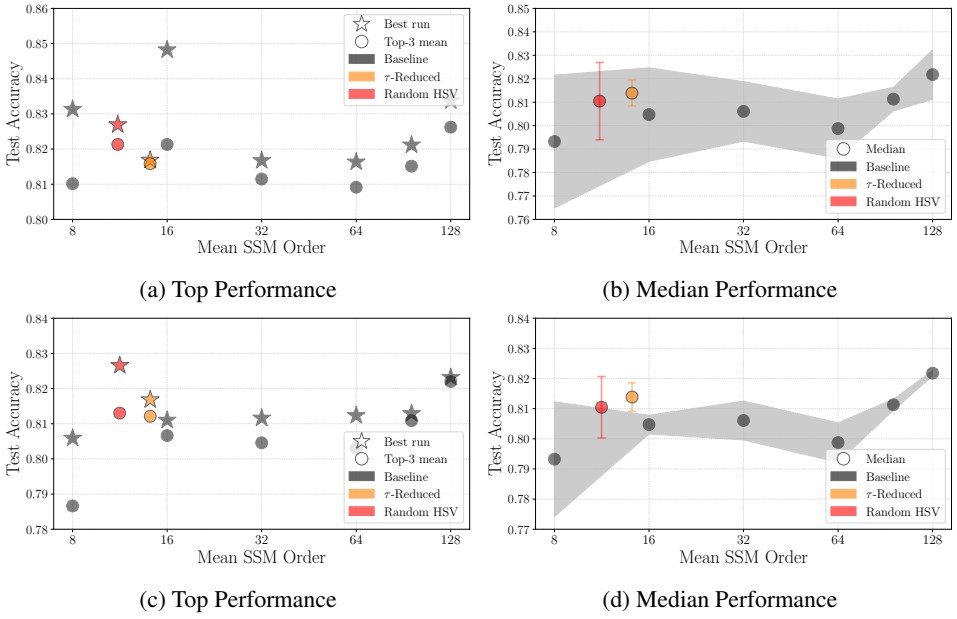

(a) Top Performance

(b) Median Performance

(c) Top Performance

(d) Median Performance

Figure 18: Performance of Mamba on IMDB on five random seeds. Grey markers correspond to non-reduced models, orange is a model reduced using $\tau = 0.001$. Red corresponds to random dropping of HSV. Plots on the left column show top-3 and top-1 performance (circle and star), plots in the right column mean performance. The first row considers all five random seeds, while the second row drops the best and the worst run.

Beyond maintaining competitive performance, our implementation achieves significant training time speedups for LTV systems. Figure 19 demonstrates that a model initialized with state dimension 128 and progressively reduced to $\sim 14$ using $\tau = 0.001$ trains in approximately the same time as a model that starts and remains at dimension 16. This represents a substantial speedup of $\sim 4\times$ compared to the full 128-dimensional baseline. These speedups are even more impressive than those reached with LRU, though that can be down to multiple factors as the codebases use different libraries (we run LRU in JAX and Mamba in PyTorch).

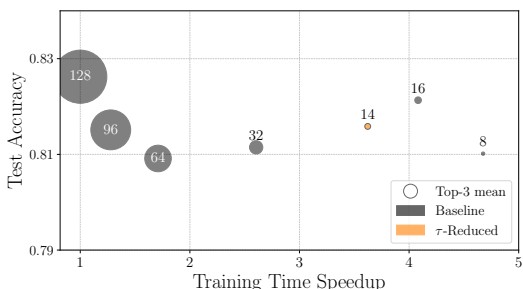

Figure 19: Training time speedup for Mamba on IMDB. The orange marker shows the COM-PRESSM-reduced model ($\tau = 0.001$, starting from dimension 128 and reducing to $\sim 14$). The gray markers represent the baseline runs. Data point size is proportional to final state dimension.

## F  COMPRESSM VERSUS OTHER COMPRESSION TECHNIQUES

In this section we pit COMPRESSM against other popular compression techniques. For SSMs, the paradigm of HSV nuclear norm regularization has established itself recently as an promising avenue. We discuss its shortcomings and make a case for the superiority of our framework both in terms of performance as well as training time. Similar arguments are also made about the student-teacher distillation paradigm. We remind the reader who might have ventured thus far that the main appeal of COMPRESSM is that it does not require training the full capacity model to completion before applying any reduction scheme, a point we hope to make a crystal clear case for in the experiments below.

### F.1  HANKEL NUCLEAR NORM REGULARIZATION

We investigate the differences in approach between COMPRESSM and spectral regularization techniques presented in the works of Forgione et al. (2024) which attempt both Hankel nuclear norm and modal $\ell_1$ regularization on the LRU model, and the contribution of Schwerdtner et al. (2025) which performs Hankel nuclear norm regularization on MIMO SSMs parametrized via scaled rotation matrices.

We shall discuss Hankel norm regularization only as $\ell_1$ regularization, although extremely cheap to compute, does not have any proper theoretical links to input-output mapping behavior apart from intuition on the frequency of dynamics as discussed in Forgione et al. (2024).

#### F.1.1  DIFFERENTIABLE HANKEL NUCLEAR NORM REGULARIZATION

We regularize the LTI dynamics inside each SSM layer by penalizing the nuclear norm of the product of the controllability and observability Gramians, $\boldsymbol{P_\theta}$ and $\boldsymbol{Q_\theta}$. This encourages the system to have rapidly decaying Hankel singular values, promoting low effective state dimension.

For a diagonal state-transition matrix $\boldsymbol{A_\theta} = \mathrm{diag}(\boldsymbol{\lambda_\theta})$, the Gramians admit closed-form solutions. The discrete-time controllability and observability Lyapunov equations,

$$\boldsymbol{A_\theta} \boldsymbol{P_\theta} \boldsymbol{A_\theta}^\top - \boldsymbol{P_\theta} + \boldsymbol{B_\theta} \boldsymbol{B_\theta}^\top = 0, \tag{23}$$

$$\boldsymbol{A_\theta}^\top \boldsymbol{Q_\theta} \boldsymbol{A_\theta} - \boldsymbol{Q_\theta} + \boldsymbol{C_\theta}^\top \boldsymbol{C_\theta} = 0, \tag{24}$$

have elementwise solutions

$$(\boldsymbol{P_\theta})_{ij} = \frac{(\boldsymbol{B_\theta} \boldsymbol{B_\theta}^\top)_{ij}}{1 - \lambda_{\boldsymbol{\theta},i} \lambda_{\boldsymbol{\theta},j}}, \tag{25}$$

$$(\boldsymbol{Q_\theta})_{ij} = \frac{(\boldsymbol{C_\theta}^\top \boldsymbol{C_\theta})_{ij}}{1 - \lambda_{\boldsymbol{\theta},i} \lambda_{\boldsymbol{\theta},j}}. \tag{26}$$

The Hankel singular values follow the same definition as in the main text (Eq. 4):

$$\boldsymbol{\sigma}(\boldsymbol{\theta}) = \mathrm{sort}_\downarrow \left( \sqrt{\mathrm{spec}(\boldsymbol{P_\theta} \boldsymbol{Q_\theta})} \right), \tag{27}$$

and we denote the $k$-th one by $\sigma_k(\boldsymbol{\theta})$.

The nuclear norm penalty is therefore

$$\|\boldsymbol{P_\theta} \boldsymbol{Q_\theta}\|_* = \sum_{k=1}^{n} \sigma_k(\boldsymbol{\theta}). \tag{28}$$

**Differentiability.**  All operations needed to compute $\sigma_k(\boldsymbol{\theta})$ are differentiable:

1. The SSM parameters $(\boldsymbol{A_\theta}, \boldsymbol{B_\theta}, \boldsymbol{C_\theta})$ are differentiable by design.
2. The diagonal Lyapunov solutions involve only multiplication and division.
3. The eigenvalues of $\boldsymbol{P_\theta} \boldsymbol{Q_\theta}$ are differentiable almost everywhere.
4. The square root and summation operations preserve differentiability.

Thus the regularized objective,

$$\mathcal{L}_{\text{reg}}(\boldsymbol{\theta}) = \alpha \, \mathcal{L}_{\text{task}}(\boldsymbol{\theta}) + \beta \sum_{l=1}^{L} \sum_{k=1}^{n_l} \sigma_{l,k}(\boldsymbol{\theta}), \qquad (29)$$

is fully compatible with automatic differentiation.

Backpropagation through the entire computation,

$$\boldsymbol{\theta} \;\mapsto\; (\lambda, \boldsymbol{B}, \boldsymbol{C}) \;\mapsto\; (\boldsymbol{P_\theta}, \boldsymbol{Q_\theta}) \;\mapsto\; \boldsymbol{P_\theta Q_\theta} \;\mapsto\; \boldsymbol{\sigma}(\boldsymbol{\theta}),$$

is supported natively in JAX.

**Numerical Stability.** During training we apply the following stabilizations:

- Add a small diagonal shift $\epsilon \boldsymbol{I}$ to $\boldsymbol{P_\theta}$ and $\boldsymbol{Q_\theta}$.
- Clamp eigenvalues of $\boldsymbol{P_\theta Q_\theta}$ to be non-negative before applying $\sqrt{\cdot}$.
- Clip denominators $1 - \lambda_i \lambda_j$ to avoid division near zero.

### F.1.2 PERFORMANCE AND COMPUTATIONAL COST

We use the sMNIST dataset with a single LRU block trained with an initial state dimension of 256 as our experimental setup. For ten different initial seeds, we train the model to completion (200k) with Hankel nuclear norm regularization as discussed in Section F.1.1 with $\alpha = 1$, $\beta = 0.1$ in Equation 29 (following values used in Forgione et al. (2024)), then reduce it to the values obtained for various tolerance levels with COMPRESSM presented in the main text in Table 1. More precisely, we apply balanced truncation down to state dimension 191, 148, 76, 47, 28, and 13, and evaluate the test performance of the reduced models. Table 6 provides the test performance and mean batch gradient step speed with respect to training at state dimension 256 *without* any regularization terms.

The results highlight three central limitations of relying on Hankel Nuclear Norm (HNN) regularization for producing compact state-space models:

1. **HNN regularization is computationally prohibitive.** Training must occur at the full state dimension with the regularizer applied at every step. Because evaluating the Hankel nuclear norm requires computing the eigenvalues of $\boldsymbol{P_\theta Q_\theta}$ at each gradient update, training is roughly $16\times$ slower than unconstrained optimization in our experiments.

2. **HNN-regularized models exhibit constrained performance.** The regularization forces rapidly decaying Hankel singular values, which reduces effective model capacity and leads to measurable accuracy degradation compared to unconstrained training—consistent with findings in Forgione et al. (2024). Even before any reduction, the HNN-trained model fails to reach baseline accuracy.

3. **HNN is dramatically less efficient than COMPRESSM for obtaining low-dimensional models.** For example, achieving a final dimension of 28 yields $96.9\%$ accuracy with COMPRESSM versus only $95.8\%$ when training the full 256-dimensional model with HNN and reducing afterward. Moreover, COMPRESSM reaches this result with a *46×* effective speed advantage, since reductions are performed early and training proceeds at much smaller state sizes.

Taken together, these points show that while HNN-regularized models are compressible, the combination of high computational cost and constrained accuracy makes them impractical for achieving high-performance low-dimensional SSMs. In contrast, COMPRESSM provides a more effective and scalable alternative, combining strong accuracy, efficient training, and substantial compression.

### F.2 KNOWLEDGE DISTILLATION

We furthermore compare COMPRESSM to knowledge distillation Hinton et al. (2015) on CIFAR10, where the teacher has a state dimension of 384 and the students have the same state dimension as the final, in-training compressed models (see Table 7).

Table 6: Top-3 mean sMNIST test performance (accuracy %) and batch gradient step speedup for different final state dimensions (rounded mean for COMPRESSM) with ten different random seeds. Results compare COMPRESSM, Baseline, and Hankel Nuclear Norm (HNN) regularization.

| Method | Metric | Final State Dimension | | | | | | |
|---|---|---|---|---|---|---|---|---|
| | | 13 | 28 | 47 | 76 | 148 | 191 | 256 |
| Baseline | Accuracy (%) | 92.6 | 96.0 | 95.9 | 96.4 | **97.3** | **97.3** | **97.3** |
| | Speed $\times$ | 3.1 | 2.8 | 2.7 | 2.4 | 2.0 | 1.7 | 1.0 |
| COMPRESSM | Accuracy (%) | **95.9** | **96.9** | **96.9** | **96.9** | 97.0 | 97.2 | - |
| | Speed $\times$ | 2.8 | 2.6 | 2.5 | 2.3 | 1.9 | 1.6 | - |
| HNN Regularization | Accuracy (%) | 91.7 | 95.8 | 95.8 | 95.8 | 95.8 | 95.8 | 95.9 |
| | Speed $\times$ | 0.06 | 0.06 | 0.06 | 0.06 | 0.06 | 0.06 | 0.06 |

The effective knowledge distillation loss for classification $\mathcal{L}$ is a superposition of the cross-entropy loss between the student and the targets and the Kullback-Leibler divergence between the teacher and the student with temperature scaling:

$$\mathcal{L} = (1 - \alpha)\mathcal{H}(y, \sigma(z_s)) + \alpha T^2 D_{\mathrm{KL}}(\sigma(z_t/T) \| \sigma(z_s/T)).$$

$\mathcal{H}$ denotes the standard cross-entropy loss, $y$ represents the ground truth labels, $z_s$ and $z_t$ are the logits of the student and teacher respectively, $\sigma(\cdot)$ is the softmax function, $T$ is the temperature parameter, and $\alpha$ is the balancing weight. We use the standard parameters $T = 2$ and $\alpha = 0.5$ (Timiryasov & Tastet, 2023).

Our experiments reveal that knowledge distillation performs better the closer the state dimension of the student is to the one of the teacher. Distilled models perform roughly on par with COMPRESSM if the state dimension of the student is similar to the one of the teacher. However, our in-training reduced models are able to maintain superior performance also for heavily reduced models, while knowledge distillation suffers from a clear drop-off.

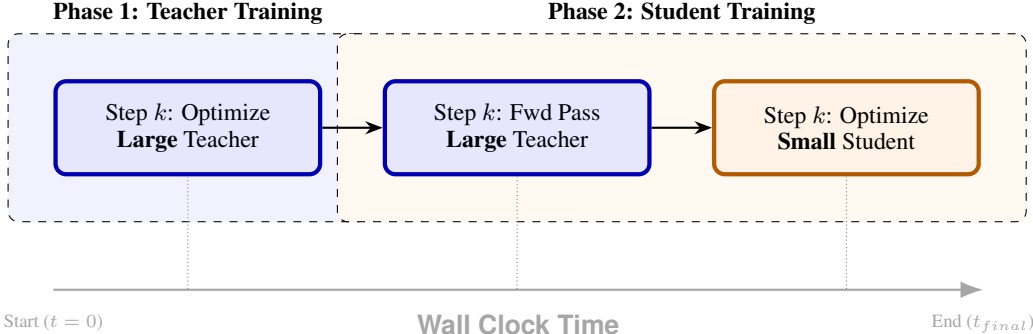

Figure 20: Wall clock time for training a student from a teacher via knowledge distillation. In the first phase, one trains a large teacher. In the second phase, its knowledge is distilled to a smaller student. Every forward step in the second phase requires a pass through both the student and the teacher.

We furthermore evaluate the total time it takes to obtain the final, distilled model. Knowledge distillation not only requires first training a high-dimensional teacher, but also doing a forward pass through the teacher while training the student to obtain the logits (see Figure 20). Consequently, even after the teacher has been trained to completion, the training speedup small students models have compared to larger ones is mitigated.

Table 7: Top-3 mean CIFAR10 test performance (accuracy %) and batch gradient step speedup for different final state dimensions (rounded mean for COMPRESSM) with ten different random seeds. Results compare COMPRESSM, Baseline, and Knowledge Distillation.

| Method | Metric | Final State Dimension | | | | | | |
|---|---|---|---|---|---|---|---|---|
| | | 57 | 93 | 126 | 161 | 214 | 327 | 384 |
| Baseline | Accuracy (%) | 78.2 | 81.8 | 83.7 | 84.2 | 84.9 | 86.0 | 86.5 |
| | Speed × | 1.69 | 1.62 | 1.53 | 1.43 | 1.22 | 1.03 | 1.0 |
| COMPRESSM | Accuracy (%) | **84.4** | **85.7** | **86.0** | **85.8** | **86.0** | 86.1 | – |
| | Speed × | 1.58 | 1.52 | 1.41 | 1.33 | 1.17 | 1.03 | – |
| Knowledge Distillation | Accuracy (%) | 79.4 | 83.5 | 84.4 | 85.3 | **86.0** | 87.0 | – |
| | Speed × | 0.55 | 0.52 | 0.61 | 0.51 | 0.49 | 0.45 | – |

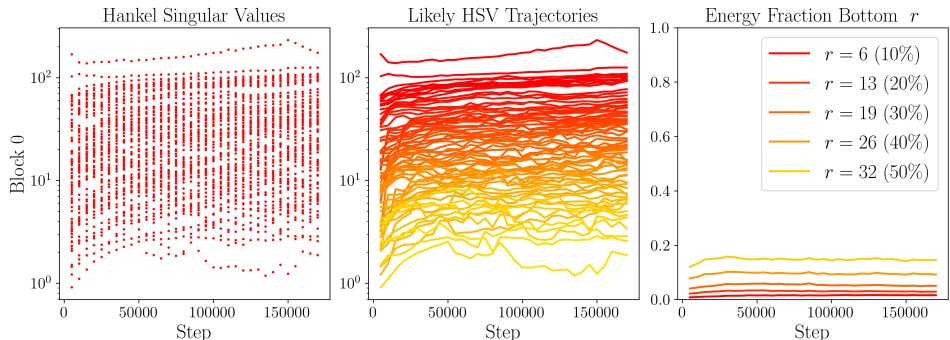

Figure 21: Single LRU block with state dimension of 64 on the sMNIST dataset.

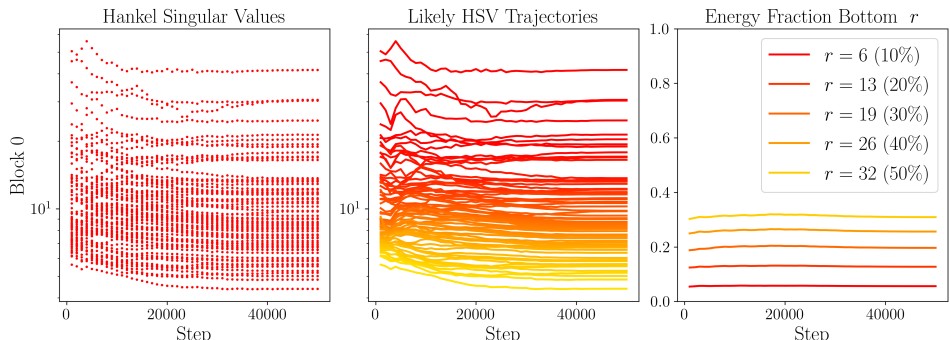

Figure 22: Single LRU block with state dimension of 64 on the IMDB dataset.

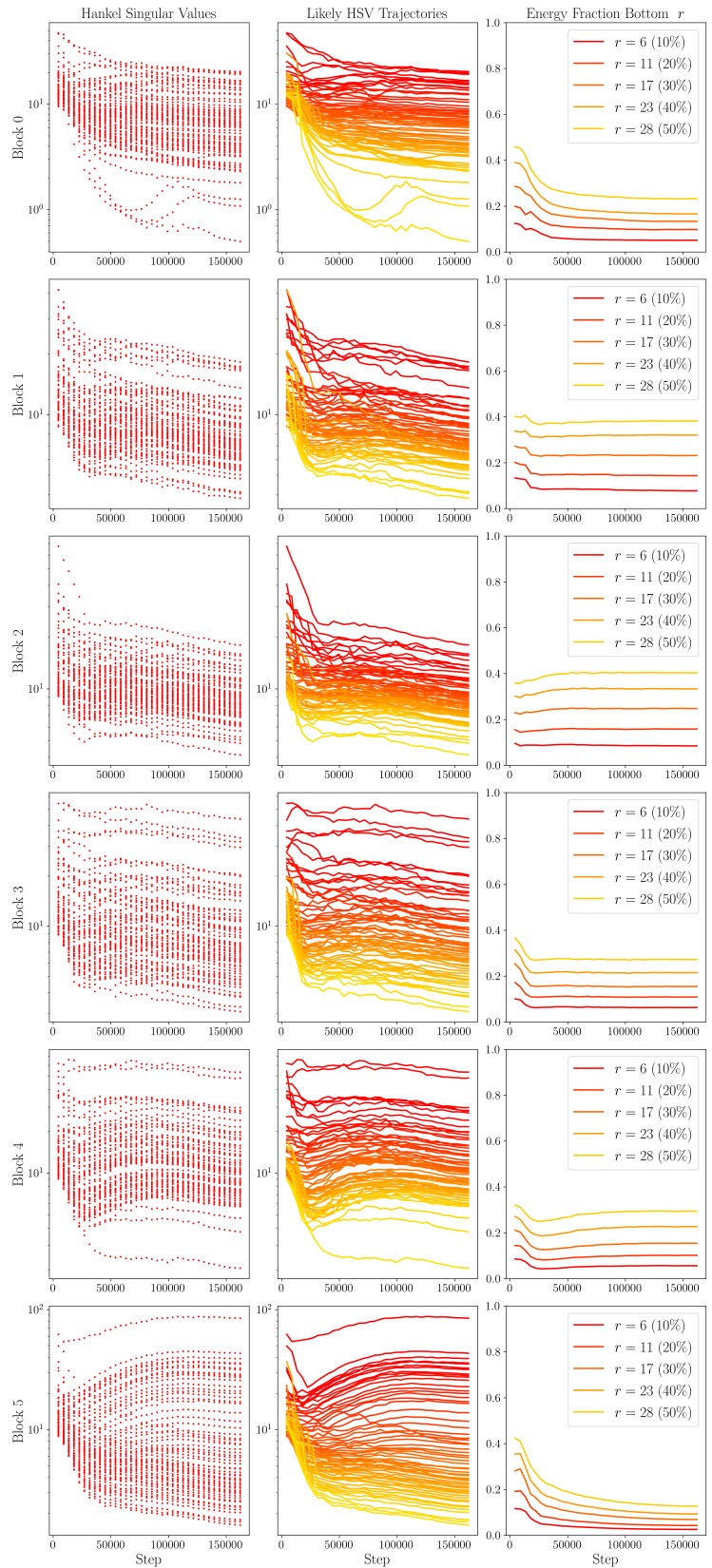

Figure 23: Six LRU blocks with state dimension of 57 on the CIFAR10 dataset.

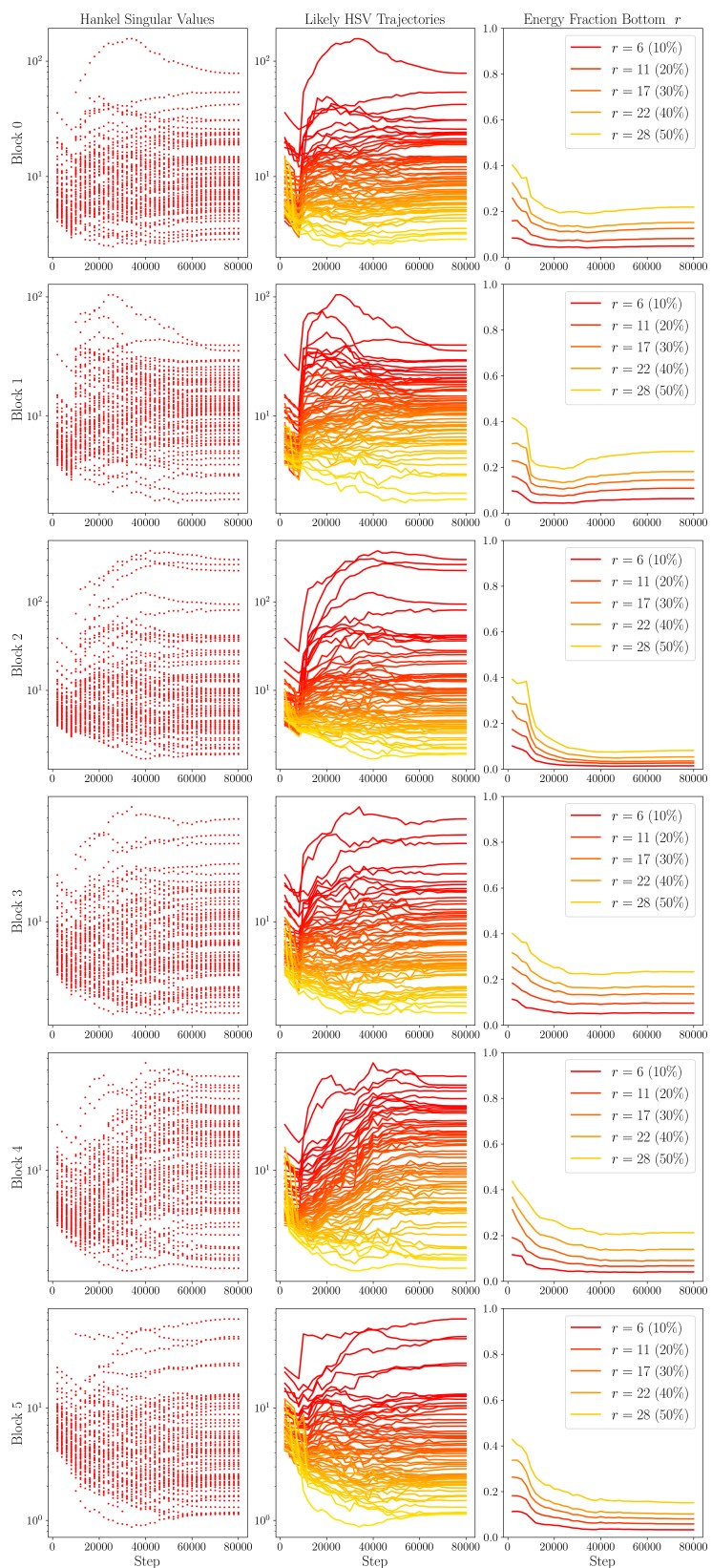

Figure 24: Six LRU blocks with state dimension of 56 on the ListOps dataset.

