# OpenReview forum: "The Curious Case of In-Training Compression of State Space Models"
_ICLR.cc/2026/Conference — ICLR 2026 Poster_

### Official Review · Reviewer_ELEM · 2025-10-20

**Soundness:** 2
**Presentation:** 3
**Contribution:** 2
**Rating:** 2
**Confidence:** 4

**Summary:**

The paper proposes an in-training compression technique of state-space models. It is based on application of the balanced truncation method adaptively during training. Empirical results on the long-range arena tasks show that the proposed method improves the model in certain cases.

**Strengths:**

* Accelerating the training speed of SSMs is an important task, and the proposed method shows the benefits of the method over directly training a smaller model from scratch.
* The method is grounded in rigorous analysis of LTI systems and the balanced truncation method from the ROM literature, which has been shown reliability.

**Weaknesses:**

* The idea proposed in the paper is very incremental. It looks like a naive application of the balanced truncation method at the training stage. The Weyl-type perturbation analysis is not new either. The proposed method can be totally viewed as a multi-stage fine-tuning process, where post-training compression and fine-tuning are applied alternately.
* The paper proposes an in-training compression method, but there is only one experiment that shows the training-time benefit in Figure 3b on a specific dataset. More importantly, reducing the latent dimension of SSMs does not significantly reduce the training speed (at least on LRA). That said, Figure 3b is visually misleading because the left end starts from somewhere between 0.55 and 0.6 instead of 0. More benefits would come from test time, especially if you plan to do it recurrently, but this is not the advantage of the proposed method over post-training compression.
* The accuracy of the proposed method is comparison with the baseline is also not compelling enough. Among the LRA tasks, only CIFAR and ListOps have shown some significant benefits. Also note that for ListOps, it is known that a small state dimension is totally capable of achieving SOTA performance. (For S4D, the SOTA performance comes from $n = 4$ and works well even when $n = 1$.)
* The paper would benefit from a discussion of the post-compression weights, and how it lands in the training loss landscape. This is a more interesting problem than the Weyl-type perturbation analysis. Note that the model is optimized via gradient descent; although compressing the system does not change its impulse response a lot, the parameters can be totally changed. This introduces more subtlety in CompreSSM.

**Questions:**

1. You choose a fixed $\tau$ throughout. If I am given a task, how would I go for a $\tau$ that achieves a time--accuracy balance that I expect? I think this is a crucial question, and especially given that you do not want to do any additional hyperparameter tuning (otherwise, why not just setting $\tau = 0$ and training only one model?).
2. Relatedly, have you experimented with having a scheduling for $\tau$, instead of fixing it throughout the training stage?
3. The "Baseline" in Table 1 is not well-described in the section. How did you train the "Baseline" model? Is it trained by fixing $n$ to be the rounded average "State dim" for all LTI systems in an SSM and train it from scratch?
4. Controllability and observability are not generally assumed in SSMs. How would you reconcile the fact that these two assumptions may not generally hold?
5. Hankel singular values are very sensitive to initialization and parameterization. For example, this work (https://openreview.net/forum?id=RZwtbg3qYD) shows that without careful initialization, you will start with fast-decaying singular values, and without careful parameterization, LTI systems are driven to low-degree ones during training. These are mainly due to the diagonal elements of $\mathbf{A}$ that are far apart from the imaginary axis. Is it possible to use that as a criterion for "when to truncate"?
6. Have you tried other ROM techniques other than balanced truncation? For example, those based on $\mathcal{H}_2$ optimization, moment-matching, or rational approximation?
7. You mentioned that the work can be adapted to time-variance (or selective) case; however, for selective SSMs, e.g., Mamba, the parameterization is completely different, and $\mathbf{B}$, $\mathbf{C}$, and even $\Delta$ are computed via more sophisticated mechanisms. This would make compression harder. Do you have a simple approach to that? If not, then better avoid overclaiming.

---

> ### Author Response · Authors · 2025-11-21
> **Initial response 1**
>
> ### Weakness 1:
> Incremental. Naive application of the balanced truncation method at the training stage. The Weyl-type analysis not novel. Multi-stage fine-tuning process.
>
> **Response:**
>
> We thank the reviewer for their assessment and take this opportunity to reiterate the contribution and novelty with CompreSSM.
>
> While the reviewer is correct in stating that our work is an application of balanced truncation method at the training stage, we respectfully refute the claim that this is done naively. In fact, the main contribution of the work is precisely making the case for early reduction that is the opposite of naive. The mere validity of reducing models early on in training without degradation is not trivial in any way. We establish mathematical machinery to measure the evolution of HSVs during training and demonstrate through empirical evidence that early reduction is principled.
>
> Also, the reviewer mentions that the application of Weyl's theorem for the tracking of HSVs is not new. We kindly invite the reviewer to provide us with precedents, as we have failed to find such applications in any of our references to papers in the realm of control theoretic analysis and reduction of SSMs. In our work, this is the very cornerstone that bridges the gap between naive hopeful reduction, and principled early truncation.
>
> Hence, the portrayal of the method as a discrete sequence of reductions and fine-tuning is one we do not share for two reasons. First, casting the problem as one of fine-tuning is misleading in our view: it suggests we have a capable trained model that we reduce, fine-tune, re-reduce and so on, when we actually show that reductions can be applied in the first 10% of training when the model is still very much acquiring skill towards solving the task. (Please refer to Appendix C.1, Figure 5 for a clear visualization of the training process). Second, this characterization completely undermines the dynamical aspect of HSV evolution during training, and hence the entire theoretical justification for the method, reducing it to a naive mechanistic succession of fine-tuning periods separated by reductions.
>
> While we fully acknowledge that the underlying mechanism of applying balanced truncation to SSMs is not itself novel, we maintain that our contribution is nonetheless substantive. We provide the first theoretical justification for performing such reductions *early in training*, and we demonstrate empirically that one can obtain compressed models that match the performance of large–state-dimension models at the compute cost of much smaller ones. We believe this combination of theory, practical guidance, and efficiency gains represents a meaningful contribution to the community. In this sense, and without overstating the significance of our work, we are confident that it is not merely incremental.
>
> ---
>
> ### Weakness 2:
>
> Single experiment training-time benefit. Not significantly reducing the training speed. Misleading Figure 3.b
>
> **Response:**
> We thank the reviewer for bringing up this important point which allows us to clarify how speedups are obtained and why they are systematic across datasets.
>
> But first we clarify the remark about Figure 3b. We hope the reviewer will agree that it is impossible to start with a large model, reduce it down to state dimension n_f in training and achieve a faster training time than a model initialized at n_f from the start. At best one would be close to the latter small model in terms of training time. This is the whole point of our plots, to show that with CompreSSM, training time is much closer to that of the small baseline than the large one. In fact, we have slightly modified Figure 3b to show relative speedups instead of training time in response to another reviewer's suggestion.
>
> Furthermore, it appears our presentation might have suggested that training time benefits are exclusive to specific datasets. This is absolutely not the case as all datasets benefit from a mechanistic improvement in training time. To address this ambiguity, we added a dedicated section in the Appendix (section D) that provides a full breakdown of the computational costs of Algorithm 3.1 and the reduction pipeline. This includes (i) empirical timing experiments (gradient-step time, inference time, and per-stage reduction overhead), (ii) a formal wall-clock model that quantifies the resulting speedups, and (iii) a full complexity analysis of the reduction pipeline, including Gramian computation, HSV extraction, and balanced truncation (see Appendix D.3).
>
> Crucially, the updated analysis makes explicit that the dominant source of training acceleration comes from the substantial reduction in *gradient-step time* once the state dimension is reduced. Since backpropagation through the SSM core scales with the state size, even moderate reductions compound over thousands of training iterations to yield significant total savings. The new section makes this tradeoff explicit and transparent.

---

> > ### Author Response · Authors · 2025-11-21
> > **Initial response 2**
> >
> > ### Weakness 3:
> > The accuracy of the proposed method is comparison with the baseline is also not compelling enough. Among the LRA tasks, only CIFAR and ListOps have shown some significant benefits. Also note that for ListOps, it is known that a small state dimension is totally capable of achieving SOTA performance. (For S4D, the SOTA performance comes from n=4 and works well even when n=1.)
> >
> > **Response:**
> > The reviewer raises a valid point, which we do not refute but attempt to provide some nuance to the remarks. First and foremost, we refer the reviewer to our extended discussion on per channel SISO (like S4D) models vs MIMO layers (like LRU), which we provide in Appendix E.2, also elaborated on in the context of Mamba in response to the reviewer's Question 7 below. We argue and are backed by observations such as the one made by the reviewer regarding S4D and ListOps, that SISO SSMs gain much smaller returns from increasing state dimension than MIMO variants.
> >
> > Furthermore, among the 5 LRA experiments considered and depicted in Figure 4 of the text, the AAN benchmark shows no correlation between state dimension and performance. In summary, CompreSSM provides performance gains over two baselines (ListOps and CIFAR10) in addition to the non-LRA MNIST, while for the two other state dimension depending experiments (IMDB and Pathfinder), naive CompreSSM is on par or at worse slightly behind the corresponding baselines. For the practitioner, the case for using CompreSSM can be made as training time is close to that of small models, while performance could potentially match that of larger instances. Given the limited evidence of performance degradation, the potential rewards well outweigh the risks when using our approach.
> >
> > ---
> >
> > ### Weakness 4:
> > The paper would benefit from a discussion of the post-compression weights, and how it lands in the training loss landscape. This is a more interesting problem than the Weyl-type perturbation analysis. Note that the model is optimized via gradient descent; although compressing the system does not change its impulse response a lot, the parameters can be totally changed. This introduces more subtlety in CompreSSM.
> >
> > **Response:**
> > We thank the reviewer for introducing an extremely interesting discussion. Indeed, understanding how post-compression weights relate to the surrounding loss landscape is an intriguing and largely open question. Conceptually, reducing an SSM by balanced truncation changes the parameterization even when the impulse response is nearly preserved, so in principle one might expect optimization to be sensitive to where the compressed model “lands’’ in parameter space.
> >
> > In practice, however, the actual effects of this observation are dwarfed by control theoretic considerations. Indeed, our ablations in Appendix C.1 suggest that this effect is empirically very small compared to the effect of *which* HSVs are removed for example. As shown in Figure 5, when reduction is performed using the correct smallest HSV truncation, the optimization trajectory after each compression resumes smoothly: the loss evolution, rate of convergence, and final accuracy remain close to the unreduced baseline, despite the fact that the weights themselves are changed. In contrast, altering only the *selection scheme* (random vs. top-HSV removal) produces sharp and unrecoverable drops in performance. This indicates that the dominant factor shaping the loss landscape after compression is the retained input–output geometry (i.e., the preserved HSV spectrum), not the specific coordinates of the compressed weights.
> >
> > Thus, although the question is compelling from a theoretical standpoint, our empirical evidence suggests that, in practice, balanced truncation lands the model in a region of parameter space that SGD can re-enter and optimize without noticeable instability.

---

> > > ### Author Response · Authors · 2025-11-21
> > > **Initial response 3**
> > >
> > > ### Questions 1 and 2:
> > > You choose a fixed τ throughout. If I am given a task, how would I go for a τ that achieves a time–accuracy balance that I expect? I think this is a crucial question, and especially given that you do not want to do any additional hyperparameter tuning (otherwise, why not just setting τ and training only one model?).
> > >
> > > Relatedly, have you experimented with having a scheduling for τ, instead of fixing it throughout the training stage?
> > >
> > > **Response:**
> > > The issue of optimal reduction scheduling is a very important one which we address in detail here. From a practical point of view, we want to train the best performing compact model as fast as possible. However, we do not have *a priori* access to the performance upper bound which requires training the initial large model to completion (something we aim to avoid). Thus the design criteria cannot rely on comparisons to the potential optimal performance. Elsewhere, the H-infinity guarantee bound between the original and reduced models (Equation (6) in the main text), and thus control theoretic approaches of relative energy conservation (in the HSV sum sense), provide little useful insights into the relationship between a given reduction at a specific training step down to a specific rank, and the subsequent effect on optimal performance of the fully trained model. This becomes more complex when considering models with multiple blocks all being reduced simultaneously, their effects through other non-linear layers and operations, and global reasoning about performance. Hence, two things can be concluded: (i) it is impossible to reliably predict the effect of a layer reduction on downstream performance *a priori*, and as a result, (ii) performance agnostic reduction inevitably risks producing sub-optimal results (in the sense of performance versus the original large model, as they might still be optimal against other models of same state dimension.
> > >
> > > That being said, we provide an alternative CompreSSM approach that inevitably requires violating the open loop irreversible single pass approach of *a priori* tolerance level selection. Section 3.2 of the revised manuscript provides a description of the approach. The idea is simple, since we cannot predict the long term effects of reductions, we only reduce until there are no negative short term damages to performance. For illustration, at fixed intervals we can reduce the model down to a fixed reasonable fraction of its size, say 10%. We save the pre-reduction checkpoint and train the model for a very small number of steps (0.1% of training steps) and evaluate it. If performance still improves we proceed with the reduced model. Else, we revert to the previous model and resume training without reductions. This pragmatic approach still benefits from similar speedup while ensuring we avoid applying too aggressive of reductions which can be detrimental to performance.
> > >
> > > ---
> > >
> > > ### Question 3:
> > > The "Baseline" in Table 1 is not well-described in the section. How did you train the "Baseline" model? Is it trained by fixing n to be the rounded average "State dim" for all LTI systems in an SSM and train it from scratch?
> > >
> > > **Response:**
> > > We thank the reviewer for bringing this lack of clarity to our attention. Indeed as described by the reviewer the baseline is exactly training the models from scratch at the rounded average dimensions mentioned in the table. We have modified the main text to alleviate any vagueness or ambiguity there.
> > >
> > > ---
> > >
> > > ### Question 4:
> > > Controllability and observability are not generally assumed in SSMs. How would you reconcile the fact that these two assumptions may not generally hold?
> > >
> > > **Response:**
> > > We appreciate the reviewer's attention to control-theoretical detail and the validity of our assumptions. We first note that states that are not controllable or observable are not desirable, as they cannot carry useful information between the input and the output. In our framework, stability of the learned SSM ensures that the controllability and observability Gramians exist and are unique. Uncontrollable or unobservable states naturally correspond to zero Hankel singular values, which are automatically truncated during balanced reduction. No additional assumptions or tuning are required: the only role of a tiny numerical regularization is to stabilize the diagonalization step in balanced truncation, not to alter the fact that such states are discarded. Thus, the procedure inherently handles potential rank deficiencies, and only the informative, controllable, and observable components are retained in the reduced model.

---

> ### Author Response · Authors · 2025-11-21
> **Initial response 4**
>
> ### Question 5:
> Hankel singular values are very sensitive to initialization and parameterization. For example, this work (https://openreview.net/forum?id=RZwtbg3qYD) shows that without careful initialization, you will start with fast-decaying singular values, and without careful parameterization, LTI systems are driven to low-degree ones during training. These are mainly due to the diagonal elements of $\mathbf{A}$ that are far apart from the imaginary axis. Is it possible to use that as a criterion for "when to truncate"?
>
> **Response:**
> The reviewer offers very valuable insight, and the paper mentioned looks at the problem of HSV efficiency from the perspective of initial parametrization while we assume we are given a model and discuss how to get rid of the low-influence singular values in early training. We have the reference to the related works section of the revised manuscript with the appropriate remark.
>
> The reviewer asks a very interesting question about using a signal from the diagonal values of $\mathbf{A}$ as a criterion to decide when to trigger reductions. Our understanding is that this would serve as a proxy for the actual computation of the HSVs. The reason we say that is that if we had access to the full HSV decomposition at each time step we could just observe the decay directly to make a decision on when to truncate. Given that for reasonable state dimensions, computing the HSVs at regular intervals is not a noticeable time sink during training, the question becomes whether we can rely on the notion of HSV decay for triggering reductions. Unfortunately we observe that in practice decay is not always occurring. In fact, we can point the reviewer to the evolution of the HSVs for the MNIST experiment illustrated in Figure 19 of the Appendix, where all HSVs actually seem to grow during training. Our experiments nonetheless show that this model is highly compressible. For this reason, we avoid heuristics based on monotonic HSV decay to avoid reliance on signals that might not generalize in practice.
>
> ---
>
> ### Question 6:
> Have you tried other ROM techniques other than balanced truncation? For example, those based on $\mathcal{H}_2$ optimization, moment-matching, or rational approximation?
>
> **Response:**
> The short answer is no we have not. MOR based on $\mathcal{H}_2$ optimization and applied to SSMs has come to life very recently [1], almost concurrently with our work. The approach requires solving an optimization problem with parameters defined as solutions of complex Lyapunov and Sylvester equations, and even uses Balanced Truncation classical solutions as initial guesses for its gradient-based optimization. The overhead of mathematical apparatus is manageable for offline applications but as is the approach appears to be too cumbersome for in-training deployment, especially given the marginal improvements reported. Elsewhere, both moment matching and rational approximation require constructing large Krylov subspaces or solving coupled Sylvester equations, often within iterative fixed-point loops, which can be computationally prohibitive at large dimensions. In addition, they do not provide any error bounds on the reduced models. Importantly, neither of these three alternatives guarantee stability of the reduced model.
>
> Balanced truncation provides the cleanest method not only to perform reductions, but also to track HSVs and understand their evolution, within a single unifying framework.
>
> [1] Sakamoto, Hiroki and Kazuhiro Sato. “Compression Method for Deep Diagonal State Space Model Based on H2 Optimal Reduction.” IEEE Control Systems Letters 9 (2025): 2043-2048.

---

> ### Author Response · Authors · 2025-11-21
> **Initial response 5**
>
> ### Question 7:
> You mentioned that the work can be adapted to time-variance (or selective) case; however, for selective SSMs, e.g., Mamba, the parameterization is completely different, and $\mathbf{B}$, $\mathbf{C}$, and even $\mathbf{\Delta}$ are computed via more sophisticated mechanisms. This would make compression harder. Do you have a simple approach to that? If not, then better avoid overclaiming.
>
>
> **Response:**
>
> We thank the reviewer for bringing this point up, we indeed do have ways to extend CompreSSM to architectures such as Mamba in spite of initialization and shared weights challenges.
>
> First, we mention that a challenge with popular SSMs both LTI (S4, S5) and LTV (Mamba, LiquidS4) is not as much the application of our pipeline, for which we have efficient LTV variants (namely averaging), yet it is the fact that they proceed on a per-channel Single-Input Single-Output (SISO) system basis. In fact, SISO dynamical systems are used in many architectures, where instead of having a vector-valued (MIMO) dynamical system predicting the evolution of the entire input feature vector, each individual channel of the input is handled by a SISO system to predict the corresponding channel in the output. Now instead of having a state that affects the mapping of an entire hidden feature of dimension 512 for example to an output of the same dimension, it is used to only predict the behavior of a 1D input-output map. This leads to many situations where the expressiveness and performance gains with larger state dimension are diluted. We discuss this in more detail in the revised Appendix E.2. We also highlight that not all LTV models must be SISO and we are exploring extensions to MIMO LTV SSMs (such as Griffin [1]) and even to matrix-valued dynamical systems akin to Linear Attention models (DeltaNet [2]). We aim to present such solutions as future work.
>
> Nonetheless, this has not prevented us from adapting CompreSSM to Mamba in order to corroborate our SISO vs MIMO observation as well as to demonstrate a practical implementation for such models. The presentation is provided in detail in our revised Appendix E.3.
>
> **Applying CompreSSM to Mamba.**
> Mamba presents unique challenges because its dynamics are input-dependent (selective) and operate on many independent SISO channels that share common projections. Our approach works as follows: During a calibration phase, we collect running averages of the input-dependent matrices $\mathbf{B}(\mathbf{x}_k)$ and $\mathbf{C}(\mathbf{x}_k)$ for each channel. These averages define a time-invariant surrogate system per channel, temporarily decoupling the channels so we can apply standard balanced truncation. For each channel $i$, we compute controllability and observability Gramians, extract Hankel singular values, and determine a reduced rank $r_i$ based on tolerance $\tau$. We then compute and store both the balancing transformation $\mathbf{T}_i$ and its inverse $\mathbf{T}_i^{-1}$.
>
> At runtime, these cached transforms are applied to the fresh input-dependent projections $\mathbf{B}(\mathbf{x}_k)$ and $\mathbf{C}(\mathbf{x}_k)$ at every timestep, rotating them into the balanced coordinate system before truncating to the first $r_i$ dimensions. The transition matrix $\mathbf{A}_i$ is similarly transformed and truncated once during calibration. Critically, we modified Mamba's selective-scan CUDA kernels to accept per-channel ranks so that inner loops terminate at dimension $r_i$ instead of the full state dimension. This ensures computational savings are realized in practice.
>
> **Performance results.**
> On CIFAR-10, baseline Mamba models show stable performance across state dimensions from 128 down to 8, confirming the weak correlation between state size and accuracy in SISO architectures.
> On IMDB, we compare unreduced baselines, CompreSSM with $\tau = 0.001$, and random HSV dropping. CompreSSM reduces from dimension 128 to approximately 14 (9× reduction) while achieving competitive accuracy with lower variance across seeds. Random HSV dropping performs surprisingly well due to the heavy-tailed HSV spectrum (again consistent with SISO discussion), though with higher variance.
>
> **Training speedup.**
> On IMDB, a CompreSSM-reduced model starting at dimension 128 and reduced down to dimension 14 trains in comparable time to the $n = 16$ baseline, representing a 4× speedup versus the full large model. This validates that rank-aware kernel modifications translate dimension reduction into practical computational savings. Speedups are consistently achieved across both LTI (LRU) and LTV (Mamba) architectures.
>
>
> [1] De, Soham et al. “Griffin: Mixing Gated Linear Recurrences with Local Attention for Efficient Language Models.” ArXiv abs/2402.19427 (2024)
>
> [2] Yang, Songlin et al. “Parallelizing Linear Transformers with the Delta Rule over Sequence Length.” ArXiv abs/2406.06484 (2024)

---

> > ### Author Response · Authors · 2025-11-21
> >
> > We sincerely hope that we have addressed the concerns of the reviewer satisfactorily in the revised version and would kindly ask the reviewer to update their score accordingly.

---

> > > ### Author Response · Authors · 2025-11-27
> > > **Invitation to engage**
> > >
> > > We are very grateful for your insightful comments and your time dedicated to our submission. We have put considerable effort into incorporating your suggestions into the revised manuscript, adding substantial experiments and analysis to meaningfully improve the quality and depth of the work.
> > >
> > > In addition to providing answers to the reviewers poignant remarks, we have discussed time complexity and the mechanistic speedups in depth in a dedicated section, introduced a pragmatic alternative to the tolerance based initialization (alleviating hyper parameter guessing), and addressed the case of selective state space models with a deep dive into the Mamba case (implementation, experiments and analysis).
> > >
> > > We very much welcome your engagement with any further questions or thoughts.

---

> > > > ### Comment · Reviewer_ELEM · 2025-11-28
> > > >
> > > > Dear Authors,
> > > >
> > > > Thank you for your careful rebuttal.
> > > >
> > > > * For the connection of Hankel singular values and error analysis, you can refer to these works back in the 80s:
> > > >
> > > >   [1] Glover, K., 1984. All optimal Hankel-norm approximations of linear multivariable systems and their L,$\infty$-error bounds. *International Journal of Control*, 39(6), pp.1115-1193.
> > > >
> > > >   [2] Adamjan, V.M., Arov, D.Z. and Kreĭn, M.G., 1971. Analytic properties of Schmidt pairs for a Hankel operator and the generalized Schur-Takagi problem. *Mathematics of the USSR-Sbornik*, 15(1), p.31.
> > > >
> > > >   [3] Glover, K., Lam, J. and Partington, J.R., 1990. Rational approximation of a class of infinite-dimensional systems I: Singular values of Hankel operators. *Mathematics of control, signals and systems*, 3(4), pp.325-344.
> > > >
> > > >   [4] Glover, K., Lam, J. and Partington, J.R., 1991. Rational approximation of a class of infinite-dimensional systems II: Optimal convergence rates of L$\infty$ approximants. *Mathematics of Control, Signals and Systems*, 4(3), pp.233-246.
> > > >
> > > >   In particular, the last two papers draw very comprehensive connections between the Hankel singular values and error estimates in multiple norms. Similar results are also found in the seminal book by Peller:
> > > >
> > > >   [5] Peller, V.V., 2003. *Hankel operators and their applications* (Vol. 15). New York: Springer.
> > > >
> > > > In general, I believe the quality of the paper improves during the rebuttal stage. I am willing to increase my score to 4. I think the paper can be made stronger with a better positioning, perhaps following one of the two options:
> > > >
> > > > 1. If the paper advocates balanced truncation for model compression during training, then it is essential to have a more careful evaluation of the impact of balanced truncation on the training dynamics. This is what it needs for the paper to gain substantial novelty.
> > > >
> > > > 2. Alternatively, the paper can also feature a comparison of canonical ROM methods, thus gaining insights into the connection between the ROM world and the ML world. As it currently stands, the choice of balance truncation, while classic, is heuristic. For instance, recently proposed data-driven ROM techniques, such as AAA are both stable and accurate, so the paper will be more comprehensive if it draws a comparison to those methods.
> > > >
> > > >   [6] Nakatsukasa, Y., Sète, O. and Trefethen, L.N., 2018. The AAA algorithm for rational approximation. *SIAM Journal on Scientific Computing*, 40(3), pp.A1494-A1522.
> > > >
> > > > Please note that OpenReview is blocking review revisions at this moment. I will update my rating once I am allowed to.
> > > >
> > > > Best regards,
> > > > Reviewer ELEM

---

> > > > > ### Author Response · Authors · 2025-11-28
> > > > > **Acknowledgement of reply**
> > > > >
> > > > > We thank the reviewer for providing us with useful references, and ideas for reinforcing novelty of the work, as well as for expressing their willingness to upgrade their score assessment.
> > > > >
> > > > > We are consulting the provided papers and considering the best way to establish a thorough evaluation of truncation on learning dynamics. We will update the reviewer as soon as possible in the hope of further satisfying their remaining concerns.
> > > > >
> > > > > Thanks again.

---

> > > > > > ### Author Response · Authors · 2025-11-30
> > > > > > **Response to reviewer's comment**
> > > > > >
> > > > > > We thank the reviewer again for their well documented reply (we use the same reference numbers as the reviewer)
> > > > > >
> > > > > > After consultation of the suggested references, we reiterate the significance of our contribution: none of the previous works (illustrious and technically masterful as they are) consider the same problem, let alone develop the same tools and solution.
> > > > > >
> > > > > > 1. explores extending optimal Hankel-norm approximation from the continuous to the discrete state space setup. The systems considered are static.
> > > > > >
> > > > > > 2. is a foundational paper for model reduction (although the term wasn't used yet when it was published), from the Hankel operator viewpoint. It establishes the optimal $ \mathcal{H}_\infty $ bound that any reduced model of degree $ r $ can achieve, with the minimal achievable error being exactly the next Hankel singular value.
> > > > > >
> > > > > > 3. studies singular values of Hankel operators for a class of infinite-dimensional systems and derives conditions and decay properties that determine how well such systems can be approximated by finite-dimensional ones. The systems are fixed (non-evolving), and the results are formulated at the operator-theoretic level.
> > > > > >
> > > > > > 4. builds on [3] to obtain optimal convergence rates of rational approximants of increasing order for a given infinite dimensional system. Again, the plant is static and the focus is order/error tradeoffs.
> > > > > >
> > > > > > 5. provides a comprehensive functional-analytic theory of Hankel operators, their singular values, and optimal approximation, which underpins the interpretation of HSVs in model reduction. However, its perturbation results concern abstract perturbations of symbols/operators.
> > > > > >
> > > > > > All of these papers contribute substantially to the understanding of Hankel operators, HSVs, and optimal rational or Hankel norm approximations, and thereby to the conceptual foundations of balanced truncation.
> > > > > > However, **none of them** look at the problem of tracking HSVs between evolving instantiations of a dynamical system, which is at the very heart of our work as we show how different state dimensions influence the input/output map as the SSM undergoes tens of thousands of gradient steps.
> > > > > >
> > > > > > We do not claim to "reinvent the wheel" regarding balanced truncation and its rich functional-analytic background. At the risk of repeating ourselves, we claim novelty in establishing machinery for **HSV monitoring** and its application to in-training reduction of SSMs.
> > > > > > In all honesty, we completely fail to see how these seminal works on Hankel operators would put the novelty of our contribution into question.
> > > > > >
> > > > > > This explanation also feeds into the reviewer's second suggestion for stronger positioning. We start by addressing AAA [6], which is not a direct alternative to balanced truncation in the usual MOR sense; it is a powerful rational approximant construction method, not a system-theoretic reduction method. Furthermore, our choice of balanced truncation as the underlying reduction mechanism provides the most direct HSV-principled approach that maintains system stability as well as error bounds.
> > > > > >
> > > > > > Finally, with respect to the first suggestion, should the reviewer again be referring to the post-compression weights in the loss landscape, our reinspection of the validation loss dynamics during training with reductions (appendix C.1) leads to the same conclusion provided in the reply to Weakness 4 :"[...] our empirical evidence suggests that, in practice, balanced truncation lands the model in a region of parameter space that SGD can re-enter and optimize without noticeable instability." Thus, although we agree that this phenomenon can be interesting, there is little practical incentive to look into it.
> > > > > > Yet, regarding the need to understand the impact of reductions during training, we argue that this is precisely the work and insight gathered from the three new ablations in Section C of the revised appendix (choice of HSVs to truncate, frequency of reductions, timing of reductions). This was also mentioned in the reply to the same weakness and serves as the justification for the pragmatic alternative CompreSSM protocol.

---

### Official Review · Reviewer_vMHR · 2025-10-27

**Soundness:** 3
**Presentation:** 3
**Contribution:** 2
**Rating:** 4
**Confidence:** 2

**Summary:**

The authors propose a complexity reduction strategy to accelerate SSM training. The idea is to remove non-essential dimensions of the state space by looking at the eigenvalues of the Hankel matrix during training. The pruned models perform better than a baseline where the dimensionality of the state space is fixed from the start.

**Strengths:**

- The idea reminds me of successful techniques for imposing sparsity when learning NNs.
- SSMs have become popular as surrogates of transformers. The possibility of compressing them during training in a principled way may help find a trade-off between computational costs and flexibility.

**Weaknesses:**

- While reducing the complexity of the trained model, the approach does not seem to imply complexity gains during the training phase.
- The analysis focuses on linear and time-invariant models, which are often outperformed by transformers on several tasks. It is unclear which parts of the idea can be applied to models with input-dependent matrices, such as Mamba-2.
- The method involves several hyperparameters, which may be expensive to tune.
- Experiments do not include alternative regularisation approaches.

**Questions:**

- Is this the first time Hankel-based regularisation techniques are applied to SSMs during training? What is the difference between the proposed method and more straightforward strategies, such as a simple $L_2$ regularisation of the matrices?
- How is the *normalised training time* mentioned in the figures defined?
- Would it make sense to include a simple alternative regularisation method as a second baseline?
- Is the truncation irreversible? Did you observe suboptimality issues associated with the greedy nature of the proposed method?
- Training complexity gains can be achieved if the reduction is irreversible and happens early enough. How is the latter compatible with the observation that there should be *"enough time in between successive reduction steps for the model to recover from pruning"*? How is the trade-off handled in practice?

---

> ### Author Response · Authors · 2025-11-21
> **Initial response 1**
>
> ### Weakness 1
> Complexity gains during the training phase.
>
> **Response:**
>
> We thank the reviewer for bringing up this important point. To address it, we added a dedicated section in the Appendix (section D) that provides a full breakdown of the computational costs of Algorithm 3.1 and the reduction pipeline. This includes (i) empirical timing experiments (gradient, inference, and reduction overhead), (ii) a wall-clock model that quantifies the speedups, and (iii) a full complexity analysis of the reduction pipeline (see Appendix D.3).
>
> Crucially, the updated analysis makes explicit that the dominant source of training acceleration comes from the substantial reduction in *gradient-step time* post-reduction. Since backpropagation through the SSM scales with the state size, even moderate reductions compound over thousands of training iterations to yield significant total savings. The new section makes this tradeoff explicit and transparent.
>
> ---
>
> ### Weakness 2
> Application to Selective SSMs.
>
> **Response:**
>
> A challenge with popular SSMs both LTI (S4, S5) and LTV (Mamba, LiquidS4) is not as much the application of our pipeline, for which we have efficient LTV variants, but the fact that they proceed on a per-channel Single-Input Single-Output (SISO) systems. In fact, SISO systems are used in many SSMs, where instead of a MIMO system predicting the evolution of the entire input vector, each individual channel is handled by a SISO system to predict the corresponding channel in the output. Now instead of having a state that affects the mapping of an entire hidden feature of dimension 512 for example to an output of the same dimension, it is used to only predict the behavior of a 1D input-output map. This leads to the expressiveness and performance gains with larger state dimension being diluted. We discuss this in more detail in the revised Appendix E.2. Not all LTV models have to be SISO, and we are exploring extensions to MIMO LTV SSMs (such as Griffin [1]) and even to matrix-valued dynamical systems akin to Linear Attention models (DeltaNet [2]).
>
> Nonetheless, we adapt **CompreSSM** to Mamba in order to corroborate our SISO vs MIMO observation as well as to demonstrate a practical implementation for such models. The presentation is provided in great detail in our revised Appendix E.3.
>
> **Applying CompreSSM to Mamba.**
> Mamba presents unique challenges because its dynamics are input-dependent (selective) with channels sharing common projections. Our approach: During a calibration phase, we collect running averages of the input-dependent matrices $\bar{\mathbf{B}}(\mathbf{x}_k)$ and $\bar{\mathbf{C}}(\mathbf{x}_k)$. These averages define a time-invariant surrogate system per channel, temporarily decoupling the channels so we can apply standard balanced truncation. For each channel $i$, we compute controllability and observability Gramians, extract HSVs, and determine a reduced rank $r^i$ based on tolerance $\tau$. We then compute and store both the balancing transformation $\mathbf{T}^i$ and its inverse $(\mathbf{T}^i)^{-1}$.
>
> At runtime, these cached transforms are applied to the fresh selective projections $\mathbf{B}(\mathbf{x}_k)$ and $\mathbf{C}(\mathbf{x}_k)$ at every timestep, rotating them into the balanced coordinate system before truncating to the first $r^i$ dimensions. The transition matrix $\mathbf{A}^i$ is similarly transformed and truncated once during calibration. Critically, we modified Mamba's selective-scan CUDA kernels to accept per-channel ranks, so they terminate their inner loops at dimension $r^i$ instead of the full state dimension. This ensures computational savings in practice.
>
> **Performance results.**
> On CIFAR-10, baseline Mamba models show stable performance across state dimensions from 128 down to 8, confirming the weak correlation between state size and accuracy in SISO architectures. On IMDB, we compare unreduced baselines, CompreSSM with $\tau=0.001$, and random HSV dropping. CompreSSM reduces from dimension 128 to approximately 14 (9× reduction) while achieving competitive accuracy with lower variance across seeds. Random HSV dropping performs surprisingly well due to the heavy-tailed HSV spectrum (again consistent with SISO discussion), though with higher variance.
>
> **Training speedup.**
> On IMDB, a CompreSSM-reduced model starting at dimension 128 and reduced down to dimension 14 trains in comparable time to the $n=16$ baseline, representing a 4× speedup versus the full large model. This validates that rank-aware kernel modifications translate dimension reduction into practical computational savings. Speedups are consistently achieved across both LTI (LRU) and LTV (Mamba) architectures.
>
> [1] De, Soham et al. “Griffin: Mixing Gated Linear Recurrences with Local Attention for Efficient Language Models.” ArXiv abs/2402.19427 (2024)
>
> [2] Yang, Songlin et al. “Parallelizing Linear Transformers with the Delta Rule over Sequence Length.” ArXiv abs/2406.06484 (2024)

---

> ### Author Response · Authors · 2025-11-21
> **Initial response 2**
>
> ### Weakness 3
> The method involves several hyperparameters, which may be expensive to tune.
>
> **Response:**
>
> The reviewer brings up a valid point as **CompreSSM** indeed requires guidance on when, how often, and how strongly to reduce models during training. In our reply to question 4, we discuss a pragmatic implementation of CompreSSM with more interpretable and performance-aware parameters.
>
> ---
>
> ### Weakness 4
> Experiments do not include alternative regularisation approaches.
>
> **Response:**
>
> The reviewer raises a valid concern, which we address with the addition of Hankel Nuclear Norm regularization experiments. They are discussed in more detail in our reply to Q3.
>
> ---
>
> ### Question 1
> Is this the first time Hankel-based regularisation techniques are applied to SSMs during training? What is the difference with simple regularisation of the matrices?
>
> **Response:**
>
> We thank the reviewer for their questions, as they allow us to emphasize the fundamental difference between **CompreSSM** and regularization techniques in the literature.
>
> First, we do **not** perform any regularization at all. Regularization methods imply penalizing some metric based on the matrices of the dynamical system during training, whether it is the Hankel nuclear norm (HNN) or the $\ell_1$ norm of the state matrices (the latter having no theoretical connection to input-output map preservation). These questions have been studied in [1] and [2], where truncation is only performed **after** regularized training at the full model dimension.
>
> We have added a comparison against Hankel norm regularization in Section F.1 of the revised Appendix showing both the exorbitant cost of regularization and its negative impact on performance.
>
> In contrast, **CompreSSM** does not constrain training in any way. It acts surgically on the network early during training to remove state dimensions that are estimated not to impact predictions, relying on balanced truncation theory for identification. This is justified by our result on HSV continuity with respect to gradient updates, which allows us to track their empirical behavior and observe favorable relative evolution for early reductions.
>
> [1] Forgione, Marco et al. “Model order reduction of deep structured state-space models: A system-theoretic approach.” 2024 IEEE 63rd Conference on Decision and Control (CDC) (2024): 8620-8625.
> [2] Schwerdtner, Paul et al. “Hankel Singular Value Regularization for Highly Compressible State Space Models.” ArXiv abs/2510.22951 (2025)
>
> ---
>
> ### Question 2
> Time normalization
>
> **Response:**
>
> We thank the reviewer for their attention to detail. The largest non-reduced model in Figure 3.b of the paper is the one with blocks of state dimension 384. It has a value of 1 in terms of normalized time, and all other variations are normalized with respect to it.
>
> Based on another reviewer's suggestion, we have slightly modified the figure to now show the **relative speedup** with respect to the same base run. This has been clarified in the RV.
>
> ---
>
> ### Question 3
> Alternative regularisation method as a second baseline?
>
> **Response:**
>
> The reviewer raises an important point, which allows us to expand on the reply to Q1.
>
> The price of principled regularization (Hankel nuclear norm) is exorbitant: at each gradient update, we must compute the eigenvalues of the Hankel matrix (via the square root of the product $\mathbf{P} \mathbf{Q}$, still $\mathcal{O} (n^3) $). A training run taking minutes would now require hours; runs already taking hours could stretch to days.
>
> Nevertheless, we ran an experiment on MNIST to provide a concrete comparison. Table 1 reports test accuracy and effective batch gradient step speed for various reduced state dimensions, comparing **CompreSSM**, baseline (no regularization), and HNN-regularized models.
>
> | Method | Metric | 13 | 28 | 47 | 76 | 148 | 191 | 256 |
> |--------|--------|----|----|----|----|-----|-----|-----|
> | Baseline | Accuracy (%) | 92.6 | 96.0 | 95.9 | 96.4 | **97.3** | **97.3** | **97.3** |
> |  | Speed × | 3.1 | 2.8 | 2.7 | 2.4 | 2.0 | 1.7 | 1.0 |
> | CompreSSM | Accuracy (%) | **95.9** | **96.9** | **96.9** | **96.9** | 97.0 | 97.2 | - |
> |  | Speed × | 2.8 | 2.6 | 2.5 | 2.3 | 1.9 | 1.6 | - |
> | HNN Regularization | Accuracy (%) | 91.7 | 95.8 | 95.8 | 95.8 | 95.8 | 95.8 | 95.9 |
> |  | Speed × | 0.06 | 0.06 | 0.06 | 0.06 | 0.06 | 0.06 | 0.06 |
>
> **Key observations:**
>
> - **HNN is extremely slow**: Gradient steps ~16× slower than unregularized training.
> - **Regularization limits performance**: HNN-trained models do not reach baseline performance due to overly aggressive decay of Hankel singular values.
> - **CompreSSM dominates** in both speed and performance: Final state dimension 28 achieves 96.9% accuracy with CompreSSM versus 95.8% with HNN, with a ~46× effective speed advantage.
>
> In short, HNN regularization is principled in theory but neither computationally feasible nor competitive versus **CompreSSM**.

---

> > ### Author Response · Authors · 2025-11-21
> > **Initial response 3**
> >
> > ### Question 4:
> > Truncation reversibility, suboptimality issues ?
> >
> >  **Response:**
> >
> >  We appreciate this question and use it as an opportunity to clarify the exact underlying protocol of CompreSSM. The truncation is indeed completely irreversible, with truncated states discarded and a new layer replacing its larger predecessor. The model subsequently trains at the smaller state dimension until the next eventual reduction.
> >
> > Regarding the issue of optimal reduction scheduling: from a practical point of view, we want to train the best performing compact model as fast as possible. However, we do not have *a priori* access to the performance upper bound which requires training the initial large model to completion. Thus, the design criteria cannot rely on comparisons to the potential optimal performance. Elsewhere, the H-infinity guarantee bound between the original and reduced models (Equation (6) in the main text), and thus relative energy conservation, provide little useful insights into the relationship between a given reduction, and the subsequent effect on optimal performance of the fully trained model. This becomes more complex when considering models with multiple blocks reduced simultaneously, their effects through other non-linear layers and operations, and global reasoning about performance. Hence, two things can be concluded: (i) it is impossible to reliably predict the effect of a layer reduction on downstream performance *a priori*, and as a result, (ii) performance agnostic reduction inevitably risks producing sub-optimal results (in the sense of performance versus the original large model, as they might still be optimal against other models of same state dimension).
> >
> > That being said, we provide an alternative CompreSSM approach that inevitably requires violating the open loop irreversible single pass approach of *a priori* tolerance level selection. Section 3.2 of the revised manuscript provides a description of the approach. The idea is simple, since we cannot predict the long term effects of reductions, we only reduce until there are no negative short term damages to performance. For illustration, at fixed intervals we can reduce the model down to a fixed reasonable fraction of its size, say 10%. We save the pre-reduction checkpoint and train the model for a very small number of steps (0.1% of training steps) and evaluate it. If performance still improves we proceed with the reduced model. Else, we revert to the previous model and resume training without reductions. This pragmatic approach still benefits from similar speedup while ensuring we avoid applying too aggressive of reductions which can be detrimental to performance.
> >
> > ### Question 5:
> >
> > Training complexity gains and early reductions. Tradeoff with recovery
> >
> >  **Response:**
> >
> >  We thank the reviewer for a raising a very important point which touches on the main mechanisms behind the efficacy of CompreSSM. First, we reiterate that indeed reduction is irreversibly applied early during training and that this is precisely why CompreSSM's training speedups are close to those of training the model at the small final dimension from the start. Compatibility between the statements holds due to the fact that the two processes occur at very different time scales. When applying reductions that are slightly too aggressive, it can take models a few hundreds of training steps to get back to their pre-reduction performance, while reductions are typically spaced by thousands of steps. All of that still fits in the first 10% of training in almost all the experiments presented. These timescales can be best visualized in the ablation experiment of the revised Appendix C.1, more specifically in Figure 5.
> >
> > Also, we realize that our original choice of words is extremely misleading in suggesting that CompreSSM requires "recovery” after each reduction step; in fact, when applied correctly, no such recovery is needed. Balanced truncation enjoys a classical guarantee: the H-infinity error between the original and reduced systems is bounded by twice the sum of the truncated Hankel Singular Values (HSVs). Assuming we properly truncate very low-impact HSVs, the perturbation to the dynamical system is inherently small, and thus the training trajectory continues smoothly without any noticeable loss that would necessitate recovery.
> >
> >  This behavior is made explicit in our HSV-selection ablation (Appendix C.1), where validation accuracy remains stable across reduction events so long as the removed modes correspond to small HSVs. In contrast, if one removes impactful directions the model suffers immediate and *irreversible* performance drops (even with 90% of training remaining)
> >
> >  In short, properly applied balanced truncation does *not* require recovery because it does not inflict meaningful damage on the model. Improper or excessive reduction, however, produces damage that subsequent training fails to fully repair. The RV has been modified to remove this ambiguity.

---

> > > ### Author Response · Authors · 2025-11-21
> > >
> > > We sincerely hope that we have addressed the concerns of the reviewer satisfactorily in the revised version and would kindly ask the reviewer to update their score accordingly.

---

> ### Author Response · Authors · 2025-11-27
> **Invitation to engage**
>
> We reiterate our thanks for your insightful comments and your time dedicated to our submission. We have put considerable effort into incorporating your suggestions into the revised manuscript, adding substantial experiments and analysis to meaningfully improve the quality and depth of the work.
>
> Namely, we have discussed time complexity and speedups in depth in a dedicated section, addressed the case of selective state space models with a deep dive into the Mamba case (implementation, experiments and analysis), have added a study and comparison against Hankel Nuclear Norm regularization, introduced a pragmatic alternative to the tolerance based initialization (alleviating hyper parameter guessing), and clarified the (lack of) recovery requirement of CompreSSM.
>
> We very much welcome your engagement with any further questions or thoughts.

---

### Official Review · Reviewer_p3jH · 2025-10-31

**Soundness:** 3
**Presentation:** 3
**Contribution:** 3
**Rating:** 6
**Confidence:** 3

**Summary:**

This paper proposes a method for compressing State Space Models (SSMs) during training rather than after training. The key innovation is using balanced truncation from control theory to identify and remove unimportant state dimensions while the model is being trained. The authors leverage Hankel Singular Values (HSVs) to measure the importance of each state dimension. Based on this information, they truncate low-importance dimensions in training, achieving computational speedups while maintaining or improving model performance compared to models trained directly at smaller dimensions.

**Strengths:**

- Novel application of control theory to SSM compression during training, with good theoretical grounding using Hankel Singular Values and Weyl's theorem to justify the approach
- The method is architecture-agnostic and works with different SSM variants
- Empirical validation showing that dominant HSVs are rank-preserving during training
- Achieves both training speedup over full dimension and better performance than smaller dimensions
- Well-written paper with clear mathematical exposition

**Weaknesses:**

1. Only tested on LRU architecture, not on other SSMs like Mamba, S4, etc.
2. No comparison with other compression techniques like knowledge distillation, quantization, or pruning
3. Method requires correlation between state dimension and model performance (acknowledged in the paper)
4. Needs sufficient training steps between reduction steps for model recovery (like pruning and QAT)
5. Missing ablation studies on key hyperparameters (reduction frequency, threshold values)
6. No analysis of what information is preserved/lost during truncation

**Questions:**

1. Step 5 of the algorithm (Section 3.1): What fraction values work well in practice? The paper doesn't provide specific guidance
2. Can you provide examples where there's no clear correlation between state dimension and model performance? I cannot think of such applications
3. Are you evaluating Sequential MNIST (SMNIST)? The paper mentions MNIST results but doesn't clarify if it's SMNIST
4. How does the computational cost of computing Gramians scale with state dimension?
5. Have you considered inverting the x-axis in Figure 3 to show "training time speedup" instead of "normalized training time"? This would make the Pareto front point to the top-right, which is more intuitive

---

> ### Author Response · Authors · 2025-11-21
> **Initial response 1**
>
> ### Weakness 1
> Only tested on LRU architecture, not on other SSMs like Mamba, S4, etc.
>
> **Response:**
>
> We thank the reviewer for bringing this point up and they are indeed correct. What we have found challenging with popular SSMs both LTI (S4, S5) and LTV (Mamba, LiquidS4) is not as much the application of our pipeline, for which we have efficient LTV variants (namely averaging), yet it is the fact that they proceed on a per-channel Single-Input Single-Output (SISO) system basis. In fact, SISO dynamical systems are used in many architectures, where instead of having a vector-valued (MIMO) dynamical system predicting the evolution of the entire input feature vector, each individual channel of the input is handled by a SISO system to predict the corresponding channel in the output. Now instead of having a state that affects the mapping of an entire hidden feature of dimension 512 for example to an output of the same dimension, it is used to only predict the behavior of a 1D input-output map. This leads to many situations where the expressiveness and performance gains with larger state dimension are diluted. We discuss this in more detail in the revised Appendix E.2. We also highlight that not all LTV models must be SISO and we are exploring extensions to MIMO LTV SSMs (such as Griffin [1]) and even to matrix-valued dynamical systems akin to Linear Attention models (DeltaNet [2]). We aim to present such solutions as future work.
>
> Nonetheless, this has not prevented us from adapting CompreSSM to Mamba in order to corroborate our SISO vs MIMO observation as well as to demonstrate a practical implementation for such models. The presentation is provided in detail in our revised Appendix E.3.
>
> **Applying CompreSSM to Mamba.**
> Mamba presents unique challenges because its dynamics are input-dependent (selective) and operate on many independent SISO channels that share common projections. Our approach works as follows: During a calibration phase, we collect running averages of the input-dependent matrices $\mathbf{B}(\mathbf{x}_k)$ and $\mathbf{C}(\mathbf{x}_k)$ for each channel. These averages define a time-invariant surrogate system per channel, temporarily decoupling the channels so we can apply standard balanced truncation. For each channel $i$, we compute controllability and observability Gramians, extract Hankel singular values, and determine a reduced rank $r_i$ based on tolerance $\tau$. We then compute and store both the balancing transformation $\mathbf{T}_i$ and its inverse $\mathbf{T}_i^{-1}$.
>
> At runtime, these cached transforms are applied to the fresh input-dependent projections $\mathbf{B}(\mathbf{x}_k)$ and $\mathbf{C}(\mathbf{x}_k)$ at every timestep, rotating them into the balanced coordinate system before truncating to the first $r_i$ dimensions. The transition matrix $\mathbf{A}_i$ is similarly transformed and truncated once during calibration. Critically, we modified Mamba's selective-scan CUDA kernels to accept per-channel ranks so that inner loops terminate at dimension $r_i$ instead of the full state dimension. This ensures computational savings are realized in practice.
>
> **Performance results.**
> On CIFAR-10, baseline Mamba models show stable performance across state dimensions from 128 down to 8, confirming the weak correlation between state size and accuracy in SISO architectures.
> On IMDB, we compare unreduced baselines, CompreSSM with $\tau = 0.001$, and random HSV dropping. CompreSSM reduces from dimension 128 to approximately 14 (9× reduction) while achieving competitive accuracy with lower variance across seeds. Random HSV dropping performs surprisingly well due to the heavy-tailed HSV spectrum (again consistent with SISO discussion), though with higher variance.
>
> **Training speedup.**
> On IMDB, a CompreSSM-reduced model starting at dimension 128 and reduced down to dimension 14 trains in comparable time to the $n = 16$ baseline, representing a 4× speedup versus the full large model. This validates that rank-aware kernel modifications translate dimension reduction into practical computational savings. Speedups are consistently achieved across both LTI (LRU) and LTV (Mamba) architectures.
>
>
> [1] De, Soham et al. “Griffin: Mixing Gated Linear Recurrences with Local Attention for Efficient Language Models.” ArXiv abs/2402.19427 (2024)
>
> [2] Yang, Songlin et al. “Parallelizing Linear Transformers with the Delta Rule over Sequence Length.” ArXiv abs/2406.06484 (2024)

---

> > ### Author Response · Authors · 2025-11-21
> > **Initial response 2**
> >
> > ### Weakness 2
> > No comparison with other compression techniques like knowledge distillation, quantization, or pruning
> >
> > **Response:**
> >
> > We thank the reviewer for giving us the opportunity to expand our experimental results, which corroborate our case for the CompreSSM approach. Indeed, we had initially excluded such techniques as we focused on *in-training* compression, i.e., asking the question: how can we, early during a single run of training, get rid of low-impact dimensions without damaging performance? The methods proposed by the reviewer are post-training approaches, requiring the full training of the large model before applying compression.
> >
> > Nevertheless, we provide a performance and computational cost comparison between CompreSSM and Knowledge Distillation on the CIFAR10 dataset to illustrate the advantages of our approach in Appendix F2. As shown in Table 1, KD performs reasonably when the student model is close in size to the teacher, but its performance drops sharply for the aggressively compressed regimes where CompreSSM remains consistently strong. Moreover, KD pays a *double computational price*: the teacher and student must both be trained, and every student update requires an additional forward pass through the fully sized teacher, effectively executing the most expensive computation twice. In contrast, CompreSSM trains a single model whose computational cost decreases over time, yielding real wall-clock savings rather than compounding overhead.
> >
> > | Method | Metric | 57 | 93 | 126 | 161 | 214 | 327 | 384 |
> > |--------|--------|----|----|-----|-----|-----|-----|-----|
> > | **Baseline** | Accuracy (%) | 78.2 | 81.8 | 83.7 | 84.2 | 84.9 | 86.0 | 86.5 |
> > |  | Speed × | 1.69 | 1.62 | 1.53 | 1.43 | 1.22 | 1.03 | 1.0 |
> > | **CompreSSM** | Accuracy (%) | **84.4** | **85.7** | **86.0** | **85.8** | **86.0** | 86.1 | – |
> > |  | Speed × | 1.58 | 1.52 | 1.41 | 1.33 | 1.17 | 1.03 | – |
> > | **Knowledge Distillation** | Accuracy (%) | 79.4 | 83.5 | 84.4 | 85.3 | **86.0** | **87.0** | – |
> > |  | Speed × | 0.55 | 0.52 | 0.61 | 0.51 | 0.49 | 0.45 | – |
> >
> >
> > ---
> >
> > ### Weakness 3
> > Method requires correlation between state dimension and model performance (acknowledged in the paper)
> >
> > **Response:**
> >
> > We thank the reviewer for reiterating this limitation which we indeed allude to in the text. This is unfortunately beyond our control as principled model order reduction on models that do not depend on the order just does not make sense.
> >
> > ---
> >
> > ### Weakness 4
> > Needs sufficient training steps between reduction steps for model recovery (like pruning and QAT)
> >
> > **Response:**
> >
> > We thank the reviewer for bringing this point up as it allows us to realize that our original choice of words is extremely misleading in suggesting that CompreSSM requires "recovery" after each reduction step; in fact, when applied correctly, no such recovery is needed. Balanced truncation enjoys a classical guarantee: the H∞ error between the original and reduced systems is bounded by twice the sum of the truncated Hankel Singular Values (HSVs). Assuming we properly truncate very low-impact HSVs, the perturbation to the dynamical system is inherently small, and thus the training trajectory continues smoothly without any noticeable loss that would necessitate recovery.
> >
> > This behavior is made explicit in our HSV-selection ablation (Appendix C.1), where validation accuracy remains stable across reduction events so long as the removed modes correspond to small HSVs. In contrast, if one removes impactful directions (either through overly aggressive reduction or through random/adversarial selection) the model suffers immediate and *irreversible* performance drops. Even with 90% of training remaining, such models do not appear to match their full pre-reduction potential.
> >
> > In short, properly applied balanced truncation does *not* require recovery because it does not inflict meaningful damage on the model. Improper or excessive reduction, however, produces damage that subsequent training fails to fully repair. The revised version has been modified to remove this ambiguity.

---

> > > ### Author Response · Authors · 2025-11-21
> > > **Initial response 3**
> > >
> > > ### Weakness 5
> > > Missing ablation studies on key hyperparameters (reduction frequency, threshold values)
> > >
> > > **Response:**
> > >
> > > We thank the reviewer for raising this valid concern, and we agree that understanding the sensitivity of CompreSSM to reduction-related hyperparameters is important. In Appendix C of the revised manuscript we have added a thorough set of ablations that directly address this question.
> > >
> > > The appendix contains three dedicated ablation studies, each isolating one of the key hyperparameters of CompreSSM:
> > >
> > > - **Choice of Hankel singular values to remove (truncation rule).**
> > >   Appendix C.1 demonstrates that proper balanced truncation (removing the *smallest* HSVs) is essential for stable training. Removing random HSVs or, worse, removing the largest HSVs leads to severe and unrecoverable degradation. This confirms that the truncation rule is a structural requirement imposed by the control-theoretic properties of SSMs, which training cannot salvage if reduced too aggressively or improperly.
> > >
> > > - **Number/frequency of reductions.**
> > >   Appendix C.2 varies the number of reductions from 1 to 16 while keeping the same initial and final state dimension. The results show that training is only slightly sensitive to this parameter: smoother, incremental reductions offer some marginal improvements, yet all settings converge to essentially close final accuracies.
> > >
> > > - **Timing of the reduction window.**
> > >   Appendix C.3 shifts the reduction window across training (0–20k, 25–45k, 50–70k, 100–120k, 150–170k). All configurations achieve the same final performance, showing that reductions may be applied early, mid-training, or late with no meaningful difference. This supports our theoretical discussion that HSVs evolve smoothly and preserve relative ordering throughout training.
> > >
> > > Taken together, these ablations show that CompreSSM remains consistently robust across key hyperparameters. The method exhibits a broad region of insensitivity: reductions can be performed early or late, and in few or many steps, without degrading performance so long as the right HSVs are removed in the right proportion. These findings support our design choice in the main text: performing a *small number of early reductions* is both sufficient and preferable. It provides meaningful computational savings by avoiding repeated passes through the largest model, while maintaining its performance.
> > >
> > > Finally, regarding the selection of tolerance levels, we refer the reviewer to our reply to Q1, where we address this point in detail.
> > >
> > >
> > > ---
> > >
> > > ### Weakness 6
> > > No analysis of what information is preserved/lost during truncation
> > >
> > > **Response:**
> > >
> > > We thank the reviewer for bringing this question up. It is difficult to assign an clear semantic sense to "information" that might be lost during truncation, or similarly gained during training. We reposition reduction in the wider context of SSM models. They apply to specific recurrent layers of the models that operate in and around other operators and nonlinearities. Control theory for balanced truncation is clear about the "information" it discards, namely the states that have the smallest observability and controllability, in other words the smallest impact on the input-output map at the level of the SSM layer. The associated $\mathcal{H}_\infty$ bound, provided in Equation (6) in the main text, limits the departure between the original and reduced model. We would require additional clarification from the reviewer regarding the meaning attached to "information" beyond the control theoretic concepts leveraged in this work.

---

> > > > ### Author Response · Authors · 2025-11-21
> > > > **Initial response 4**
> > > >
> > > > ### Question 1
> > > > Step 5 of the algorithm (Section 3.1): What fraction values work well in practice? The paper doesn't provide specific guidance
> > > >
> > > > **Response:**
> > > >
> > > > The issue of optimal reduction scheduling is a very important one which we address in detail here. From a practical point of view, we want to train the best performing compact model as fast as possible. However, we do not have *a priori* access to the performance upper bound which requires training the initial large model to completion (something we aim to avoid). Thus the design criteria cannot rely on comparisons to the potential optimal performance. Elsewhere, the $\mathcal{H}_\infty$ guarantee bound between the original and reduced models (Equation (6) in the main text), and thus control theoretic approaches of relative energy conservation (in the HSV sum sense), provide little useful insights into the relationship between a given reduction at a specific training step down to a specific rank, and the subsequent effect on optimal performance of the fully trained model. This becomes more complex when considering models with multiple blocks all being reduced simultaneously, their effects through other non-linear layers and operations, and global reasoning about performance. Hence, two things can be concluded: (i) it is impossible to reliably predict the effect of a layer reduction on downstream performance *a priori*, and as a result, (ii) performance agnostic reduction inevitably risks producing sub-optimal results (in the sense of performance versus the original large model, as they might still be optimal against other models of same state dimension).
> > > >
> > > > That being said, we provide an alternative CompreSSM approach that inevitably requires violating the open loop irreversible single pass approach of *a priori* tolerance level selection. Section 3.2 of the revised manuscript provides a description of the approach. The idea is simple: since we cannot predict the long term effects of reductions, we only reduce until there are no negative short term damages to performance. For illustration, at fixed intervals we can reduce the model down to a fixed reasonable fraction of its size, say 10%. We save the pre-reduction checkpoint and train the model for a very small number of steps (0.1% of training steps) and evaluate it. If performance still improves we proceed with the reduced model. Else, we revert to the previous model and resume training without reductions. This pragmatic approach still benefits from similar speedup while ensuring we avoid applying too aggressive of reductions which can be detrimental to performance.
> > > >
> > > >
> > > > ### Question 2
> > > > Can you provide examples where there's no clear correlation between state dimension and model performance? I cannot think of such applications
> > > >
> > > > **Response:**
> > > >
> > > > The reviewer brings up an important point for which there are answers at multiple levels. First, an empirical instance, among the LRA experiments considered with LRU models, we can point the reviewer to the AAN experiment. The best way to visualize the lack of real dependency on state dimension is looking at Figure 4 in Appendix B, which illustrates the numerical results of Table 1 in Section 4.2 of the main text. What it shows is that whether we set the state dimension to 256, 111, or 54, the performance is not affected.
> > > >
> > > > A more nuanced answer requires distinguishing per-channel Single-Input Single-Output (SISO) models from true Multi-Input Multi-Output (MIMO) SSMs. We provide a detailed discussion in Appendix E.2. In many architectures, each feature channel is processed by an independent SISO dynamical system (like Mamba), rather than using a single vector-valued MIMO system that jointly evolves all features (like LRU). Consequently, the state of a SISO model influences only a 1D input–output mapping, whereas in a MIMO SSM a single state vector may govern the evolution of an entire 512-dimensional feature. This structural difference dilutes the expressive benefits of increasing the state dimension in SISO settings. The LRA experiments on Mamba in Section E.3.3 of the revised Appendix also illustrate this point.
> > > >
> > > > ---
> > > >
> > > > ### Question 3
> > > > Are you evaluating Sequential MNIST (SMNIST)? The paper mentions MNIST results but doesn't clarify if it's SMNIST
> > > >
> > > > **Response:**
> > > >
> > > > We appreciate the reviewer's attention to detail and confirm that the experiment is indeed the sequential MNIST task where the images to classify are flattened along the pixel dimension and fed into the SSMs as a sequence. We have added the "s" back in sMNIST to avoid any confusion.

---

> > > > > ### Author Response · Authors · 2025-11-21
> > > > > **Initial response 5**
> > > > >
> > > > > ### Question 4
> > > > > How does the computational cost of computing Gramians scale with state dimension?
> > > > >
> > > > > **Response:**
> > > > >
> > > > > We thank the reviewer for the question and inform them that we have now incorporated Section D in the revised Appendix where a detailed discussion on computation complexity and timing is presented.
> > > > >
> > > > > The key point is that, in our setting, *computing the Gramians is not the computational bottleneck*. For diagonal transition matrices such as in LRU, the Gramians admit closed-form expressions (Appendix D, Eq. (32)), so forming them only requires computing $\mathbf B \mathbf B^\top$ and $\mathbf C^\top \mathbf C$, with cost $\mathcal{O}(n^2 p)$ and $\mathcal{O}(n^2 q)$, respectively. This is quadratic in the state dimension although $p$ (input dim) and $q$ (output dim) could also be of the same order of magnitude.
> > > > >
> > > > > Another important computational cost that is unavoidable for any balanced truncation method is the diagonalization step: computing the Hankel singular values amounts to performing an SVD of $\mathbf P \mathbf Q$, which scales as $\mathcal{O}(n^3)$. Likewise, transforming the system into its balanced coordinates requires another $\mathcal{O}(n^3)$ operation. These cubic steps also heavily contribute to the practical cost of the reduction pipeline.
> > > > >
> > > > > Importantly, these punctual cubic overheads are *independent of sequence length* and are quickly amortized during training, since subsequent gradient steps operate on the reduced model. As shown in Section 5 and Appendix D, the resulting speedups dominate the reduction cost even for moderate compression ratios.
> > > > >
> > > > > ---
> > > > >
> > > > > ### Question 5
> > > > > Have you considered inverting the x-axis in Figure 3 to show "training time speedup" instead of "normalized training time"? This would make the Pareto front point to the top-right, which is more intuitive
> > > > >
> > > > > **Response:**
> > > > >
> > > > > We thank the reviewer for this good suggestion which has been incorporated into the revised version and allowed us to spot and correct an issue with the time computation of one of the experiments.

---

> > > > > > ### Author Response · Authors · 2025-11-21
> > > > > >
> > > > > > We sincerely hope that we have addressed the concerns of the reviewer satisfactorily in the revised version and would kindly ask the reviewer to update their score accordingly.

---

> ### Author Response · Authors · 2025-11-27
> **Invitation to engage**
>
> We are grateful for your insightful comments and your time dedicated to our submission. We have put considerable effort into incorporating your suggestions into the revised manuscript, adding substantial experiments and analysis to meaningfully improve the quality and depth of the work.
>
> Specifically we have addressed the case of selective state space models with a deep dive into the Mamba case (implementation, experiments and analysis), have added a comparison to compression via Knowledge Distillation, clarified the (lack of) recovery requirement of CompreSSM, included 3 insightful ablations on CompreSSM parameters, introduced a pragmatic alternative to the tolerance based initialization, and discussed time and compute complexity in depth in a dedicated section.
>
> We very much welcome your engagement with any further questions or thoughts.

---

### Official Review · Reviewer_qcBm · 2025-11-06

**Soundness:** 2
**Presentation:** 3
**Contribution:** 3
**Rating:** 6
**Confidence:** 3

**Summary:**

The paper introduces CompreSSM, a generalized in-training compression framework for state space models to accelerate training while maintaining accuracy. The method uses Hankel singular value analysis to identify and prune low-energy state dimensions during training (inspired by Model Order Reduction and balanced truncation from control theory). The authors show that the rank ordering of Hankel singular values remains relatively stable throughout training. The authors use six datasets to provide empirical evidence for its ability to maintain or even improve accuracy as compared to a less efficient baseline training.

**Strengths:**

The authors provide good theoretical grounding for CompreSSM, bringing concepts from Hankel singular value analysis and truncation; they address a very relevant topic with a well-outlined algorithm.

Authors provide reasonable justification of the in-training truncation and its validity. Analysis of Hankel singular value trajectories supports their analysis.

The authors provide convincing results across six different datasets and five random seeds, while describing the experimental setup in great detail facilitating reproducibility.

**Weaknesses:**

If I understand correctly, empirical evidence is centered only around LRU, without other Deep SSMs or so-called "selective" (input dependent) architectures.

The authors should provide additional analysis or detail on the computational complexity of the Algorithm (3.1) and overhead/clock time of each reduction step, more details into the recovery of the model after each reduction step (mentioned in 4.2 as part of the results, but not analyzed as far as I could tell) and how this translated into end-to-end training cost and efficiency on GPUs versus the vanilla method.

**Questions:**

N/A

---

> ### Author Response · Authors · 2025-11-21
> **Initial response 1**
>
> ### Weakness 1
> Empirical evidence is centered only around LRU, without other Deep SSMs or so-called "selective" (input dependent) architectures.
>
> **Response:**
>
> We thank the reviewer for bringing this point up and they are indeed correct. What we have found challenging with popular SSMs both LTI (S4, S5) and LTV (Mamba, LiquidS4) is not as much the application of our pipeline, for which we have efficient LTV variants (namely averaging), yet it is the fact that they proceed on a per-channel Single-Input Single-Output (SISO) system basis. In fact, SISO dynamical systems are used in many architectures, where instead of having a vector-valued (MIMO) dynamical system predicting the evolution of the entire input feature vector, each individual channel of the input is handled by a SISO system to predict the corresponding channel in the output. Now instead of having a state that affects the mapping of an entire hidden feature of dimension 512 for example to an output of the same dimension, it is used to only predict the behavior of a 1D input-output map. This leads to many situations where the expressiveness and performance gains with larger state dimension are diluted. We discuss this in more detail in the revised Appendix E.2. We also highlight that not all LTV models must be SISO and we are exploring extensions to MIMO LTV SSMs (such as Griffin [1]) and even to matrix-valued dynamical systems akin to Linear Attention models (DeltaNet [2]). We aim to present such solutions as future work.
>
> Nonetheless, this has not prevented us from adapting CompreSSM to Mamba in order to corroborate our SISO vs MIMO observation as well as to demonstrate a practical implementation for such models. The presentation is provided in detail in our revised Appendix E.3.
>
> **Applying CompreSSM to Mamba.**
> Mamba presents unique challenges because its dynamics are input-dependent (selective) and operate on many independent SISO channels that share common projections. Our approach works as follows: During a calibration phase, we collect running averages of the input-dependent matrices $\mathbf{B}(\mathbf{x}_k)$ and $\mathbf{C}(\mathbf{x}_k)$ for each channel. These averages define a time-invariant surrogate system per channel, temporarily decoupling the channels so we can apply standard balanced truncation. For each channel $i$, we compute controllability and observability Gramians, extract Hankel singular values, and determine a reduced rank $r_i$ based on tolerance $\tau$. We then compute and store both the balancing transformation $\mathbf{T}_i$ and its inverse $\mathbf{T}_i^{-1}$.
>
> At runtime, these cached transforms are applied to the fresh input-dependent projections $\mathbf{B}(\mathbf{x}_k)$ and $\mathbf{C}(\mathbf{x}_k)$ at every timestep, rotating them into the balanced coordinate system before truncating to the first $r_i$ dimensions. The transition matrix $\mathbf{A}_i$ is similarly transformed and truncated once during calibration. Critically, we modified Mamba's selective-scan CUDA kernels to accept per-channel ranks so that inner loops terminate at dimension $r_i$ instead of the full state dimension. This ensures computational savings are realized in practice.
>
> **Performance results.**
> On CIFAR-10, baseline Mamba models show stable performance across state dimensions from 128 down to 8, confirming the weak correlation between state size and accuracy in SISO architectures.
> On IMDB, we compare unreduced baselines, CompreSSM with $\tau = 0.001$, and random HSV dropping. CompreSSM reduces from dimension 128 to approximately 14 (9× reduction) while achieving competitive accuracy with lower variance across seeds. Random HSV dropping performs surprisingly well due to the heavy-tailed HSV spectrum (again consistent with SISO discussion), though with higher variance.
>
> **Training speedup.**
> On IMDB, a CompreSSM-reduced model starting at dimension 128 and reduced down to dimension 14 trains in comparable time to the $n = 16$ baseline, representing a 4× speedup versus the full large model. This validates that rank-aware kernel modifications translate dimension reduction into practical computational savings. Speedups are consistently achieved across both LTI (LRU) and LTV (Mamba) architectures.
>
>
> [1] De, Soham et al. “Griffin: Mixing Gated Linear Recurrences with Local Attention for Efficient Language Models.” ArXiv abs/2402.19427 (2024)
>
> [2] Yang, Songlin et al. “Parallelizing Linear Transformers with the Delta Rule over Sequence Length.” ArXiv abs/2406.06484 (2024)

---

> > ### Author Response · Authors · 2025-11-21
> > **Initial response 2**
> >
> > ### Weakness 2
> > Details on the computational complexity of the algorithm and overhead/clock time of each reduction step
> > Details into the recovery of the model after each reduction step
> > How this translates into end-to-end training cost and efficiency on GPUs versus the vanilla method.
> >
> > **Response:**
> >
> > We thank the reviewer for raising this important point. To address it, we added a dedicated section in the Appendix (Section D) that provides a full breakdown of the computational costs of Algorithm 3.1 and the reduction pipeline. This includes:
> > (i) empirical timing experiments (gradient-step time, inference time, and per-stage reduction overhead),
> > (ii) a formal wall-clock model that quantifies the resulting speedups, and
> > (iii) a full complexity analysis of the reduction pipeline, including Gramian computation, HSV extraction, and balanced truncation (see Appendix D.3).
> >
> > Crucially, the updated analysis makes explicit that the dominant source of training acceleration comes from the substantial reduction in gradient-step time once the state dimension is reduced. Since backpropagation through the SSM core scales with state size, even moderate reductions compound over thousands of training iterations to yield significant total savings. The new section makes this tradeoff explicit and transparent.
> >
> >
> > **Calrifying recovery.**
> >
> > Our original wording may have unintentionally implied that CompreSSM requires recovery after each reduction step. In fact, when applied correctly, no such recovery is needed. Balanced truncation enjoys a classical guarantee: the H∞ error between the original and reduced systems is bounded by twice the sum of the truncated Hankel singular values (HSVs). Assuming we properly remove only low-impact HSVs, the perturbation to the dynamical system is very small, and training continues smoothly without any noticeable performance drop.
> >
> > This behavior is made explicit in our HSV-selection ablation (Appendix C.1), where validation accuracy remains stable across reduction events so long as only small HSVs are discarded. In contrast, removing impactful directions (e.g., overly aggressive reduction or random/adversarial selection) causes immediate and irreversible performance drops. Even with 90% of training remaining, such models do not recover their full potential.
> >
> > In short:
> > Proper balanced truncation does **not** require recovery.
> > Improper or excessive reduction produces damage that training cannot fully repair.
> > The revised manuscript has been updated for clarity on this point.

---

> > > ### Author Response · Authors · 2025-11-21
> > >
> > > We sincerely hope that we have addressed the concerns of the reviewer satisfactorily in the revised version and would kindly ask the reviewer to update their score accordingly.

---

> ### Author Response · Authors · 2025-11-27
> **Invitation to engage**
>
> We thank you again for your insightful comments and your time dedicated to our submission. We have put considerable effort into incorporating your suggestions into the revised manuscript, adding substantial experiments and analysis to meaningfully improve the quality and depth of the work.
>
> Specifically we have addressed the case of selective state space models with a deep dive into the Mamba case (implementation, experiments and analysis) and have broken down the computational costs in a dedicated section, in addition to clarifying the (lack of) recovery requirement of CompreSSM.
>
> We very much welcome your engagement with any further questions or thoughts.

---

### Author Response · Authors · 2025-11-21
**Global Reply to All Reviewers**

We would like to sincerely thank all the reviewers for their careful reading of our paper and their constructive feedback.
It is encouraging to see that the reviewers appreciated the novelty, rigor, and clarity of our contribution.
All reviewers acknowledged the relevance of addressing *in-training compression for State Space Models (SSMs)* and the sound theoretical grounding of our method.

Indeed, the reviewers recognized the paper’s technical merit and originality:
“good theoretical grounding for CompreSSM, bringing concepts from Hankel singular value analysis and truncation” **[qcBm]**,
“novel application of control theory to SSM compression during training” **[p3jH]**,
and “the possibility of compressing SSMs during training in a principled way may help find a trade-off between computational costs and flexibility” **[vMHR]**.
We also thank reviewer **[ELEM]** for emphasizing that “accelerating the training speed of SSMs is an important task, and the proposed method shows the benefits of the method over directly training a smaller model from scratch.”

We also appreciate the reviewers’ recognition of the methodological soundness and empirical validation:
“authors provide reasonable justification of the in-training truncation and its validity” **[qcBm]**,
“the method is architecture-agnostic and works with different SSM variants” **[p3jH]**,
“empirical validation showing that dominant HSVs are rank-preserving during training” **[p3jH]**,
and “the method is grounded in rigorous analysis of LTI systems and the balanced truncation method from the ROM literature” **[ELEM]**.
We are also encouraged by the positive feedback on presentation quality and reproducibility:
“well-written paper with clear mathematical exposition” **[p3jH]** and
“results across six datasets and five random seeds, while describing the experimental setup in great detail facilitating reproducibility” **[qcBm]**.

In the individual replies below, we address the specific concerns raised by each reviewer.
We also highlight the main additions to the revised version (RV) of the manuscript (tracked in blue), which include:

1. Proposed pragmatic variant to the tolerance-based hyperparameterization
2. Detailed analysis of computational complexity and time-budget breakdown as a function of state dimension
3. Extended discussion on state expressivity in SISO and MIMO systems
4. LTV case study of Mamba, with implementation details and experiments
5. Hankel Nuclear Norm regularization and truncation experiment
6. Teacher–student Knowledge Distillation experiment
7. Ablation studies on HSV selection and recovery, number of reductions, and reduction timing

**Note:** All references to Sections, Figures, and Tables refer to those in the revised manuscript.

---

### Author Response · Authors · 2025-11-30
**Post-leak Summary for Chair**

### **A) Recognition and apology**

We regret the unusual situation, which is very unsettling not just for us as authors but also for the chairs given the extra work imposed on them. We thank you for your time and service, and wish to facilitate your task with a summary of the proceedings so far.

During the rebuttal period, we have gone to great lengths to make significant improvements to the manuscript in direct response to the reviewers’ feedback. We believe it will be clear to the chair that the revised version brings forth substantial new insights supported by novel experiments, ablations, and analysis. We are confident that we have thoroughly addressed all reviewer questions and concerns, and we apologize for the volume of additions now placed before you.

---

### **B) One Sentence Paper Summary**

We implement **CompreSSM**, an in-training compression scheme for SSMs that relies on balanced truncation as its underlying mechanism, motivated by mathematical machinery that allows us to track the significance of each state dimension throughout training.

---

### **C) Appreciation**

There is a near-consensus among reviewers regarding the work’s technical merit, originality, technical soundness, and empirical validation. We point the chair to our global reply for a summary of reviewer quotes supporting those aspects of our contribution.

---

### **D) Exchanges with reviewers**

We are genuinely frustrated with the abrupt conclusion of the discussion period with the reviewers. We were eagerly looking forward to engaging in enriching exchanges with all reviewers given our commitment to implementing their suggestions and clarifying their questions.

This was (partially) achieved with the fourth reviewer [ELEM] who had expressed doubts about the novelty and significance of the work, with some divergence about where the important contribution would lie. We have vehemently and rigorously defended our position, backing our arguments with results from the newly added ablations, while taking into account the valuable perspective of reviewer ELEM. We had thus far managed to convince the reviewer to **commit to a score increase** (from 2 to 4, in writing in their last comment). Furthermore, based on their ultimate comment, we are confident that our reply (which unfortunately will not reach the reviewer) would drive the point fully home.

---

> ### Author Response · Authors · 2025-11-30
> **Post-leak Summary for Chair (continued)**
>
> ### **E) Main Avenues for Improvement and How We Addressed Them**
>
> The work has received valuable suggestions for improvement and thoughtful questions of high quality. We focus on novel results and analysis that address the former here and refer the chair to the replies to the reviewers for the detailed answers to the latter (for this to remain a summarization exercise).
>
> ---
>
> ### **1. Lack of Experimental Study for Selective SSMs**
> **Reviewers:** qcBm, p3jH, vMHR, ELEM
>
> All reviewers raised this valid point. A full section (E) in the new appendix has been dedicated to this question. Our response is twofold:
>
> **a. Clarifying SISO vs. MIMO distinctions**
> We nuance the discussion between *per-channel SISO systems* (e.g., **S4**, **Mamba**) and *full input-to-output MIMO models* (e.g., **LRU**, **Griffin**), emphasizing that this distinction is more consequential for understanding the effects of model-order reduction than the selective vs. LTI system design. Indeed, the role of the state dimension is diluted for SISO systems and conclusions are often (and more specifically on LRA tasks) harder to draw.
>
> **b. Full Mamba case study**
> We include a complete case study of **Mamba**, outlining the full implementation procedure and presenting numerical experiments that empirically support the claims made in point (a).
>
> ---
>
> ### **2. Unclear speedup / compute complexity gains**
> **Reviewers:** qcBm, p3jH, ELEM
>
> There has been some confusion as well as questions relating to the speedup gains as well as the complexity of algebraic operations involved in our solution. To address this we have added a new dedicated Section (D) in the Appendix that provides a full breakdown of the computational costs of the reduction pipeline. This includes:
> (i) empirical timing experiments (gradient-step time, inference time, and per-stage reduction overhead),
> (ii) a formal wall-clock model that quantifies the resulting speedups, and
> (iii) a full complexity analysis of the reduction pipeline, including Gramian computation, HSV extraction, and balanced truncation.
>
> ---
>
> ### **3. Tricky tolerance and hyperparameter tuning**
> **Reviewers:** p3jH, vMHR, ELEM
>
> Reviewers rightly mentioned that tuning the CompreSSM hyperparameters, namely the tolerance level, and the number/frequency/interval of reductions can be tricky to optimize. We detail a pragmatic alternative approach in the revised manuscript, based on short term performance monitoring. This is backed by empirical observations drawn from ablations discussed below.
>
> ---
>
> ### **4. Absence of Ablations**
> **Reviewers:** p3jH
>
> The reviewers raised the point about lacking ablations which we unfortunately did not manage to include in the initial submission. We rectify our shortcoming and present 3 ablations in the new Section C of the revised appendix:
>
> The appendix contains three dedicated ablation studies, each isolating one of the key hyperparameters of CompreSSM:
>
> 1. **Choice of Hankel singular values to remove (truncation rule):** the truncation rule is a structural requirement imposed by the control-theoretic properties of SSMs, which training cannot salvage if reduced too aggressively or improperly.
> 2. **Number/frequency of reductions:** smoother, incremental reductions offer some marginal improvements, yet all settings converge to essentially close final accuracies.
> 3. **Timing of the reduction window:** no observed differences with shifts of the reduction window reaching same final performance. This supports our theoretical discussion that HSVs evolve smoothly and preserve relative ordering throughout training.
>
> ---
>
> ### **5. No comparison to other compression techniques**
> **Reviewers:** p3jH
>
> We extended our experiments to include a comparison with Knowledge Distillation, even though it is a post-training method (so is pruning) and thus outside our original in-training focus. The results (new Appendix F2) show that KD performs well only for mild compression and becomes unstable under aggressive reductions, while also incurring roughly double the computational cost due to requiring successive teacher then student training. In contrast, CompreSSM compresses during a single training run and reduces compute over time, offering both stronger performance and genuine wall-clock savings.
>
> ---
>
> ### **6. No comparison to regularization techniques**
> **Reviewers:** vMHR
>
> The only principled regularization technique that acts on the systems in a control theoretic sound fashion is Hankel Nuclear Norm Regularization which has been studied in the literature. Our added experiments in Section F.1 of the revised appendix showcase the twofold issues of HNN regularization: (i) computationally prohibitive, requiring expensive Hankel eigenvalue computations at every gradient step (~46× slower), (ii) underperforming, regularization contraint prevents the model from even reaching baseline accuracy.

---

### Meta-Review · Area_Chair_jQuz · 2026-01-07

**Summary:**

The paper propose an interesting approach for pruning the size of Mamba throughout training such that it starts with a big model and ends up with a smaller and more efficient model. They compare to distillation and perform a thorough analysis of their approach. The work is novel and interesting and provide a useful tool for the research community

**Reviewer Concerns:**

Reviews raised many concerns with respect to complexity, comparison to other works, parameter tuning, etc. All main questions were answered in the rebuttal in a good way.

**Reviewer Scores:**

I believe the reviewers would have raised their score in a way that would merit acceptance. I therefore recommend accepting the paper given the main concerns were answered in an adequate way.

---

### Decision · Program_Chairs · 2026-01-26

Accept (Poster)